# Implications from palaeoseismological investigations at the Markgrafneusiedl Fault (Vienna Basin, Austria) for seismic hazard assessment

Esther Hintersberger[1], Kurt Decker[1], Johanna Lomax[2,3], Christopher Lüthgens[2]

[1]Department of Geodynamics and Sedimentology, University of Vienna, 1090 Vienna, Austria
[2]Institute of Applied Geology, University of Natural Resources and Life Sciences (BOKU), 1190 Vienna, Austria
[3]Department of Geography, Justus Liebig University Gießen, 35390 Giessen, Germany

*Correspondence to*: Esther Hintersberger (esther.hintersberger@univie.ac.at)

**Abstract.** Intraplate regions characterized by low rates of seismicity are challenging for seismic hazard assessment, mainly for two reasons: Firstly, evaluation of historic earthquake catalogues may not reveal all active faults that contribute to regional seismic hazard. Secondly, slip rates determination is limited by sparse geomorphic preservation of slowly moving faults. In the Vienna Basin (Austria), moderate historical seismicity (Imax/Mmax = 8/5.2) concentrates along the left-lateral strike-slip Vienna Basin Transfer Fault (VBTF). In contrast, several normal faults branching out from the VBTF show neither historical nor instrumental earthquake records, although geomorphological data indicate Quaternary displacement along those faults. Here, located about 15 km outside of Vienna, the Austrian capital, we present a palaeoseismological dataset of three trenches that cross one of these splay faults, the Markgrafneusiedl Fault (MF), in order to evaluate its seismic potential. Comparing the observations of the different trenches, we found evidence for 5-6 surface-breaking earthquakes during the last 120 ka, with the youngest event occurring at around ~14 ka ago. The derived surface displacements lead to magnitude estimates ranging between 6.2±0.5 and 6.8±0.4. Data can be interpreted by two possible slip models, with slip model 1 showing more regular recurrence intervals of about 20-25 ka between the earthquakes with M≥6.5, and slip model 2 indicating that such earthquakes cluster in two time intervals in the last 120 ka. Direct correlation between trenches favours slip model 2 as the more plausible option. Trench observations also show that structural and sedimentological records of strong earthquakes with small surface offset have only low preservation potential. Therefore, the earthquake frequency for magnitudes between 6 and 6.5 cannot be constrained by the trenching records. Vertical slip rates of 0.02-0.05 mm/a derived from the trenches compare well to geomorphically derived slip rates of 0.02-0.09 mm/a. Magnitude estimates from fault dimensions suggest that the largest earthquakes observed in the trenches activated the entire fault surface of the MF including the basal detachment that links the normal fault with the VBTF. The most important implications of these paleoseismological results for seismic hazard assessment are that: (1) The MF is an active seismic source, capable of rupturing the surface, despite the lack of historical earthquakes. (2) The MF is kinematically and geologically equivalent to a number of other splay faults of the VBTF. It is reasonable to assume that these faults are potential sources of large earthquakes as well. The frequency of strong earthquakes near Vienna is therefore expected to be significantly higher than the earthquake frequency reconstructed for the MF alone. (3)

Although rare events, the potential for earthquake magnitudes equal or greater than M=7.0 in the Vienna Basin should be considered in seismic hazard studies.

## 1 Introduction

During the last few years, earthquakes have tended to "surprise" the scientific community, either by unexpectedly high magnitudes (e.g., Sumatra Earthquake 2004 and Tohuko Earthquake 2011, Chlieh et al., 2007, Geller, 2011) or/and by the fact that the generating faults were either unmapped (Darfield Earthquake 2010, Gledhill et al., 2011) or not considered in hazard assessments (e.g., Haiti Earthquake 2010, Giardini et al., 1999). Thus, it seems to be clear that historical and instrumental seismicity data are not sufficient to fully characterize the seismogenic potential of a certain region, especially in regions of low to moderate seismicity (e.g., Camelbeeck et al., 2007, Liu et al., 2011, Landgraf et al., 2017). Therefore, during the last decade, geomorphological and palaeoseismological approaches have been increasingly used to map active faults and to determine the related slip rates (e.g., Clark et al., 2012, in Australia, and Vanneste et al., 2013, for the Lower Rhine graben system in Central Europe). The results of those studies have dramatically changed the picture and the estimation of seismogenic potential in the analysed regions, mainly in the following aspects: Firstly, palaeoseismological results show that the magnitude for the maximum credible earthquake may be significantly higher than the magnitude of the largest earthquake observed during historical times (e.g, Central Europe north of the Alps, Figure 1B and references mentioned there). Secondly, the number of active faults that are considered to be capable of generating earthquakes increased by additional investigations (e.g., Clark et al., 2012, in Australia). The identification of such "silent" faults as potential seismic sources has become a vital aspect of geological contribution to seismic hazard assessment. Finally, another important question is whether faults (especially single faults within fault systems) show regular earthquake patterns during time (quasi-periodic earthquakes occurring in more or less regular time intervals) or if earthquakes occur in clusters, where periods of high activity alternate with intervals of seismic quiescence (Wallace, 1987, Friedrich et al., 2003).

Historical and instrumental seismicity in the Vienna Basin (Austria) between the eastern margin of the European Alps and the Carpathians (see Figure 1A) concentrates at the main active strike-slip fault, the Vienna Basin Transfer Fault (VBTF). On the other hand, several Quaternary active normal faults are part of the same presently strain-accumulating fault system, but seem to be seismically "silent". Therefore, the Vienna Basin provides a good example for firstly, addressing the question whether historical and instrumental seismicity are enough to identify active faults and secondly, analysing the behaviour of slowly moving faults in a fault system. Here, we present results of a paloseismological study along a Quaternary active fault close to the city of Vienna. Even though there is no historical nor instrumental seismicity that has been recorded along this fault, three trenches across the fault show evidence for 5-6 surface-breaking earthquakes. Correlation between the trenches and integration of geomorphological and borehole data helps putting weight between the two hypotheses of quasi-periodic or clustered behaviour of the MF.

## 2 Geological setting

### 2.1 The Vienna Basin

The Vienna Basin formed as a pull-apart basin between the Eastern Alps and the Western Carpathians in the Middle and Upper Miocene (e.g. Royden, 1985; Decker et al., 2005). It is located between two left-stepping segments of the NE-SW striking sinistral strike-slip Vienna Basin Transfer Fault (VBTF, Figure 1A). Faulting along this fault system is related to NE-directed movement of the block east of the Vienna Basin, caused by lateral extrusion of the central Eastern Alps towards the Pannonian Basin (Ratschbacher et al, 1991, Linzer et al, 1997, 2002). GPS data (Grenerczy et al., 200Figure5, Métois et al., 2015) indicate that the VBTF moves at horizontal velocities in the order of 1 mm/y or less, whereas geological reconstruction of Quaternary sediment deposition within the basin (Decker et al., 2005) suggests slip rates between 1.6 and 2.5 mm/y. Seismic slip rates calculated from cumulative scalar seismic moments for different segments along the fault are heterogeneous, varying from 0.5-1.1 mm/a at the southern and northern tips to an apparently locked segment in the central part of the basin, close to the city of Vienna (Hinsch and Decker, 2003, 2011). Fault mapping using 2D/3D reflection seismic, gravity, and geomorphology shows that these seismotectonically defined segments are delimited by major fault bends including a restraining bend at the northern end of the Vienna Basin and three releasing bends with negative flower structures overlain by Pleistocene pull-apart basins with up to 150 m of growth strata (Beidinger and Decker, 2011). The releasing bends are connected via pure strike-slip segments.

In addition to the overall geometry of the strike-slip fault with releasing and restraining bends, the transfer of displacement to several normal faults splaying from the strike-slip system at the edges of the releasing bends appears to be an important factor controlling fault segmentation (Beidinger and Decker, 2011). The splay faults were generated during the Middle to Upper Miocene formation of the Vienna pull-apart basin (Decker et al., 2005) and are kinematically linked to the VBTF via a common detachment (i.e., the Alpine floor thrust, Figure 2, Hölzel et al., 2010, Hinsch and Decker, 2011, Beidinger and Decker, 2011). The secondary splay normal faults at the central part of the Vienna Basin are seismically inactive according to instrumental and historical seismicity data (Figure 1A). However, geomorphologic and subsurface geophysical data reveal that these faults record tens of meters of Quaternary displacement (Chwatal et al., 2005; Decker et al., 2005, Weissl et al., 2017). Paleoseismological trenching in the Vienna Basin was carried out so far at one of the splay normal faults, the Aderklaa-Bockfliess Fault (ABF in Figure 1A); the trench did not expose the fault, but the displacement of the base of Quaternary sediments shown in combined electric resistivity measurements and remote sensing analysis suggests a Quaternary slip rate of 0.05 mm/a for this fault (Weissl et al., 2017).

Moderate historical and instrumental seismicity (maximum magnitude Mmax ~ 5.3/ maximum epicentral intensity Imax = 8) is concentrated along the VBTF with the 1972 Seebenstein (M~5.3), 1906 Dobra Voda (M~5.7), and the ~ AD 350 Carnuntum (M~6) earthquakes being the largest known events (Gutdeutsch et al., 1987; Decker et al., 2006; Lenhardt et al., 2007). The scarcity of strong earthquakes and the generally low to moderate seismicity result in Mmax estimations for the Vienna Basin of M = 6.0 to 6.5 (Lenhardt et al., 1995; Procházková and Šimunek, 1998; Šefara et al., 1998; Tóth et al., 2006). However,

these estimations are based solely on historical and instrumental seismicity and may therefore not reflect the long-term seismic potential of the Vienna Basin.

## 2.2 The Markgrafneusiedl Fault (MF)

Our palaeoseismological study is focused on the SE-dipping Markgrafneusiedl Fault (MF) in the central part of the Vienna Basin (Figure 3A). It is one of six normal splay faults that were formed during the Middle to Upper Miocene to accommodate transtension at a releasing bend of the VBTF (Beidinger and Decker, 2011). The location of the fault, Miocene fault displacement and fault dimensions are evident from 2D and 3D industry seismic (Hinsch et al., 2005, Spahic et al., 2013). An exemplary seismic section is shown in Figure 3C. Detailed observations based on 3D industry seismic data on the fault plane suggests that movement along the MF started on different fault segments that eventually merged together as one larger fault (Spahic et al., 2013). Quaternary fault reactivation is inferred from geomorphological evidence of a 12 m high linear scarp paralleling the outcrop trace of the fault, high-resolution geophysical profiling (ground-penetrating radar, reflection seismic, geoelectrics; Chwatal et al., 2005) and the ca. 40 m offset of the base of the Quaternary sediments across the MF (Decker et al., 2005, Figure 3B). The visible fault scarp coincides with the SE edge of the Pleistocene Gaenserndorf terrace (Figures 3A and 4A-D). The Gaenserndorf terrace (GDT) is a large river terrace north of the Holocene flood plain consisting of coarse gravels in sandy matrix and sandy deposits typical for braided river systems (Weissl et al., 2017 and references therein). IRSL dating suggest a minimum deposition age of 200 – 300 ka (Weissl et al., 2017). Despite the well documented Quaternary displacement along the MF, no historical seismicity is associated with this fault. Whether this apparently slowly moving fault can produce larger earthquakes is the key question of our study. Three trenches (from north to south WAG, SDF1, and SDF3) were excavated across the MF between the villages of Markgrafneusiedl and Gaenserndorf, about 15 km from the city limits of Vienna, the Austrian capital (Figures 3A and 4A).

## 3 Methodology

### 3.1 Field work

Prior to siting the trenches SDF1 and SDF3, 40 MHz ground penetration radar (GPR) profiles were carried out to locate the MF at the base of the present-day scarp (see location in Figure 4A and profiles across the fault scarp at the trench locations in C and D). The chosen trench sites are located in a forested area to minimize anthropogenic influence (Figure 4B). The approximately N(W)-S(E) trending trenches were excavated for about 40-50 m with the NE wall benched. Both walls were covered by a grid of 0.5 m x 0.5 m with rows named from A (top) to J and L (bottom), respectively, and columns counted from 1 at the SE end upwards to the NW end. Photomosaics of all walls were obtained by taking pictures of two grid rectangles (1 m x 0.5 m) and subsequent manually ortho-rectifying and stitching in ArcGIS. Trench logging was conducted at a scale of 1:10 for the entire (S)W walls of the trenches and the section around the fault zone on the (N)E walls.

## 3.2 Luminescence dating

Luminescence dating is commonly used to date the time that feldspar grains in sandy or silty sediments were not exposed to sunlight, and therefore to constrain deposition ages of those sandy or silty sediment bodies. Regarding the physical background and the basics of luminescence dating methods we refer to previously published reviews of Preusser et al. (2008), Wintle (2008), and Rhodes (2011). In analogy to the procedure described by Weissl et al. (2017), samples were collected in the field by driving an opaque steel cylinder into the freshly cleaned sediment surface and transferring the material into light tight plastic bags. All subsequent sample preparation steps were conducted under subdued red-light conditions in the Vienna laboratory for luminescence dating. Samples were first dried and dry sieved. The grain size fraction of 100 - 200 µm was used for further preparation steps. Potassium-rich feldspar was used as luminescence dosimeters for age determination. All fractions were measured with multi-grain aliquots of 1 mm diameter mask size. All measurements for determination of the equivalent dose were conducted in the Vienna laboratory for luminescence dating on RISØ TL-OSL DA 20 automated luminescence reader systems (Bøtter-Jensen et al., 2000, 2003). For De determination of the feldspar fraction, a conventional SAR IRSL protocol was applied (Wallinga et al., 2000; Blair et al., 2005), using a preheat of 250°C for 20 s and a stimulation at 50°C for 300 s. Stimulation was carried out with IR-LEDs, and signals were detected after passing through a blue interference filter (410 ± 20 nm). Over-dispersion (Galbraith et al. 1999) was below 11% in all samples confirming a generally well-bleached nature of the sediments. Radionuclide concentrations for dose-rate estimation were determined on ~900 g of bulk sediment using high resolution, low-level gamma-spectrometry. Samples were first dried, homogenised and stored in sealed Marinelli beakers (500 ml, about 1 kg dry weight) for at least a month to establish secondary secular radon equilibrium. Measurements were conducted using a Canberra HPGe detector (40% n-type). Relevant luminescence data is listed in Table 1.

It needs to be stressed that the feldspar based ages were not corrected for fading. Fading describes an anomalous signal loss very commonly observed for potassium-rich feldspar (Wintle, 1973). If not corrected for, fading leads to the underestimation of the burial age. However, samples from the same study area investigated by Weissl et al. (2017) showed little or no fading, as demonstrated by a comparison between quartz and feldspar luminescence ages. Nevertheless, all ages presented here need to be treated with caution for potential age underestimation.

## 3.3 Modelling occurrence times and recurrence intervals

For the calculation of earthquake occurrence times and recurrence intervals, we used the Bayesian statistical computer program OxCal v4.3.2 (Bronk Ramsey, 1995, 2001). Each trench is modelled separately as a chronological sequence with the respective IRSL dating results in stratigraphic order constraining the earthquake occurrence times. IRSL dating results are initially represented as uniform distributions covering the respective errors. In the following step, the chronological sequences of all trenches are combined by the "Phase" command, so that the constraints from all trenches for each single earthquake occurrence time are considered at the same time. Intervals between earthquakes are calculated using the command "Difference". Detailed code information is given in the supplementary material.

## 4 Trenching results

We excavated two trenches along the geomorphic fault scarp between the villages of Markgrafneusiedl and Gaenserndorf (Figures 3A and 4). The third trench consists of a construction trench of a gas pipeline exposing the northern tip of MF and providing additional, but limited, information. In general, all outcrops show similar characteristics: at all trenching locations, the MF is exposed as a narrow (1 - 2 m) fault zone consisting of one or two fault branches striking parallel to the regional strike of the fault scarp of the MF (dip direction/dip: ~120/75). The footwall cut by the MF comprises deposits typical of the Pleistocene Gaenserndorf terrace. The hanging wall of the trenches expose sequences of almost horizontally layered, fine-grained sediments of probably fluvial and lacustrine origin intercalated with colluvium deposited discontinuously during the last 160 ka. High-resolution photomosaics of trenches SDF1 and SDF3 are provided in the supplementary material.

## 4.1 Trenching at SDF1

The 40-m-long, 3-m-wide and up to 4 m deep trench SDF1 was located about 20 km from the city limits of Vienna (Figures 3A and 4A). It was excavated in a small dry valley at the central part of the NE-SW trending geomorphological fault scarp (Figure 4C). The trench SDF1 exposed about 30 m of Gaenserndorf terrace deposits in the footwall and ca. 10 m of mostly fine-graded sandy to silty sediments in the hanging wall, divided by the 1.5 m wide fault zone of the SE-dipping MF. The fault zone includes two parallel steeply dipping faults F1 and F2, with F2 reaching almost the present-day surface (see Figures 5 and 6A).

At the NW part in the footwall, alluvial deposits of the Gaenserndorf terrace are exposed, consisting of poorly sorted coarse gravels and boulders in a grain-supported sandy matrix (U2). Pebbles show consistent NW-dipping imbrication throughout the entire footwall section. The inferred dominantly SE-directed paleocurrents are comparable to the flow direction of the Recent Danube. A lense of medium light grey sand with mm-size white mica minerals (U1 in Figure 5), partly exposed at the trench bottom, is cut by fault F2. The uppermost 0.5 m of the terrace deposit directly below the recent soil horizon does not show horizontal consistency and is probably reworked and repositioned.

In the hanging wall SE of the fault zone, three types of sediments are exposed:

(A) Sequences of horizontal layers of light-grey and light-brown silt and fine sand with varying thicknesses up to 20 cm (Units U3, U4, U7). Sediments show lamination on cm-scale and intercalations of cm-thick horizons of coarse sand. The layers also include singular well-rounded pebbles and granules aligned in horizontal layers. Some sand/silt layers show fining-upward trends. Pedogenetic carbonate cementation is observed along the top of the uppermost silt layer and along recent root paths. The sediments are intercalated with and onlap on the wedge-shaped colluvial deposits described below. We relate the deposits to high-stage floods in the floodplain of the Pleistocene Danube.

(B) Colluvial wedge deposits (W2-W5 in Figure 5) and associated tension crack fills (T2-T4 in Figure 5). These colluvial sediments are adjacent to both faults and decrease in thickness towards the SE (i.e., away from the fault scarp, Figure 6B). The steep contact with the SE-dipping faults and the thinning of the deposits towards SE results in a wedge-shape of the sediment

layers. The tails of wedges W2, W3 and W5 can be followed throughout the exposed part of the hanging wall. All wedges are associated with tension cracks adjacent to the faults (T2-T4), which are filled with the same material as the overlying wedge, but showing no preferred orientation. The lowest wedge W5 consists of matrix-supported reddish brown medium gravel with a matrix composed mainly by sand and silt together with a low content of clay. The overlying wedge comprises two parts, W4 and W4a, delimited by steep irregular boundary adjacent high-stage flood sediments. While W4 consists of brown to reddish brown fine to medium sand with some fine granules and pebbles in a matrix-supported fabric, W4a include rounded pebble-size clasts of the reddish wedge material interpreted as mud balls (Figure 6C and E). We interpret this peculiar contact to result from the partial erosion of the wedge and the wedge tail during high-stage floods and the re-deposition of the colluvial material by fluvial processes or small slumps. Wedge W3 consists of well-sorted reddish-brown medium-grained sand with a few pebbles (fine gravel) showing lamination dipping away from fault F1 (Figure 6C). The red and reddish-brown colour of W3-W5 contrasts from the light grey-beige intercalated high-stage flood deposits. The sedimentary wedge material was identified as redeposited soil, because its colour resembles ferretto soils (5YR 4/4, Y5YR 5/4 and 5YR 58 of the standard soil colour chart; L. Smolíková, pers. comm.). The source for the redeposited soil would be the previous soil cover of the terrace gravels in the footwall of the MF. While W3, W4, and W5 are bounded by F1, wedge W2 is adjacent to F2 and overlies the trace of F1as well as the W3 deposits. W2 consists of large well-rounded pebbles and cobbles oriented sub-horizontally in a grain-supported fabric, similar to the terrace deposits found in the foot wall. On top of W2 and U2, a paleo-soil has been developed after deposition of W2, visible by black and brownish colouring of the upper 25 cm (Figures 5 and 6A, B). The underlying tension crack T2 is composed by the same material, but with slightly larger cobbles with no horizontal layering.

(C) Fine-grained alluvium (units U5, U6, U8) and loess (U9) found in both the hanging wall and the footwall consists of several thin layers of sand and fine gravel overlain by up to 1 m of unstructured silt and fine sand (U10). The latter is transitional to the overlying dark brown to black soil horizon (U11). The succession is interpreted as alluvium of the dry valley and loess or redeposited loess. Fault 2 offsets the alluvial sand layers (U5, U6, U8) for about 15 to 20 cm, but terminates within the overlying sediments of U10 several cm above the base of the layer (Figure 6A).

Structural data obtained from the two faults exposed in the outcrop show that both faults strike parallel to the regional strike of the fault scarp of the MF. The faults are marked by bands of rotated pebbles with preferred orientations parallel to the fault planes. Pebbles in the 75-cm thick fault zone between F1 and F2 show orientations, which geometrically resemble S-C-type fabrics. Deformation bands are found in the sandy deposits of W3 and W4, and the related tension cracks T3 and T4 (Figure 6c). Detailed mapping reveals that these microfaults do not penetrate into W5 most probably due to the higher clay content of these sediments. The deformation bands show orientations consistent with the main faults of the outcrop. At the lower parts, the deformation bands dip parallel to F1 (dip direction/dip 130/80). The upper parts of the deformation bands are rotated away from the fault resembling horsetail splays. Finally, small-scale normal faults with displacement in the order of several centimetres are observed within units U6-U8 (Figure 5 and 6D).

### 4.1.1 Evidences for seismic events observed within trench SDF1

Slip events identified in SDF1 will be labelled as A1-A5 from the youngest to the oldest. Offset of units U5, U6, and U8 at the tip of F2 provides direct evidence for the youngest surface-breaking slip event A1 (Figure 6A). The small-scale faults observed in the same units U6, U7, and U8 (Figure 6D) are at the same stratigraphic level and indicate faulting within the hanging wall during A1. Neither F2 nor the small faults cut units 9 and 10. In addition, W2 has been also offset by A1 along F2, clearly seen by the coloured paleo-soil (Figure 6A). The observed colluvial wedges W2-W5, their geometrical relation to the adjacent faults, and the tension cracks (T2-T4) filled with same material prove four distinct events (A2 to A5) of co-seismic displacement at the MF. In addition, the existence of deformation bands within the sandy deposits of W3, T3, W4, and T4 indicates further deformation of both wedges during younger slip event, either A1 or, more probably A2.

Due to the subhorizontal layering within the hanging wall, the occurrence of the events A2, A3, and A5 can be bracketed by sediments lying below and above the respective wedge tails. The deposits of W2 lie directly on top of W3 (Figures 5 and 6B). The associated earthquakes A2 and A3 must have occurred after the deposition of U4 and before U5 and U6 were deposited. A5 must have occurred between the deposition of units U3 and U4. The colluvial wedge associated with A4 has been partly eroded (W4 and W4a), which must have happened during the deposition of U4. Therefore, A4 must have happened after the occurrence of A5 and before the deposition of U4.

Direct measurement of offset is only possible for A1, which is constrained by the dip-slip offset of layers correlated across fault F2, i.e., units U5 and U6 and the top of W2. The 1.5 m of apparent vertical offset of U5 and U6 away from the fault between the top of the footwall and SE end of the hanging wall are due to the sedimentation on pre-existing topography. Following the generally accepted rule of thumbs that colluvial wedge height is approximately half of the surface displacement of an earthquake (McCalpin, 2008), the measured maximum thickness of each colluvial wedge W2-W5 can be used to estimate the minimum displacement for the associated event (Table 2).

### 4.1.2 Age control at SDF1

The fine-graded sandy and silty subhorizontal deposits of the hanging wall of trench SDF1 provide good dating material for IRSL dating. We sampled the sandy sediments in stratigraphic positions below and above the tails of the colluvial wedges A5, and A2/A3. In addition, we took samples from unit 7 which is affected by faulting related to A1, and from unit U10 sealing fault F2. In general, luminescence dating results fit well to the stratigraphic hanging wall sedimentary sequences observed in the trench, showing continuous decrease in age from the bottom towards the top (Figure 5 and Table 1). In addition, the age derived for the Gaenserndorf terrace in the footwall fit well with other ages from this terrace (Weissl et al., 2017).

Event A1 is well constraint between the IRSL dating results derived for unit 10 (13.8 ± 1.4 ka) and unit 7 (16.3 ± 1.8 ka) by samples from an offset sand layer and overlying undeformed sediments. Events A2 and A3 are bracketed by the ages of unit 7 (16.1 ± 1.7 ka) and the uppermost sediments of unit U4 below W2/W3, which were dated to 48.9 ± 4.8 ka. The occurrence

times of A4 and A5 are constraint by the ages of the lowest sediments of U4 (56.6 ± 5.7 ka) and the age of deposits of U3 (104 ± 12 ka) below the W5, the stratigraphically lowest sediments in the hanging wall.

## 4.2 Trenching at SDF3

We opened the second trench SDF3 about 1.5 km SW of trench SDF1 (Figure 3A and 4A). This 33 m long, 3 m wide and up to 5 m deep trench was located along a clearing in the forest. The about 0.5 m wide fault zone of the SE-dipping MF divides the N-S trending trench into two parts. The footwall W of the fault mainly consists of gravels of the Gaenserndorf terrace whereas the hanging wall in the E shows a succession of fluvial sediments, colluvial deposits originating from the uplifted footwall and reworked silty sediments (Figure 7).

The footwall consists of poorly sorted, well rounded sandy gravels within a grain-supported fabric. The clasts include metamorphic rocks, gneisses, quartzite along with minor sandstone and limestone. The few magmatic components found within the gravels are completely weathered. The lower 1-1.5 m of the terrace exposed in the trench contains coarse cobbles with diameters up to 25 cm (U1). The upper part shows typical characteristics of braided river deposits, including crossbedding of good-sorted gravel layers intercalated with sand layers of up to 0.5 m of thickness and several meters of lateral extent (U2). Furthermore, this part consists of gravel and small cobbles with diameters up to 10 cm. All layers show a slight inclination towards the SE. Throughout the terraces deposits, vadose gravitational carbonate cementation ("dripstone cementation") along the lower side of larger gravels is observed.

The hanging wall consists of horizontally layered fluvial, probably alluvial, and erosional deposits. In the following, we describe the most important units of the hanging wall, starting with the lowermost unit. Unit 4 consists of intercalated beige to grey, medium to fine sand, and gravel layers comprising well-rounded, poorly sorted clasts. The thickness of the layers varies between ~3 cm and 20 cm, whereas the sand layers are generally thicker than the intercalated gravel layers. This sequence is the result of alternate high-stage Danube floods (sand layers) and erosional events transporting gravels from the footwall into the hanging wall. Colluvial wedge W5 is bounded by F3 towards the N, decreases in thickness towards the SE and disappears approximately 3.5 m away from the fault zone. W5 consists of matrix-supported conglomerate with a sandy matrix and poorly sorted, well-rounded clasts. On top of W5, separated by a sand layer of few centimetres, the layer W4 spreads through the whole hanging wall. The W4 deposits consist of matrix-supported conglomerate with clay-rich, Fe-rich red fine sandy matrix and poorly sorted, well-rounded clasts with diameters up to 15 cm. Grain size decreases with increasing distance from the fault, as well as the layer thickness from 70 cm directly at the fault to less than 50 cm further away (Figure 8A). This decrease in grain size together with the similarity of the W4 and U1 deposits suggest that this material has been eroded from the Gaenserndorf terrace. The contact to the overlying unit U5 is diffuse and irregular. U5 consists of red clay-rich Fe-rich fine sand with intercalated layers and up to 5 cm thick lenses of brownish slightly coarser sand without clay or Fe components. The layering shows a slight inclination of a few degrees towards the NW, i.e., towards the fault. The material is typical for distal flood basin deposits of fluvial environments. A few well-rounded clasts with diameters of 2 - 20 cm were observed that we interpret as dropstones (Figure 8B). Furthermore, an animal burrow of small animals, refilled with beige coarse sand is

observed at the bottom of this layer. The contact between this unit and the overlying unit U6 is characterized by a horizontal sharp contact, which has been affected by liquification, either caused by an earthquake or by deposition of the overlying coarse gravels over the still water-bearing sediments of U5 (Figure 8B, C). Close to the fault, there are several wedge-shaped deposits between U5 and U6. W3 is exposed for about 3 m in the hanging wall close to the fault. It consists of approximately 25 cm thick gravels and small cobbles with diameters up to 10 cm. On top of W3, a package of layered sediments dips towards S (W2). Its thickness close to F1 is about 0.8 m and decreases within 1 m away from the fault. The layers consist of well-sorted pebbles, partly without matrix (red layer). W2 is then covered with U6.  U6 consists of grain-supported conglomerate with well-rounded, poorly sorted cobbles with grain sizes up to 10 cm. Those gravels originate from the footwall and form a colluvial wedge, which decreases in thickness with increasing distance to the fault. Unit U7 consists of olive-coloured medium sand with rare mica components. This 10-cm thick unit decreases in thickness towards the fault. This fact, together with the colour of the sand, suggests that it is a flood deposits of the Danube. Unit U8 covers both, the hanging and the foot wall, and consists of a matrix-supported conglomerate with silt matrix and around 25% of components comprising poorly sorted, well-rounded pebbles with grain sizes up to 3 cm. The silt matrix consists of reworked loess that has probably eroded from the footwall, including smaller clasts from the Gaenserndorf terrace. In the top of this unit, secondary carbonate cemented a horizontal layer of up to 30 cm thickness. The layer is observed throughout the entire hanging wall. Carbonate cementation occurs due to meteoric waters dissolving carbonate from the upper layers and precipitating it at lower pH values in greater depth. Conjugated planar carbonate fissures of up to 60 cm length branch off from the cemented layer. They strike approximately parallel to the orientation of the MF. Unit U9 is the AC soil horizon, consisting of a matrix-supported conglomerate of fine sand and 30-40% of components containing partly angular and rounded pebbles with grain sizes up to 2 cm.  The contact to both, the underlying and overlying units is rather diffuse. Finally, unit U10 is the A soil horizon that increases in thickness with increasing distance from the MF. Its thickness coincides with a layer of silt or loess that has been reworked as soil.

Structural data obtained from the outcrop show that both faults strike parallel to the regional strike of the fault scarp of the MF (dip direction/dip: 116/74). The MF is marked by the contact between the footwall gravels and the sandier deposits of the hanging wall. In addition, at the lower 1.5 m, clasts within a zone of about 50 cm to the fault are rotated parallel to the fault (116/69). The upper part of the MF is marked by a small band of rotated clasts (F1). In this upper part of the fault, layers that can be correlated on both sides of the fault are displaced by about 15 cm, indicating movement along F1. In addition to the main fault strands F1-F3, several conjugated sets of normal faults are observed within lower units of the hanging wall. These faults are consistently oriented parallel to the MF. The NW-dipping antithetic faults (303/79) are generally longer than the SE-dipping faults (137/72). Displacement observed along the faults is in the range of about 10 cm in U4, and up to 1 cm in U5. Within this layer, the small faults are recognised as a few mm thin deformation bands, most probably filled with carbonate cement, and show almost no displacement. None of the small faults seem to penetrate into the gravels of U6. The sand layers of U4, consisting of intercalated layers of sand and matrix-supported gravels, comprises small deformation bands with lengths up to 20 cm. They are arranged parallel to the small faults and accordingly dip towards the SE or the NW.

## 4.2.1 Evidences for seismic events observed within trench SDF3

Single slip events identified in SDF3 are labelled as B1-B5. The MF within the trench SDF3 is a narrow fault zone of 0.5 m width at its lowest point excavated within the trench, and reduces to a fault represented only by a few rotated clasts in the uppermost part (F1). This reduction of thickness occurs in distinct steps which can be related to different earthquakes. The oldest earthquake B5 is identified within the trench created a colluvial wedge W5 along the fault trace F3. This fault trace is then covered by deposits of W4. Despite its uncommon shape (continuously observed within the trench, non-wedge-shaped deposit), we observe a decrease in grain size away from the fault zone. In addition, there are no rivers parallel to the fault scarp that could provide such coarse material at this location. The gullies visible in Figure 4A are too far away to transport this material to the trench location. In addition, the deposits clearly cover the fault trace of F3. Therefore, we interpret W4 as a colluvial material that has been deposited after an earthquake B4 associated with movement along F2. W5 and W4 are bracketed between the hanging wall units U4 and U5. Evidence for event B3 is a tension crack T3 between F2 and F2', that is also identified by the thin sand layer that is smeared into the crack, parallel to F2'. W3 is most probably deposited before the occurrence of B3, and the material was smeared into the fault zone along F2' during B2. As the deposits of W3 partly cover the deposits of U5, the occurrence of B3 must be later than U5. The primary evidence for B2 is the ~ 0.8 m thick colluvial wedge W2 (Figures 7, 8A). However, this colluvial wedge shows inclined layers that continues from horizontal layers within the terrace. This layering is not typical for a colluvial wedge, but may be explained by the following scenario: In the case that coseismic surface rupture offsets sediments under periglacial conditions, seasonal thawing and freezing may cause geli-solifluction with material gliding down to the hangingwall and destroying a free surface previously formed by B2 (Vanneste et al., 2001). The described situation allows for two different interpretations of the surface displacement of B2. Interpreting the wedge-shaped deposit as a classical colluvial wedge adjacent to a fault plane which is not readily seen due to unfavourable outcrop conditions, a minimum displacement can be estimated by multiplying the maximum wedge height by two (McCalpin, 2008), which suggests displacement of 2 x 0.8 m = 1.6 m for B2. For the case of wedge formation by gelifluction, the coseismic surface displacement would be approximately the same as the colluvial wedge height, i.e. 0.80 m. We consider the latter scenario (referred to as the "periglacial wegde" in the following) as the more fitting one and therefore use 0.80 m as the preferred value for the surface displacement for B2. Since B2 is directly on top of unit W3, and is covered by the gravels of U6, its occurrence can be bracketed between the deposition of units U5 and U6, similar to B3. Insights for the youngest event B1 in the trench are the most obvious. Dip-slip displacement of ~ 10 cm along F1 affected all layers excluding only the soil horizons (U9 and U10), suggesting that even with such a small displacement of 10 cm, event B1 ruptured the surface (Figure 8A).

## 4.2.2 Age control at SDF3

The fine-graded sandy deposits of U3 (AIP93, 158 ± 21 ka), U4 (AIP95, 123 ± 16 ka), U5 (between 111 ± 12 ka, AIP95 and 70.8 ± 8.0 ka, AIP97), U7 (32.9 ± 4.1 ka, AIP98) and U10 (4.8 ± 0.5 ka, AIP102) in the hanging wall provide material suitable

for IRSL sampling. The dating results (details see Table 1) follow the chronological order of the sedimentary deposits well. In addition, two samples from sand lenses within unit U2 (AIP103 and AIP114) determine the minimum age of the footwall to $205 \pm 37$ - $259 \pm 35$ ka. Those obtained ages agree well with other IRSL ages for the Gaenserndorf terrace (Weissl et al., 2017). The IRSL data of the hanging wall constrain roughly the occurrence times of the five observed paleo-earthquakes along the main fault. B1 has affected U6 and the base of U8, and therefore also U7, constraining its occurrence after $32.9 \pm 4.1$ ka. As units U9 and U10 seal F1, B1 must have occurred before $4.8 \pm 0.5$ ka. As mentioned above, B2 and B3 are bracketed between units U7 and the top of U5, limiting its occurrence time between $32.9 \pm 4.1$ ka and $70.8 \pm 8.0$ ka. B4 and B5 are also jointly constraint between the deposition of unit U4 ($111 \pm 12$ ka) and the base of U5 ($123 \pm 16$ ka)

### 4.3 Outcrop WAG

Additional evidence for active faulting at the MF is available from the construction pit of a gas pipeline, which crosses the northern part of the fault scarp close to the city of Gaenserndorf, 6 km north of trench SDF1 (Figure 3A). The outcrop revealed a 1-m-wide localized fault zone (Figures 9, 10A). The fault cuts light-grey gravel and sand of the Gaenserndorf Terrace and overlying approximately 1 m thick fine-graded silty to sandy deposits constituting its footwall. The exposed hanging wall succession includes poorly sorted sandy gravels, which is then overlaid by a banded sequence of silty sediments. This cover layer can be found all along the pipeline construction pit and has been described in detail by Weissl et al. (2017). Two IRSL samples from fine-graded silty to sandy cover were dated to ages of $15.1 \pm 1.5$ ka and $16.1 \pm 1.7$ ka, respectively (samples AIP25, 26, Table 1), giving the only time constrain at the WAG site. Both the hanging wall and footwall are overlain by c. 30-50 cm thick brown soil, which has been removed prior to excavation. The exposed fault zone consists of several faults within the terrace gravel marked by aligned and fractured pebbles, faults offsetting sand layers, and faults offsetting the contact between gravel units and the overlying cover silts (Figure 9 and 10B, C). Several sheared pebbles indicate dip-slip movement along the faults (Figure 10D, E). The displacements of these faults are between 10 and 20 cm. On the SW wall, a fault cuts up through the entire silty section to the base of the overlying soil, offsetting a thin white layer within the upper part of the cover silty sediments by 20 cm (Figure 10A). This dip-slip offset indicates evidence for the youngest earthquake at the WAG site, C1. This earthquake must have happened after the deposition of the silty cover sediments, i.e. after $15.1 \pm 1.5$ ka. In Figure 10A, the poorly sorted sandy gravels in the hanging wall form a wedge-shaped deposit close to the fault (marked with CW). This can be interpreted as a colluvial wedge associated with an earthquake C2 that occurred before the deposition of the silty cover sediments. Therefore, C2 must be older than $16.1 \pm 1.7$ ka. Since the base of the colluvial wedge is not identifiable, a displacement of C2 cannot be assessed.

### 5 Correlation of events between sites

Palaeoseismological investigations along the MF include three locations, the trenches SDF1 and SDF3 as well as the pipeline outcrop WAG. For all three locations, detailed mapping and dating have been carried out and described above. Evidence for

five possible earthquakes have been observed in the trenches SDF1 (named A1-A5) and SDF3 (B1-B5), while in the pipeline trench WAG, observations indicate two paleo-earthquakes (C1 and C2). Table 2 summarizes the observations, time constraints and occurrence times for each earthquake and each trench separately. The central panel of Figure 11 shows the constraints of earthquake occurrence times for each trench separately. Based on this information, together with comparison of trench observations and displacement estimates for all three outcrops, we correlate trench observations and occurrence timing results to generate a synthesis of earthquake occurrence along the MF. In the following, we discuss each earthquake and the possible correlations between the trenches as well as the resultant occurrence time, displacement and magnitude estimate, starting with the youngest. The summary of this discussion is shown in Table 3.

### 5.1 Event E1 (A1 = B1 = C1)

In all three excavations, the youngest event is evident from a measurable offset of layers across the MF. At trench site SDF1, the youngest event A1 shows dip-slip displacements of 15-25 cm (Figure 6A). Its occurrence time is bracketed between 13.8 ± 1.4 ka and 16.3 ± 1.8 ka. At trench site SDF3, markers have been displaced by the youngest event B1 by 10-15 cm along dip (Figure 8A). ISRL dating results limits the occurrence time of B1 to the time range between 4.8 ± 0.5 ka and 32.9 ± 4.1 ka. In the pipeline trench WAG, the loess cover is dated between 15.1 ± 1.5 ka and 16.1 ± 1.7 ka. It is displaced along dip by 17-20 cm, visible by a white marker within the loess (Figure F10A). Therefore, C1 must have happened after 15.1 ± 1.5 ka. Combination of constraints for E1 from all three sites yields to an occurrence time of 14.2 ± 0.8 ka. Using the empirical relationship between surface displacement and magnitude (Wells and Coppersmith, 1994) for the maximum displacement of 25 cm, the magnitude estimate for E1 is $M(d_{max}) = 6.2 ± 0.5$. The average displacement of 17 cm would lead to a similar magnitude estimate of $M(d_{ave}) = 6.3 ± 0.6$.

### 5.2 Event E2 (A2 = B2 = C2)

Event E2 is also observed at all three sites as a triangular-shaped colluvial wedge mainly consisting of reworked gravels derived from the terrace in the footwall (Figures 6B, 8A, 9). In addition, the colluvial wedges deposits related to A2 and B2 are displaced by the younger event E1, confirming the correlation of the colluvial wedges to the penultimate seismic event E2. At trench site SDF1, the displacement related to A2 (1.5-1.9 m) is estimated from the height of the associated colluvial wedge (0.75 - 0.95 m). IRSL samples from sediments above and below the colluvial wedge constrain the occurrence time for A2 between 16.1 ± 1.7 ka and 48.9 ± 4.8 ka. At trench site SDF3, the interpretation of deposits related to B2 are more ambiguous (see sec. 3.2), and therefore, the estimated displacement is either 0.8 m (periglacial wedge scenario) or 1.6 m (colluvial wedge scenario). B2 is constrained between 32.9 ± 4.1 ka and 70.8 ± 8.0 ka by the IRSL ages of sediments above and below the colluvial wedge material. In the pipeline construction trench WAG, the displacement of E2 is not constrained, because the base of the wedge was not excavated. The maximum age of 16.1 ± 1.7 ka for the loess covering the colluvial wedge gives a minimum age for the occurrence of C1.

Combining the time constraints from all trench sites allow to determine the occurrence time of E2 to 37.1 ± 4.9 ka. Magnitude calculation using the maximum of the observed surface displacements results in a magnitude of $M(d_{max}) = 6.8 ± 0.4$ (Wells and Coppersmith, 1994). With the maximum value for the observed surface displacement coming from trench SDF1, the magnitude estimate does not depend on the interpretation for the B2 deposits in trench SDF3.

**5.3 Event E3 (A3, probably correlated with B3)**

For this event, a correlation based on field observations between the trenches SDF1 and SDF3 is not as clear as in the cases of E1 and E2, especially since the evidence for B3 does not allow to determine a displacement for this event. However, the maximum height of a well-developed sandy colluvial wedge in SDF1 gives a good estimate of an earthquake with $M(d_{max}) = 6.6 ± 0.4$ (Figure 6C). Because of the similar stratigraphic constraints, the combined possible occurrence time of E3 at 43.4 ± 4.9 ka overlaps with that one of E2 (Figure 11, lower and upper panels).

**5.4 Event E4a (A4, if correlated with B3)**

Due to the loose time constraint of B3, another possible correlation scenario between the trench sites SDF1 and SFD3 is the correlation of events A4 and B3 (slip model 1 in Figure 11, upper panel). If A4 and B3 are correlated to the same seismic event E4a, the overlap of possible occurrence times of A4 and B3 narrows the resultant occurrence time for E4a to 64.9 ± 5.6 ka. Observations of the maximum wedge height at trench site SDF1 indicate the magnitude of A4 (and therefore for E4a) to $M = 6.8 ± 0.5$.

**5.5 Event E4b (A4, if correlated with B4)**

In an alternative scenario, A4 could also correlate to B4 (slip model 2 in Figure 11, lower panel). In this case, the combined occurrence time for the resultant seismic event E4b must be older than the 111 ± 12 ka old sediments below the wedge associated with B4 in trench SDF3 (Figure 7) and younger than the lowest sediments in trench SDF1 dated to 104 ± 12 ka (Figure 5). Thus, the time constraints for E4b would lead to a narrow time window for its occurrence of 106 ± 3 ka. Similar to E4a, the magnitude for E4b can be estimated from the observations of the maximum wedge height of A4 at trench site SDF1, indicating a magnitude for E4b of $M = 6.8 ± 0.5$.

**5.6 Event E5a (possible correlation between A5 and B4) and Event 5b (possible correlation between A5 and B5)**

The further back in time, the more uncertain the correlation between both trench sites becomes. So, whether A5 and B5 are correlated to one event E5a, or A5 and B4 are correlated to one event E5b is not clearly determined neither by observations nor dating. Both alternatives are possible and only depend on whether A4 is correlated to B3 or to B4. In slip model 1 (Figure 11, upper panel), where A4 is correlated with B3, it appears reasonable to assume subsequently that A5 equals to B4. In contrast, if A4 is correlated with B4 (slip model 2, Figure 11, lower panel), the remaining correlation for the next older event would be that A5 corresponds to B5. Due to the loose time constraints in the lower part of all trenches, the available chrono-

stratigraphic constraints for E5a and E5b are identical to those used for E4b (Figure 11, central panel). However, additional constraints of chronological order demand that E5a and E5b must have happened earlier than E4a and E4b, respectively. This yields to slightly different occurrence times of $107 \pm 4$ ka and $109 \pm 3$ ka for E5a and E5b, respectively. Since both magnitude estimates for E5a and E5b are based on the maximum colluvial wedge height of A5 from trench SDF1, the magnitude of E5a and E5b is $M = 6.5 \pm 0.4$.

### 5.7 Event E6a (B5, if not correlated with any event in SDF1)

For the case that B5 is not correlated with any events recorded in trench SDF1 (slip model 1), the timing of E6a would be bracketed by the IRSL dating results of $111 \pm 12$ ka and $123 \pm 16$ ka. In addition, with A5 (=E5a) being the oldest earthquake evidence in SDF1, E6a might be older than the oldest deposits in SDF1 and therefore not visible there. Both time constraints yield to an occurrence time of $120 \pm 7$ ka for E6a. Unfortunately, the magnitude of this possible seismic event cannot be constrained by trench observations.

## 6 Seismotectonic implications

### 6.1 Completeness and preservation potential of earthquake records along the MF

The comparison between the evidences for youngest surface breaking earthquake E1, which did not lead to the formation of colluvial wedges, and the older events, which are characterized by larger offsets and colluvial wedges throughout, raises the question on the completeness of the paleoseismological earthquake record. It seems that the colluvial wedges associated with larger earthquakes that rupture the same surface-breaking fault conceal or even erase evidence for smaller offsets by earthquakes such as E1. The displacement of the markers by E1 is only preserved because the event happened after the last earthquake (E2) that caused a colluvial wedge to form. Any future event with a surface displacement that is large enough to lead to the erosion of the offset markers in the footwall will destroy the evidence for E1. We conclude that this restriction also applies to earthquakes with small surface offset that occurred prior to E2. Therefore, earthquake records for magnitudes less than about 6.5 are most probably incomplete, and thus excluded from the recurrence calculation.

### 6.2 Recurrence intervals for earthquakes with magnitudes larger than 6.5 along the MF

The possible correlations of paleoearthquakes between the trenches allow for two different interpretations to assess the recurrence intervals of earthquakes with magnitudes larger than $M = 6.5$. The event E1 will be excluded during the following considerations, since the related magnitude estimate is lower than those obtained for E2-E5a/b.

As mentioned in sect. 5, the crucial part for the reconstruction of recurrence intervals therefore is whether A4 is correlated either to B3 or to B4 and, consequently, whether A5 is correlated to B4 or to B5, resulting in the following slip models:

Slip model 1: E2-E3(not correlated to B3)-E4a-E5a-E6a (5 earthquakes);

Slip model 2: E2-E3(correlated to B3)-E4b-E5b (4 earthquakes).

Recurrence intervals are calculated using OxCal v.4.3.2 (Bronk Ramsey, 1995, 2001) and the results are shown in Figure 12. Despite the large uncertainties, the plots show that the slip models may represent different types of earthquake distributions. Slip model 1 (E2-E3-E4a-E5a-E6a) shows a quasi-periodic recurrence of earthquakes with magnitudes larger than M = 6.5 with an average recurrence interval of 21 ± 15 ka (mean ± 1σ). The most likely elapsed time between E4a and E5a (42 ± 7 ka) is significantly higher than the average, and that one between E2 and E3 (6.2 ± 4.6 ka) is significantly lower than the average (see Figure 12, upper panel). The probability distribution for the interval between E3 and E2 peaks at 0, because the occurrence times of E3 and E2 are limited by the same stratigraphic constraints. Because of the large variation of recurrence intervals for slip model 1, an estimate about the expected next M≥6.5 earthquake along the MF is difficult to make. However, considering the elapse time of 37 ± 5 ka since the last M≥6.5 earthquake, E2, is above the average, the likelihood of an earthquake of this magnitude along the MF should be considered high.

For slip model 2 (E2-E3-E4b-E5b), the average recurrence interval of 24 ± 34 ka is predictably similar to that of slip model 1, but with a larger standard deviation. However, the distribution of earthquake recurrence intervals in slip model 2 differs significantly from slip model 1 by showing a bimodality (Figure 12, lower panel). As mentioned above, the probability distribution for the interval E2/E3 peaks at 0, and so does the interval for E4b/E5b. The most likely interval between E3/E4b (63 ± 6 ka) is 10-20 times larger than those for E4b/E5b (3.2 ± 2.6 ka) and E2/E3 (6.2 ± 4.7 ka). Inferring a clustered behaviour for earthquakes along the MF for slip model 2, the average cluster length would be 4.7 ± 3.8 ka. The elapsed time interval between clusters E2/E3 and E4b/E5b corresponds to the interval E3/E4b. In addition, the time since the occurrence of E2 until today can be also considered as a minimum time between clusters (37 ± 5 ka). Furthermore, another estimation of the minimum time between clusters can be estimated from the oldest layers in trench SDF3 (unit 8) dated to 158 ± 21 ka (sect. 3.2). Since there is no older record than B5, it is reasonable to assume that there was no earthquake during the time between B5 and the oldest unit 8 exposed in SDF3. Therefore, the minimum time elapsed between B5 (=E5b) and any older cluster must be at least 49 ± 13 ka. The average recurrence interval between clusters would be therefore at least 49 ± 13 ka. Adopting slip model 2 and assuming that M≥6.5 earthquakes cluster along the MF may lead to the presumption that the MF is in between clusters at the moment. Because the time since the last cluster (37 ± 5 ka) is below the average recurrence interval for clusters, the likelihood of a M≥6.5 earthquake along the MF might be considered low at the moment, with the next cluster most likely to occur in approximately 10 ka.

## 6.3 Comparison of long-term Quaternary slip rates with paleoseismological slip rates

Paleoseismological vertical slip rates are calculated based on trench observations. In trench SDF1, the total sum of displacement is 4.85 ± 0.50 m, using the minimum displacement obtained mostly from colluvial wedge heights (Table 2). As the oldest sediments in trench SDF1 are 104 ± 12 ka old, the resultant vertical slip rate is 0.05 ± 0.01 mm/a. For trench SDF3, this straight forward calculation is not possible, because there is no displacement estimate for the older three earthquakes. However, since the terrace top is not seen in the hanging wall, the difference in elevation of 3.5 m between the top of the terrace in the footwall and the bottom of the trench, can be used as a minimum displacement. Together with the IRSL dating

results for the oldest sediments in the hanging wall 158 ± 21ka, the resultant minimum vertical slip rate for SDF3 is 0.02 mm/a. The slip rate for the last two earthquakes B1 and B2, where displacement estimates are available, the vertical slip rate would be between 0.02-0.05 mm/a for the time since E2 (37 ± 5 ka) using either 0.9 m (E1 + periglacial wegde) or 1.7 m (E1 + colluvial wedge).

Long-term Quaternary slip rates of the MF can be estimated from the offset of two marker horizons, the top of the Gaenserndorf terrace and the base of the Quaternary deposits. The top of the Gaenserndorf terrace forms the current land surface in the footwall of the MF. As the top of the Gaenserndorf terrace is not exposed in the hanging wall sections of the trenches, and thus is buried below the present-day surface at the hanging wall, the height of the morphological scarp of about 17 m (Figure 3B) represents minimum displacement accumulated since the abandonment of the fluvial terrace 200 ka ago (Weissl et al., 2017).

The resulting minimum Quaternary slip rate is 0.09 mm/a.

The base of the Quaternary gravels in the hanging wall of the MF, which is equivalent to the top of Neogene sediments, is offset by approximately 40 m with respect to the base of the Gaenserndorf terrace in the footwall of the MF (Figure 3B). In the absence of age information for the gravels in the hanging wall of the MF and the bottom of the Gaenserndorf terrace, the oldest available age for the Gaenserndorf terrace, 322 ± 41 ka (Weissl et al., 2017), is used for estimating a maximum long-

term slip rate for the MF of 0.12 mm/a. A minimum slip rate of 0.02 mm/a is calculated from the 40 m displacement of the Quaternary gravels which must have accumulated after the Neogene-Quaternary boundary (2.6 Ma). Figure 13 shows the range of possible slip rates for both slip models falling in between the bracket of the geomorphic slip rates, which are in reasonable agreement.

**6.4 Comparison of magnitude estimates with fault rupture area and length**

For faults with known fault geometry, empirical relations allow to evaluate the maximum magnitude that a fault can produce from rupture length and area. The surface expression of the MF is only recognizable for about 10 km along the eastern margin of the Pleistocene Gaenserdorf terrace (Figure 3A). Further to the south, the River Danube has erased any geomorphic expression in its Holocene flood plain. However, the geometry and the length of the MF are well known from the distribution of Quaternary sediments in the hanging wall of the fault and 2D/3D reflection seismic (Hölzel et al., 2010, Hinsch and Decker,

2011, Salcher et al., 2012, Spahic et al., 2013). These data constrain the length of the MF to 25 km (Salcher et al., 2012). In addition, Hinsch and Decker (2011) proposed a generalized detachment for the Vienna Basin (see Figure 2). Beneath the MF, the detachment is assumed to be at the depth of about 10 km (Wessely et al., 2006). Taking into account the general dip of 55° for the MF observed in seismic lines, the rupture area of the MF above the detachment would amount to 315 km², leading to a maximal credible magnitude of 6.5 ± 0.3.

However, in case that the MF is indeed linked to the VBTF via the common detachment as proposed by Beidinger and Decker (2011), the area of the detachment between the MF and VBTF might be also activated during large events (Figure 14). The total fault surface activated during such events is derived as the sum of the fault surface of the MF and the portion of the basal detachment between the MF and the VBTF, which has a size of about 130 km². The fault length of the MF in this tectonic

scenario is 36 km and the total fault area amounts to about 580 km². These fault parameters correspond to a maximal credible magnitude of 6.7 ± 0.3 for earthquakes along the MF using the relationships of Wells and Coppersmith (1994), which is in good agreement with the paleoseismological magnitude estimations derived from our trenches.

## 6.5 Mmax for the Vienna Basin

Previous estimates of the maximum earthquake (Mmax) for the Vienna Basin are based on the evaluation of historical and instrumental seismicity only and present values for Mmax between M = 6.0 to 6.5 (Lenhardt et al., 1995; Procházková and Šimunek, 1998; Sefara et al., 1998; Tóth et al., 2006). In contrast to these assessments, the results of our study suggest that earthquakes with magnitudes up to about M=7.0 are likely to occur on longer time scales and with larger recurrence rates. Based on the data from the MF presented here, the upper limit of both scaling relationships, observed surface ruptures in trenches and the rupture area of the MF yield to a value of Mmax=7.0. Considering that the MF and the other splay normal faults are connected to the VBTF via the common detachment (Figures 2 and 14), a possible combined rupture along the MF and the VBTF would increase the Mmax for the Vienna Basin. Hinsch and Decker (2011) present different rupture scenarios along the VBTF with Mmax in the order of 5.9-6.8 for single-segment ruptures of different strike-slip segments of the VBTF. The assumption of scenarios such as earthquakes rupturing several strike-slip segments or events rupturing parts of the VBTF combined with secondary rupture along one of the splay normal faults would definitively increase the Mmax. Therefore, Mmax=7.0 should be considered in seismic hazard studies as a minimum for the Vienna Basin. However, there is not enough data so far to prove or disprove any of those scenarios, but recent earthquakes suggest that masked or blind faults may contribute to the development of multi-segment ruptures (e.g., Mw=7.2 El Mayor-Cucapah Earthquake 2010, Fletcher et al., 2014; Mw =7.8 Kaikoura EQ, Clark et al., 2017).

## 7 Conclusions and implications for seismic hazard assessment

In this study, we report constrains for the paleoseismological history of a splay normal fault of the VBTF that previously has not been considered as a source for seismic hazard calculations. We show evidence that the fault produced at least 5-6 earthquakes with magnitudes larger than 6.2 in the last 120 ka. The magnitude of the earthquake with the largest surface displacement derived from observed colluvial wedge thickness is evaluated with 6.8 ± 0.4. This value compares well with the maximum magnitude of 6.7 ± 0.3 estimated from the potential rupture area of the MF. The fault area is about 580 km² when including the detachment that links the normal fault with the VBTF. The vertical fault slip rate of 0.02 to 0.05 mm/a derived from trench observations agrees well with the geomorphologically determined vertical slip rates for the MF, which range from 0.09 to 0.02 mm/a.

Trench observations and uncertainties of IRSL age dating do not allow for an unequivocal conclusion of earthquake recurrence rates. Two earthquake scenarios, referred to as slip models 1 and 2, are possible considering the available time constraints and the possible correlations between identified paleoearthquakes in the three trenches (Figure 11). Among these, slip model 1,

appears less likely as it seems improbable that an earthquake with a magnitude around 6.6 has not been recorded in trench SDF3, while it produced a surface displacement of 80-90 cm in trench SDF1 at a distance of less than 2 km. For us, the more plausible correlation between the trenches is therefore slip model 2, suggesting that earthquakes with magnitudes larger than 6.5 may cluster in time. This may have consequences for the application of the reconstructed recurrence intervals in future seismic hazard assessments, e.g., by using cluster recurrence intervals rather than average single event recurrence intervals.

Trench evidence for the youngest event E1 (magnitude $6.2 \pm 0.5$) further suggests that earthquakes with magnitudes less than about 6.5 can also occur outside of the proposed clusters. Unfortunately, the recurrence intervals of such events cannot be constrained by trenching results. The sedimentary and structural records of events with surface displacements, which are too small to produce colluvial wedges, may be masked or even erased by subsequent larger earthquakes that lead to the erosion and redeposition of material into colluvial wedges. Therefore, earthquake records for magnitudes less than about 6.5 are most probably incomplete at the MF.

The issues discussed above lead us to conclude the following main implications for seismic hazard assessment in the Vienna Basin:

1. Although "silent" in historical times, the paleoseismological results demonstrate that the MF has been seismically active during the recent geological past with the youngest event occurring at $14.2 \pm 0.8$ ka. Therefore, the fault should be considered as a seismogenic source in seismic hazard assessment. Earthquakes with magnitudes larger than about 6.5 occur at average recurrence times of about 25 ka (slip model 1), or, more likely, may be clustered (slip model 2; Figures 11 and 12). The frequency of surface-breaking earthquakes with magnitudes less than about 6.5 along the MF cannot be constrained by the trenching results due to the low preservation potential of such earthquake records within the excavated trenches. The data presented in our study was used in the SHARE project to incorporate the MF in its active fault database (Basili et al., 2013).

2. Data from the MF provides evidence that a value of at least M=7.0 should be adopted for the maximum credible earthquake (Mmax) in the Vienna Basin. This value is significantly higher than previous estimates of Mmax = 6.0 to 6.5 (Lenhardt et al., 1995; Procházková and Šimunek, 1998; Sefara et al., 1998; Tóth et al., 2006). We stress that the value obtained from the MF only characterizes one out of a number of secondary normal faults to the VBTF. The VBTF may produce even larger earthquakes.

3. The MF is kinematically and geologically analogous to a number of other splay normal faults of the VBTF close to the Austrian capital, Vienna (Figures 1 and 14; Beidinger and Decker, 2011). It is prudent to assume that these faults are potential sources of large earthquakes as well. However, except for the Aderklaa-Bockfliess faults (Weissl et al., 2017), paleoseismic characterisation of these faults has yet to be conducted. The frequency of strong earthquakes near Vienna is therefore expected to be higher than the earthquake frequency reconstructed for the MF.

4. The magnitude of the largest earthquake recorded at the MF ($6.8 \pm 0.4$) is regarded to support the assumption of a listric fault with an active basal detachment that links the normal fault with the VBTF strike-slip system. The consequences of this geometry may be large in terms of ground motion patterns resulting from earthquakes that

activate large parts of the listric fault because of the hanging wall effect that amplifies the ground motion prediction (Passone and Mai, 2016). Although such directivity effects may reduce the hazard arising from the MF for Vienna, the opposite is true for other listric faults stretching into the city limits of Vienna (Figure 14).

## Acknowledgements

We wish to thank Dana Tschegg and all other colleagues who helped us to gain a deeper understanding of the paleoseismology of the MF by their discussion in the trenches, among them G. Bokelmann, M. Fiebig, C. Grösel, A. Landgraf, W. Lenhardt, P. Spacek, K. Reicherter, P. Štěpančíková, M. Strecker, and K. Vanneste. Thanks to E. Brückel, W. Chwatal, and K. Roch for their geophysical support and M. Meghraoui, who shared his morphological expertise in the site selection process. We further appreciate the hands-on support of many students of the Department of Geodynamics during the trenching action. Thanks also to P. Havlíček and L. Smolíková for analysing paleosoil from trench SDF1. Finally, we want to thank the reviewers D. Clark, R. Gold, and M. Ortuño, whose thoughtful and detailed reviews helped shape the manuscript into its present form. This research was funded by the Austrian Bundesministerium für Land- und Forstwirtschaft, Umwelt und Wasserwirtschaft, BMLFUW-UW-.1.1.4/00009-V/6/2009.

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

**Tables**

| Sample | Location | Depth (m) | Moisture (%) | 238U (ppm) | 232Th (ppm) | K (%) | De (Gy) | $D_0$ (Gy/ka) | Age (ka) |
|---|---|---|---|---|---|---|---|---|---|
| AIP25* | WAG cover | 1.0 | 12 ± 7 | 2.58 ± 0.04 | 8.42 ± 0.23 | 1.12 ± 0.02 | 44.7 ± 2.3 | 2.78 ± 0.26 | 16.1 ± 1.7 |
| AIP26* | WAG cover | 0.5 | 12 ± 7 | 2.35 ± 0.04 | 7.62 ± 0.21 | 1.01 ± 0.01 | 39.0 ± 1.4 | 2.01 ± 0.16 | 15.1 ± 1.5 |
| AIP38* | SDF1 fw | 3.7 | 10 ± 7 | 0.82 ± 0.02 | 3.54 ± 0.11 | 1.17 ± 0.02 | 543.1 ± 33.7 | 2.13 ± 0.20 | 255 ± 29 |
| AIP39* | SDF1 hw | 3.6 | 10 ± 7 | 1.78 ± 0.03 | 7.56 ± 0.21 | 1.18 ± 0.02 | 273.5 ± 20.2 | 2.64 ± 0.25 | 104 ± 12 |
| AIP40* | SDF1 hw | 3.3 | 10 ± 7 | 1.84 ± 0.03 | 7.64 ± 0.21 | 1.24 ± 0.02 | 154.0 ± 5.5 | 2.72 ± 0.25 | 56.6 ± 5.7 |
| AIP41* | SDF1 hw | 2.6 | 10 ± 7 | 2.69 ± 0.04 | 10.46 ± 0.28 | 1.61 ± 0.02 | 168.7 ± 5.9 | 3.45 ± 0.32 | 48.9 ± 4.8 |
| AIP44* | SDF1 hw | 1.9 | 10 ± 7 | 1.49 ± 0.03 | 5.92 ± 0.17 | 0.91 ± 0.01 | 36.7 ± 2.1 | 2.26 ± 0.22 | 16.3 ± 1.8 |
| AIP46* | SDF1 hw | 1.2 | 10 ± 7 | 2.39 ± 0.04 | 9.12 ± 0.25 | 1.44 ± 0.02 | 43.6 ± 1.6 | 2.01 ± 0.16 | 13.8 ± 1.4 |
| AIP93 | SDF3 hw | 4.1 | 12 ± 7 | 1.64 ± 0.03 | 8.06 ± 0.22 | 1.25 ± 0.02 | 419.2 ± 39.4 | 3.17 ± 0.30 | 158 ± 21 |
| AIP95 | SDF3 hw | 3.1 | 12 ± 7 | 1.87 ± 0.03 | 9.41 ± 0.26 | 0.97 ± 0.02 | 317.4 ± 29.1 | 2.58 ± 0.24 | 123 ± 16 |
| AIP96 | SDF3 hw | 2.6 | 12 ± 7 | 2.22 ± 0.04 | 8.96 ± 0.24 | 1.26 ± 0.02 | 292.0 ± 13.5 | 2.62 ± 0.25 | 111 ± 12 |
| AIP97 | SDF3 hw | 2.1 | 12 ± 7 | 2.22 ± 0.04 | 8.96 ± 0.24 | 1.26 ± 0.02 | 205.1 ± 13.3 | 2.90 ± 0.26 | 70.8 ± 8.0 |
| AIP98 | SDF3 hw | 1.8 | 12 ± 7 | 2.19 ± 0.04 | 8.28 ± 0.25 | 1.19 ± 0.02 | 91.6 ± 7.5 | 2.78 ± 0.25 | 32.9 ± 4.1 |
| AIP102 | SDF3 hw | 0.3 | 15 ± 7 | 3.01 ± 0.04 | 12.14 ± 0.33 | 1.30 ± 0.02 | 15.7 ± 0.9 | 3.29 ± 0.30 | 4.8 ± 0.5 |
| AIP103 | SDF3 fw | 1.4 | 10 ± 7 | 0.57 ± 0.01 | 1.79 ± 0.06 | 1.04 ± 0.02 | 384.6 ± 59.7 | 1.88 ± 0.18 | 205 ± 37 |
| AIP114 | SDF3 fw | 0.65 | 10 ± 7 | 0.57 ± 0.01 | 1.79 ± 0.06 | 1.04 ± 0.02 | 468.3 ± 43.4 | 1.81 ± 0.17 | 259 ± 35 |

**Table 1: Infrared stimulated Luminescence (IRSL) and optically stimulated luminescence (OSL) dating results from the trenches SDF1, SDF3, and WAG at the Markgrafneusiedl Fault (MF). * refer to samples published by Weissl et al. (2017). Location: refers to either of the trenches (SDF1, SDF3, WAG) and the location in respect to the MF, with hw = hanging wall and fw = footwall, N: number of used aliquots, De (Gy): equivalent dose in Gray (Gy), $D_0$ (Gy/ka): dose rate in Gray values (per 1.000 years); Depth (m): depth of the sampling location in meters below present-day surface.**

| Event # | Evidence | Thickness of colluvial wedge @ NE wall | Thickness of colluvial wedge @ SW wall | Displacement | Occurrence time |
|---|---|---|---|---|---|
| A1 | displ | - | - | 0.15 - 0.25 m | 15.1 ± 2.2 ka |
| A2 | cw, tc | 0.95 m | 0.75 m | 1.50 - 1.90 m | 27.5 ± 13.4 ka |
| A3 | cw, tc | 0.45 m | 0.40 m | 0.80 - 0.90 m | 38.4 ± 14.4 ka |
| A4 | cw, tc | 0.75 m | 0.72 m | 1.40 - 1.50 m | 74.0 ± 21.8 ka |
| A5 | cw | 0.25 m | 0.40 m | 0.50 - 0.80 m | 90.1 ± 22.9 ka |
| B1 | displ | - | - | 0.10 - 0.15 m | 18.3 ± 13.5 ka |
| B2 | cw | | 0.8 | 0.8/1.6 m | 44.9 ± 17.4 ka |
| B3 | tc | - | - | - | 57.7 ± 18.5 ka |
| B4 | cw | - | - | - | 115 ± 14 ka |
| B5 | cw | - | - | - | 123 ± 14 ka |
| C1 | displ | - | - | 0.17 - 0.20 m | <15.1 ± 1.5 ka |
| C2 | cw | - | - | - | >16.1 ± 1.7 ka |

**Table 2: Type of evidence, inferred displacement for the paleoearthquakes A1 to A5 (trench SDF1), B1 to B5 (SDF3), and C1 to C2 (WAG) and possible occurrence time. Colluvial wedge thickness observed on NE and SW trench walls used for estimating displacement. Displacement is taken as twice the colluvial wedge thickness. Evidence: dip-slip displacement of correlated layers (displ.), occurrence of colluvial wedges (cw), and sediment-filled tension cracks below the colluvial wedges (tc). Occurrence times (mean ± 2σ) are calculated with OxCal using chronological constraints from respective trenches.**

| Event # | Correlation | *Antequem*<br>Event older than<br>(age / sample location) | *Postquem*<br>Event younger than<br>(age / sample location) | *Magnitude*<br>M($d_{max}$)<br>*M($d_{ave}$)* | *Occurrence interval*<br>OxCal result<br>(mean ± 1σ) |
|---|---|---|---|---|---|
| E1 | A1 = B1 = C1 | **13.8 ± 1.4 ka (SDF1),**<br>4.8 ± 0.5 ka (SDF3) | 32.9 ± 4.1 ka (SDF3),<br>16.3 ± 1.8 ka (SDF1),<br>**15.1 ± 1.5 ka (WAG)** | 6.2 ± 0.5<br>*6.3 ± 0.6* | 14.2 ± 0.8 ka (SM1/2) |
| E2 | A2 = B2 = C2 | **32.9 ± 4.1 ka (SDF3),**<br>16.1 ± 1.7 ka (SDF1) | 70.8 ± 8.0 ka (SDF3),<br>**48.9 ± 4.8 ka (SDF1)** | 6.8 ± 0.4 | 37.1 ± 4.9 ka (SM1/2) |
| E3 | A3, ?= B3? | **32.9 ± 4.1 ka (SDF3),**<br>16.1 ± 1.7 ka (SDF1) | 70.8 ± 8.0 ka (SDF3),<br>**48.9 ± 4.8 ka (SDF1)** | 6.6 ± 0.4 | 43.4 ± 4.9 ka (SM1/2) |
| E4a | A4, ?= B3? | **56.6 ± 5.7 ka (SDF1)**<br>32.9 ± 4.1 ka (SDF3), | 104 ± 12 ka (SDF1),<br>**70.8 ± 8.0 ka (SDF3)** | 6.8 ± 0.5 | 64.9 ± 5.6 ka (SM1) |
| E4b | ?A4 =? B4 | **111 ± 12 ka (SDF3),**<br>56.6 ± 5.7 ka (SDF1) | 123 ± 16 ka (SDF3),<br>**104 ± 12 ka (SDF1)** | 6.8 ± 0.5 | 106 ± 3 ka (SM2) |
| E5a | A5 ?= B4? | **111 ± 12 ka (SDF3),**<br>56.6 ± 5.7 ka (SDF1) | 123 ± 16 ka (SDF3),<br>**104 ± 12 ka (SDF1)** | 6.5 ± 0.4 | 107 ± 4 ka (SM1) |
| E5b | ?A5 = B5? | **111 ± 12 ka (SDF3),**<br>56.6 ± 5.7 ka (SDF1) | 123 ± 16 ka (SDF3),<br>**104 ± 12 ka (SDF1)** | 6.5 ± 0.4 | 109 ± 3 ka (SM2) |
| E6a | B5 | **111 ± 12 ka (SDF3)** | **123 ± 16 ka (SDF3)** | - | 120 ± 7 ka (SM1) |

**Table 1: Overview of characteristics of each possible earthquake derived from all sites. IRSL constraints in bold mark the upper and lower limit for each occurrence time. Occurrence times are calculated with OxCal (see Figure 11) using stratigraphic constraints from all sites. SM1 and SM2 refer to slip model 1 and 2, respectively. For details about correlation between the trenches, see sect. 5.**

**Figure captions**

Figure 1: (A) Active faults (black solid and dashed lines), seismicity (black circles) and Quaternary basins (light grey areas) within the Vienna Basin (Austria) plotted on a shaded DEM (90 x 90 m resolution, SRTM data). Seismicity is based on the ACORN catalogue (2004). The borders of the Austrian capital, Vienna, is outlined by a dashed white line. Modified after Beidinger and Decker (2011). VBTF = Vienna Basin Transfer Fault. Important splay normal faults of the VBTF: ABF = Aderklaa-Bockfliess Fault, BNF = Bisamberg-Nussdorf Fault, LF = Leopoldsdorf Fault, MF = Markgrafneusiedl Fault. White box shows the location of Figure 3A, white line the position of the cross section of Figure 2; (B) Major earthquakes from historical, instrumental and paleoseismological data in intra-plate Central Europe. Historical and instrumental seismicity is based on the CENEC Catalogue by Grünthal et al., 2009. Paleoseismological study sites are compiled from Camelbeeck and Meghraoui, 1998; Camelbeeck et al., 2000; 2007; Meghraoui et al., 2001; Vanneste and Verbeeck, 2001; van den Berg et al., 2002, Peters, et al., 2005; Štěpančíková et al., 2010. Labels indicate the magnitudes of the largest paleoearthquakes observed at the respective site. Black box shows area of the close up in (A). VBFS = Vienna Basin Fault System.

Figure 2: Cross section through the Vienna Basin at its central part based on reflection seismic and deep boreholes indicating the common detachment of the Alpine floor thrust, which links the splay normal faults, such as the Markgrafneusiedl Fault (MF), to the Vienna Basin Transfer Fault (VBTF). The generalized detachment corresponds to the Alpine-Carpathian floor thrust, which is reactivated by normal faulting.

Figure 3: Overview of the Markgrafneusiedl Fault (MF). (A) DEM (10 x 10 m resolution, provided by the Government of Lower Austria) showing the Pleistocene terraces north of the River Danube dissected by faults creating fault scarps (fs). GDT = Gaenserndorf terrace, SHT = Schlosshof terrace. Dashed line: trace of the topographic profile in B, solid line: trace of the seismic line in C, white circle: locations of the villages Markgrafneusiedl (MGNS) and Gaenserndorf (GD). (B) Topographic profile (black) and cross-section indicating the base of Quaternary sediments (grey) across the MF. Note the thickness of Quaternary growth strata in the fault-delimited basin above the MF. Vertical exaggeration: 8.6x (C) seismic section across the same area showing offset along the MF and the flower structure at the Vienna Basin Transfer Fault (VBTF). Vertical exaggeration at 2s TWT: 4.5x. See text for details.

Figure 4: Trench locations of SDF1 and SDF3. (A) Locations of trenches SDF1 and SDF3 relative to the Markgrafneusiedl Fault (MF) and the margin of the Gaenserndorf terrace NW of the MF. Gullies normal to the MF are currently dry valleys resulting from Pleistocene drainage of the terrace. LiDAR image with 1x1m resolution provided by the Government of Lower Austria. (B) View of the fault scarp south of the forested area where the trenches are located, looking towards the N. The assumed fault trace of the MF is marked by the white dashed line. Approximate viewpoint is marked in (A) by white circle. (C) and (D) Topographic profiles across the fault scarp of the MF at trench locations SDF1 and SDF3, respectively. Vertical exaggeration: 16x. See profile locations in (A).

Figure 5: Trench log of the SW-facing wall of the trench SDF1 across the Markgrafneusiedl Fault (for location see Figures 3A and 4). Stratigraphic units are marked with U1-U11 from the oldest to the youngest unit, colluvial wedges and underlying tension cracks related to earthquakes A2-A5 are numbered as W2-W5 and T2-T5. Numbers indicate the age and the location of IRSL samples. See text for further explanation. See supplementary material for high-resolution photomosaic.

**Figure 6: Details from the SW-facing wall of the trench SDF1.** (A) Evidence for earthquake A1 from the displacement of a marker horizon (white arrows), which is correlated across the fault (red arrows). (b) Evidence for earthquake A2 from a colluvial wedge composed of sandy gravel overlying a tension gash filled with the same material. Top and base of the wedge are marked by white and yellow arrows, respectively. To the right the wedge abuts against fault 1 (red arrows). (C) Colluvial wedge associated with earthquake A3 overlying a tension gash adjacent to fault 2 (yellow and orange arrows: top and base of the wedge/tension gash; red arrows: fault). Several deformation bands that branch from fault 2 and formed during a later earthquake cut the wedge. It overlies wedge 4, which equally contains reddish redeposited soil. Wedge 4 shows an erosional contact to grey high-stage flood sediments (around box E). (D) Deformation bands offsetting laminated fluvial sand (red arrows) above wedge 2. The deformation bands are correlated to the event horizon of E1 (detail of picture B). (E) Detail of (C). Erosional contact of wedge 4 to flood sediments. Armoured mudballs (arrow) derive from the eroded colluvium.

**Figure 7: Trench log of the SW-facing wall of the trench SDF3 across the Markgrafneusiedl Fault** (for location see Figures 3A and 4). Colluvial wedges and underlying tension cracks related to earthquakes B2-B5 are numbered. The displacement related to B1 is marked. Numbers indicate the age and the location of IRSL samples. See text for further explanation. See supplementary material for high-resolution photomosaic.

**Figure 8: Details from the W-facing wall of the trench SDF3.** (A) Wedge associated with earthquake B2 (top: white arrows; base: yellow arrows; see text for discussion) overlying an upward widening fault (red arrows), recognized from pebbles, which are oriented parallel to the fault. The top of the wedge (white arrows) is offset by a narrow deformation band that emerges from the fault below the wedge (purple arrows). Offset occurred during B1. (B, C) Laminated flood sediments (clay, silt and fine sand) underlying colluvium of wedge B2. Pebbles sunken into the soft sediment (B) and flame structures protruding into the overlying gravel (C) are indicative for liquefaction.

**Figure 9:** (A) Trench WAG, photo mosaic of the W-facing trench wall. Red arrows denote locations of faults. Yellow arrows point to offset contact between colluvium and overlying loess. Orange arrows denote location of the faulted marker layer depicted in Figure 10. Boxes refer to details shown in Figure 10 B and C.

**Figure 10:** (A) Trench WAG looking NW toward the footwall of the MF (fault trace denoted by red arrows). Note offset of bright layer of loess (white arrows) corresponding to C1. CW denotes the colluvial wedge related to earthquake C2. (b) Detail of the SW-facing trench wall. Red arrows denote locations of faults, orange arrows point to the offset contact between grey and brown silt and clay. Box shows location of details shown in D. (C) Offset of the top of the colluvial wedge associated with earthquake C2 (yellow arrows). (D, E) Fractured and sheared pebbles indicating normal displacement parallel to the slip of the MF. Note that fractures in pebbles are filled with sandy matrix excluding fracture formation during construction work.

**Figure 11: Comparison of age constraints from all trench sites SDF1, SDF3, and WAG and possible occurrence times of the observed earthquakes for the two possible slip models.** The central panel shows the age constraints and earthquake occurrence times for each site separately. For the WAG site, only maximum constraint for C1 and minimum constraint for C2 are available. The upper and lower panels show the resultant earthquake occurrence times considering the combined chrono-stratigraphic constraints from all

sites together for slip models 1 and 2, respectively. For details about correlation see sect. 5. Calculations carried out using OxCal v4.3.2 (Bronk Ramsey, 1995, 2001).

Figure 12: Recurrence intervals for slip model 1 (upper panel) and slip model 2 (lower panel), respectively.

5    Figure 13: Comparison of surface slip rates for the Markgrafneusiedl Fault (MF) from geomorphic constraints and from trench results. On the left the constraints for slip model 1 are plotted, on the right, those for slip model 2 are shown.

Figure 14:  Geometry and fault area of the Markgrafneusiedl Fault (MF).  Also shown are the Vienna Basin Transfer Fault (VBTF) and other active normal splay faults branching from the VBTF. ABF: Aderklaa-Bockfliess fault system; BNF: Bisamberg-Nussdorf fault; LF: Leopoldsdorf fault; SF: Seyring fault (redrawn from Decker et al., 2015). Broken grey line marks the city limits of Vienna.

**Figures**

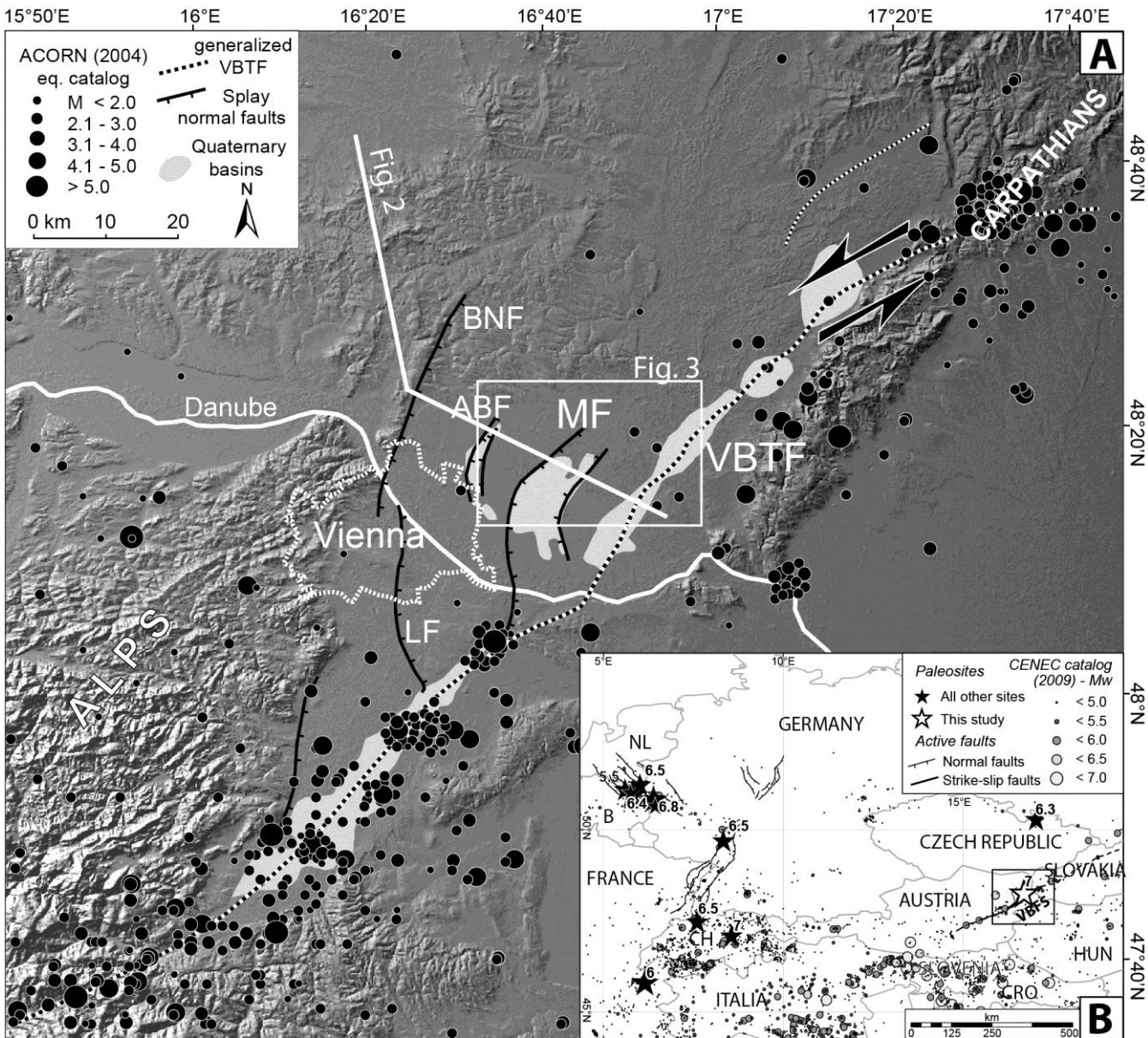

**Figure 1**

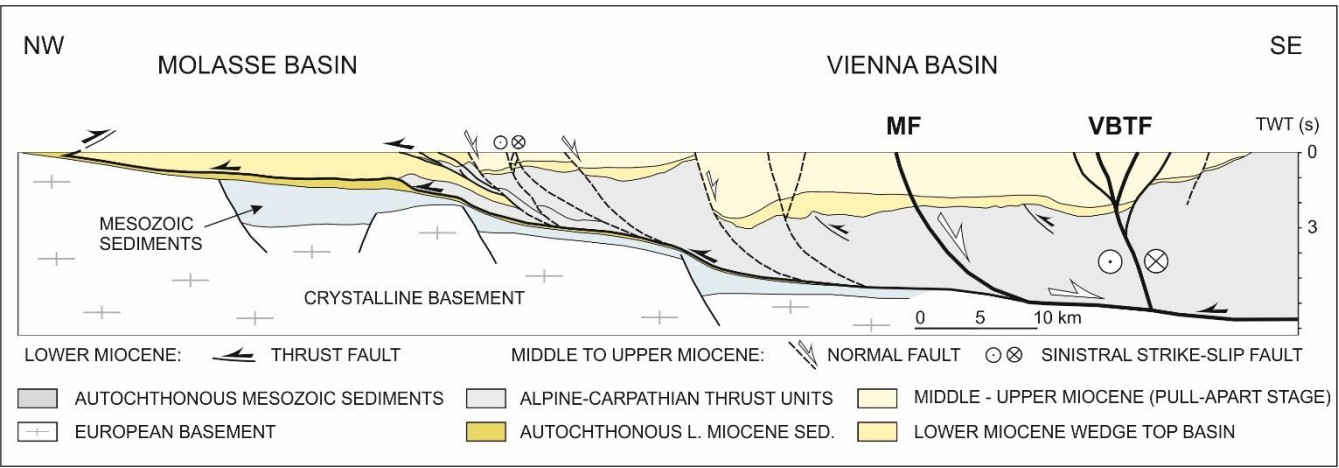

**Figure 2**

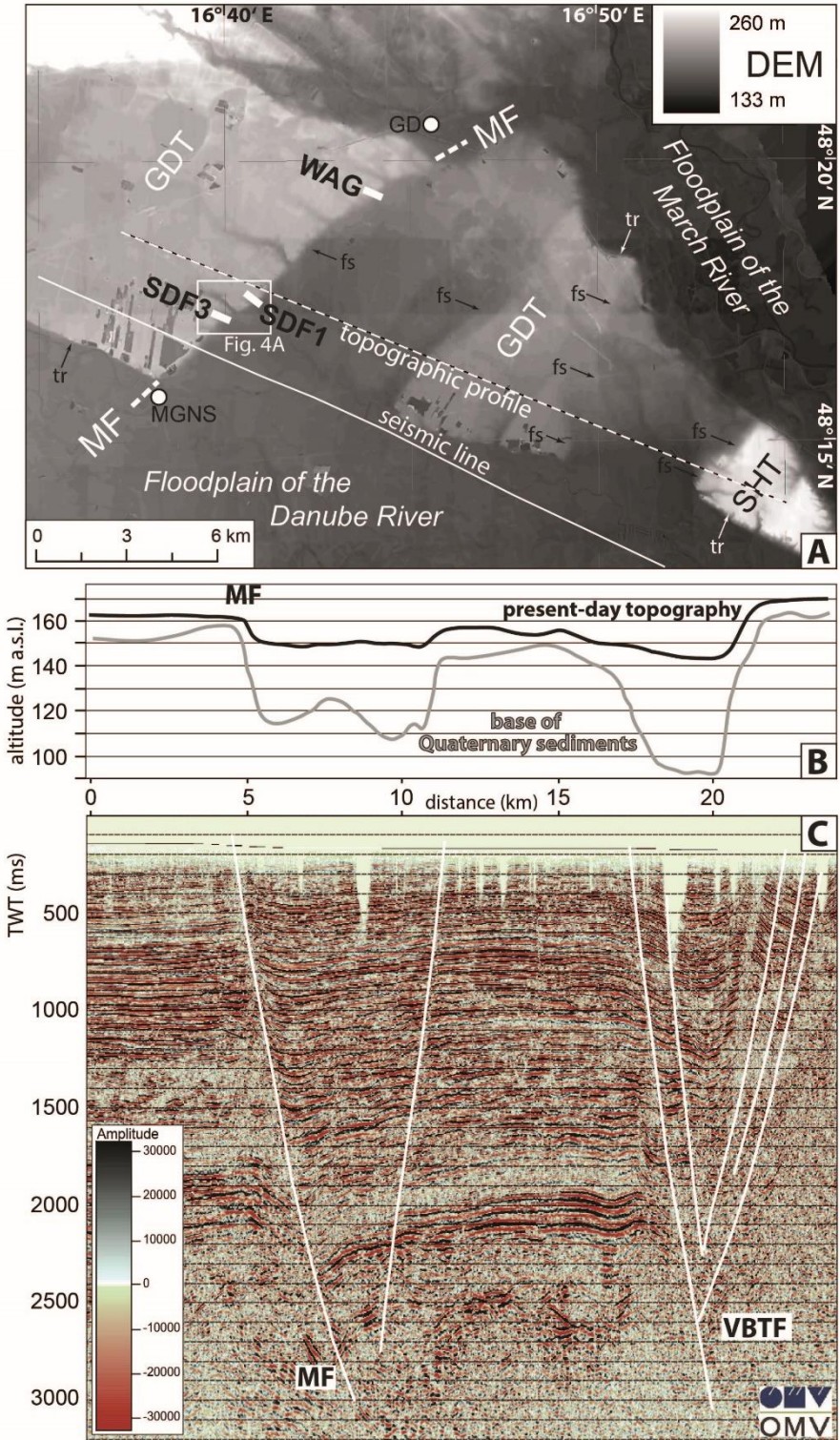

**Figure 3**

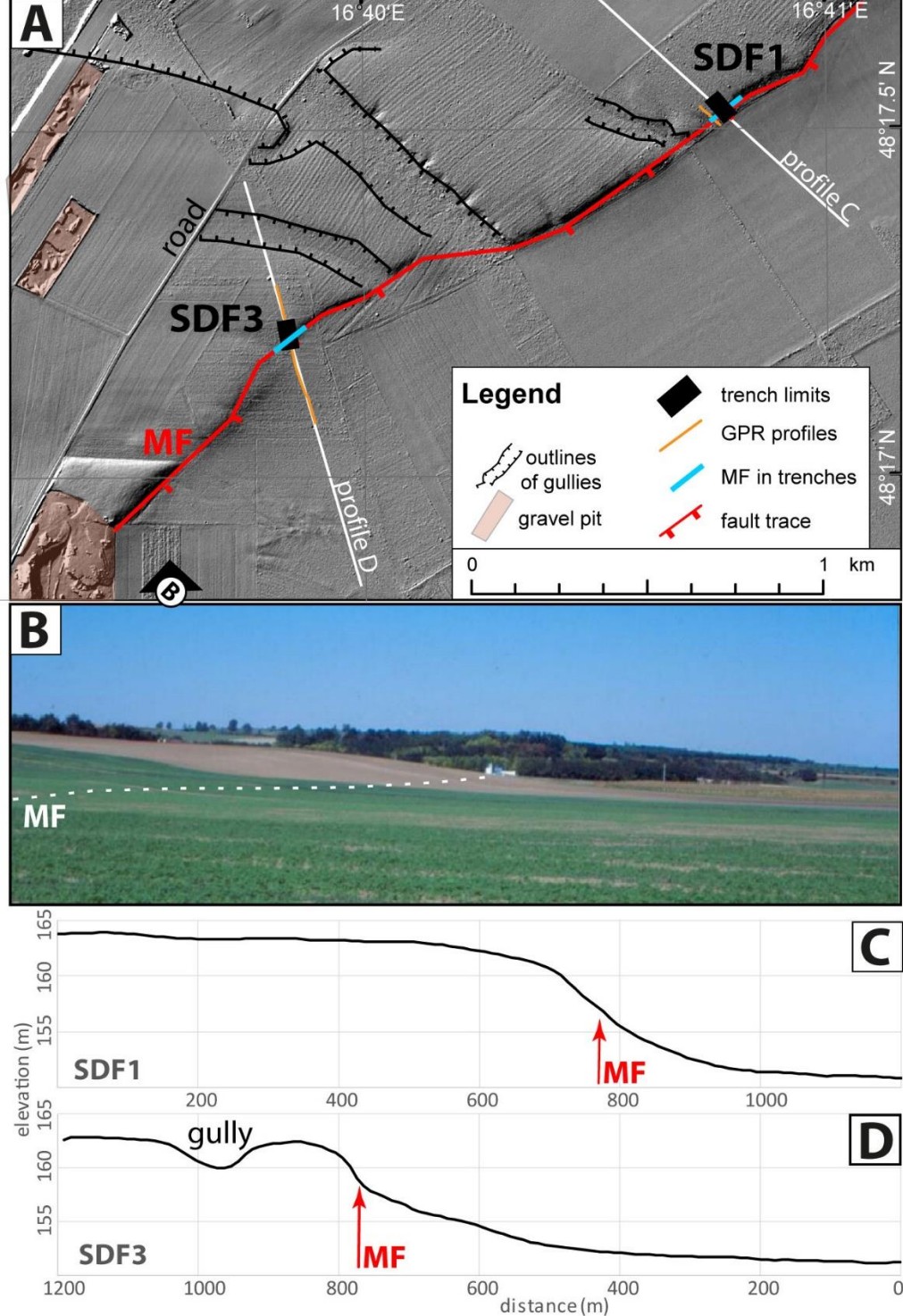

**Figure 4**

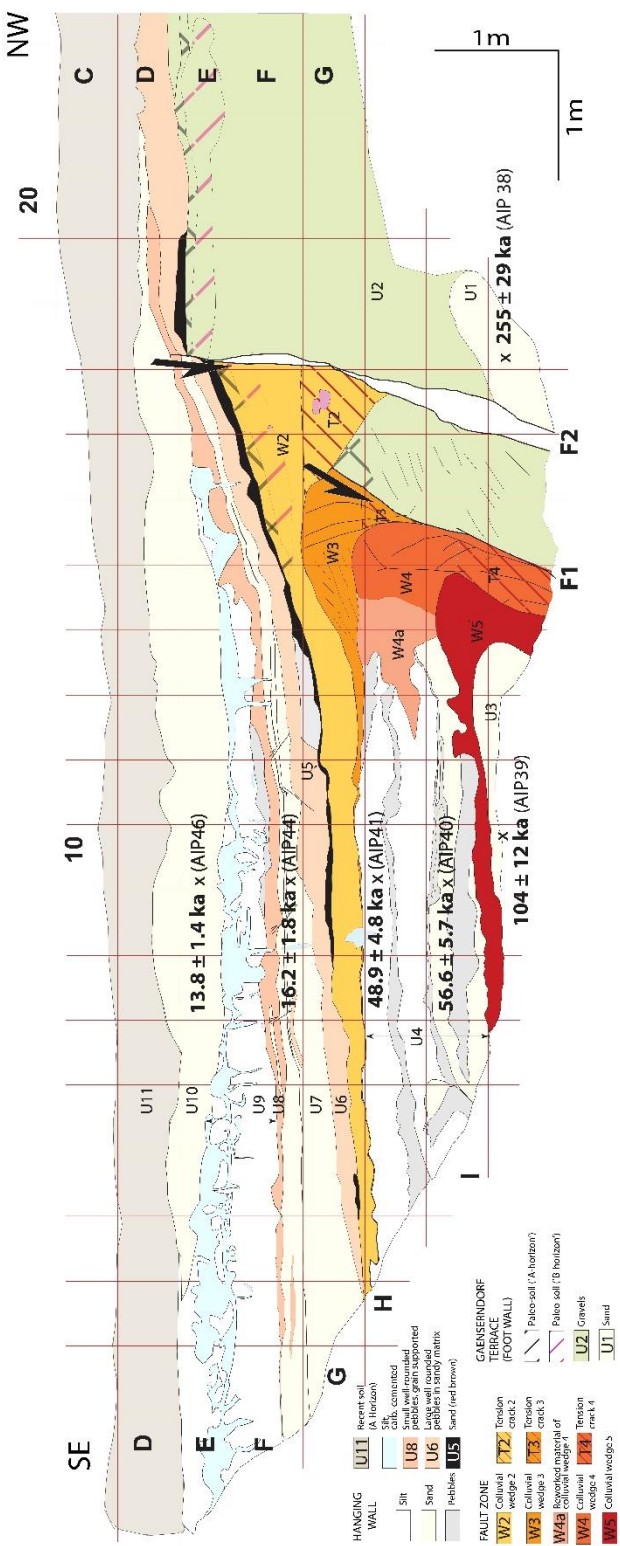

**Figure 5**

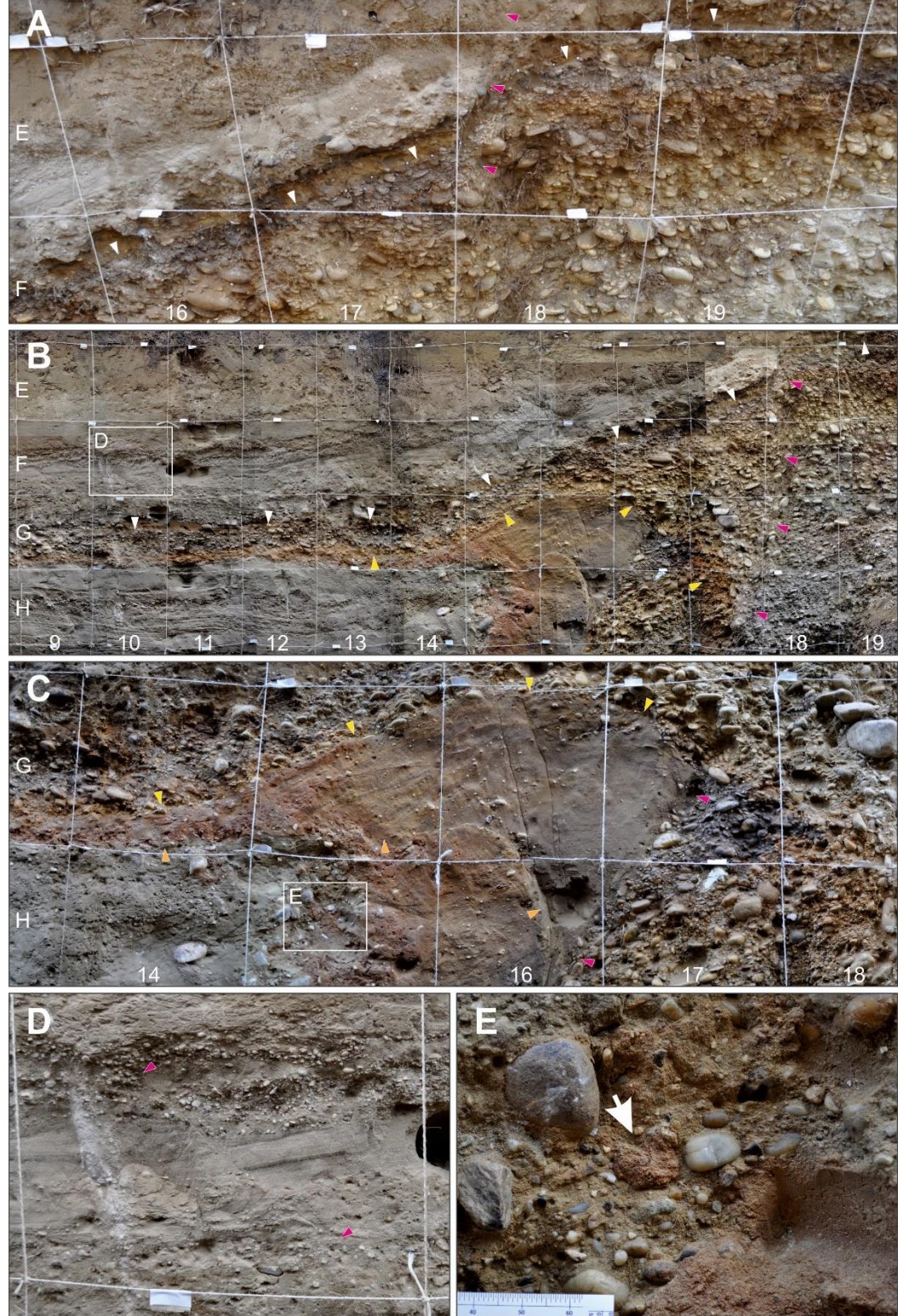

**Figure 6**

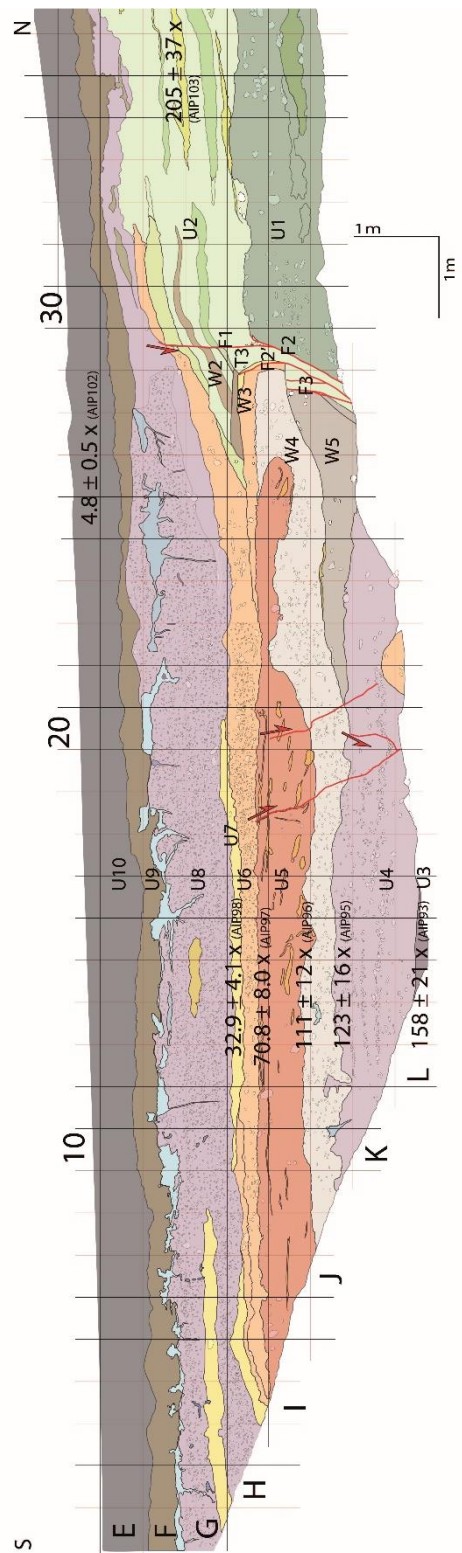

**Figure 7**

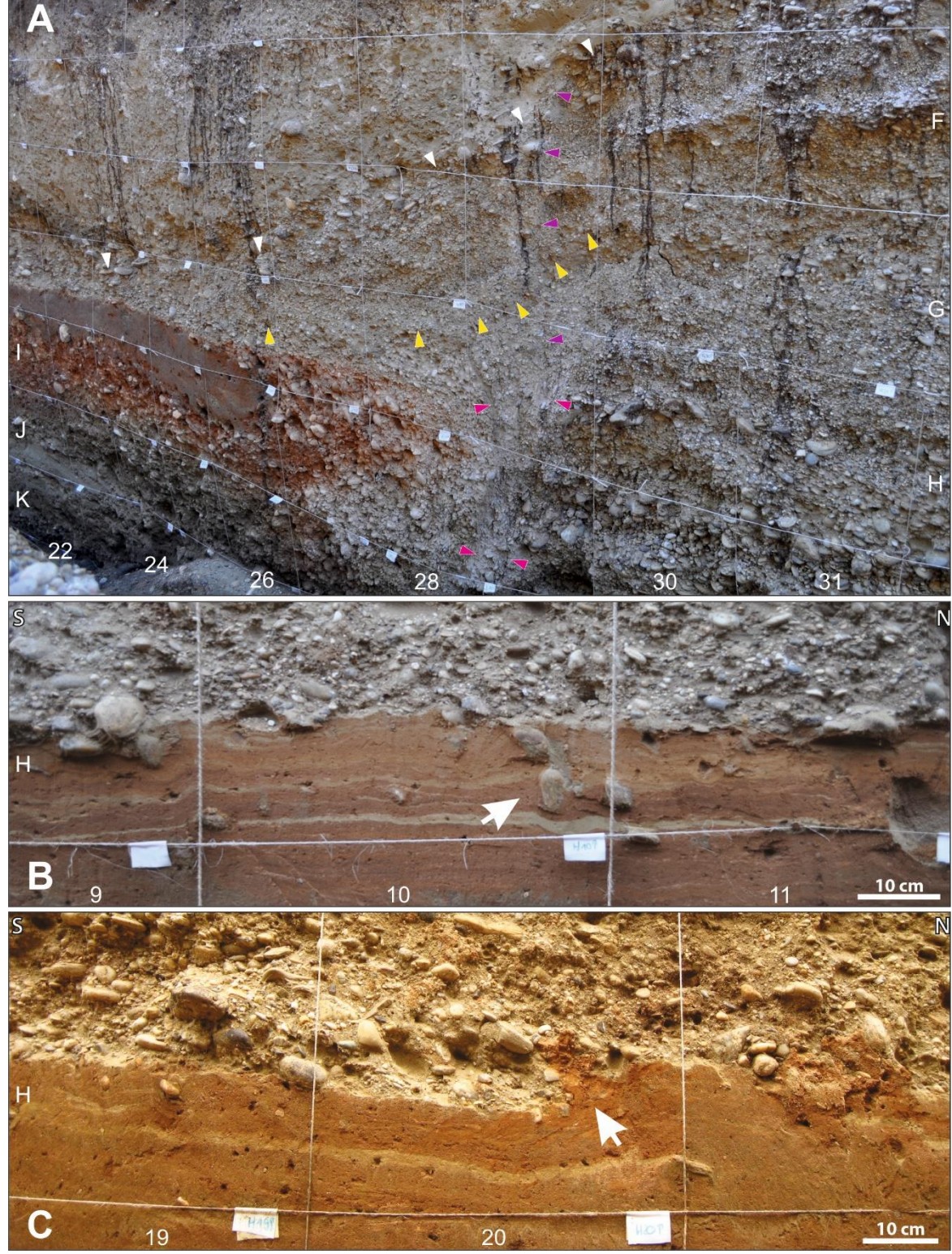

**Figure 8**

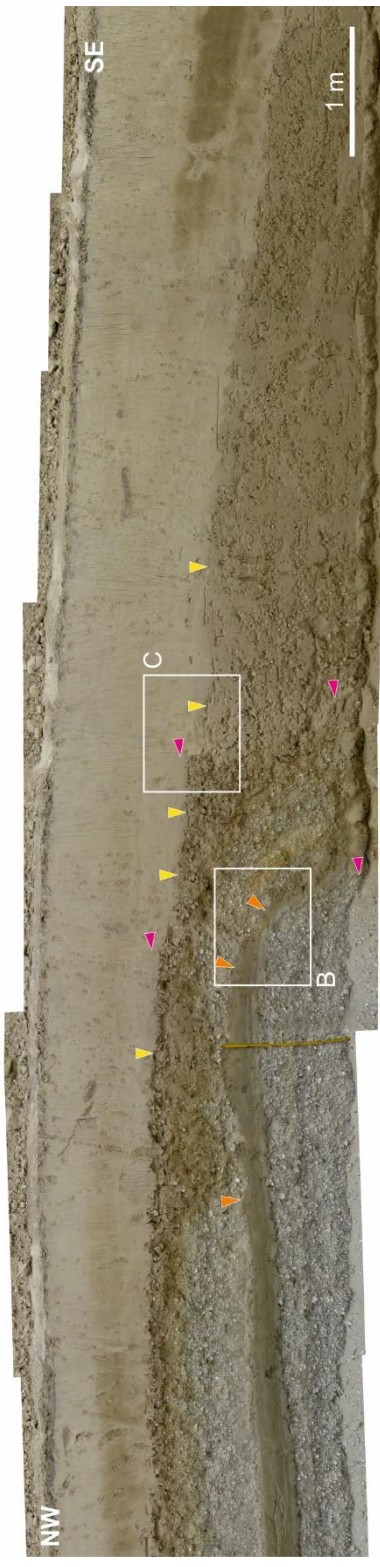

**Figure 9**

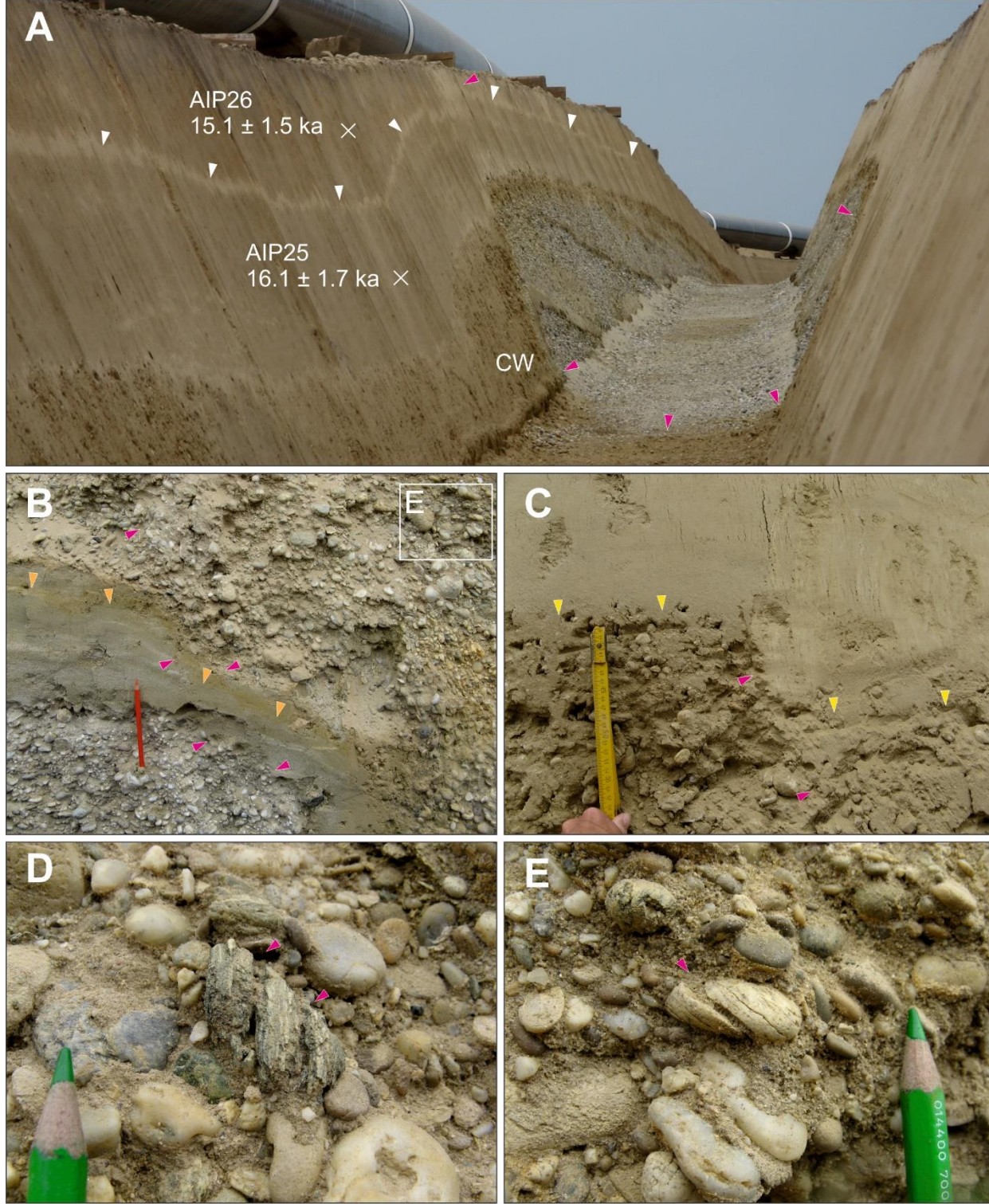

**Figure 10**

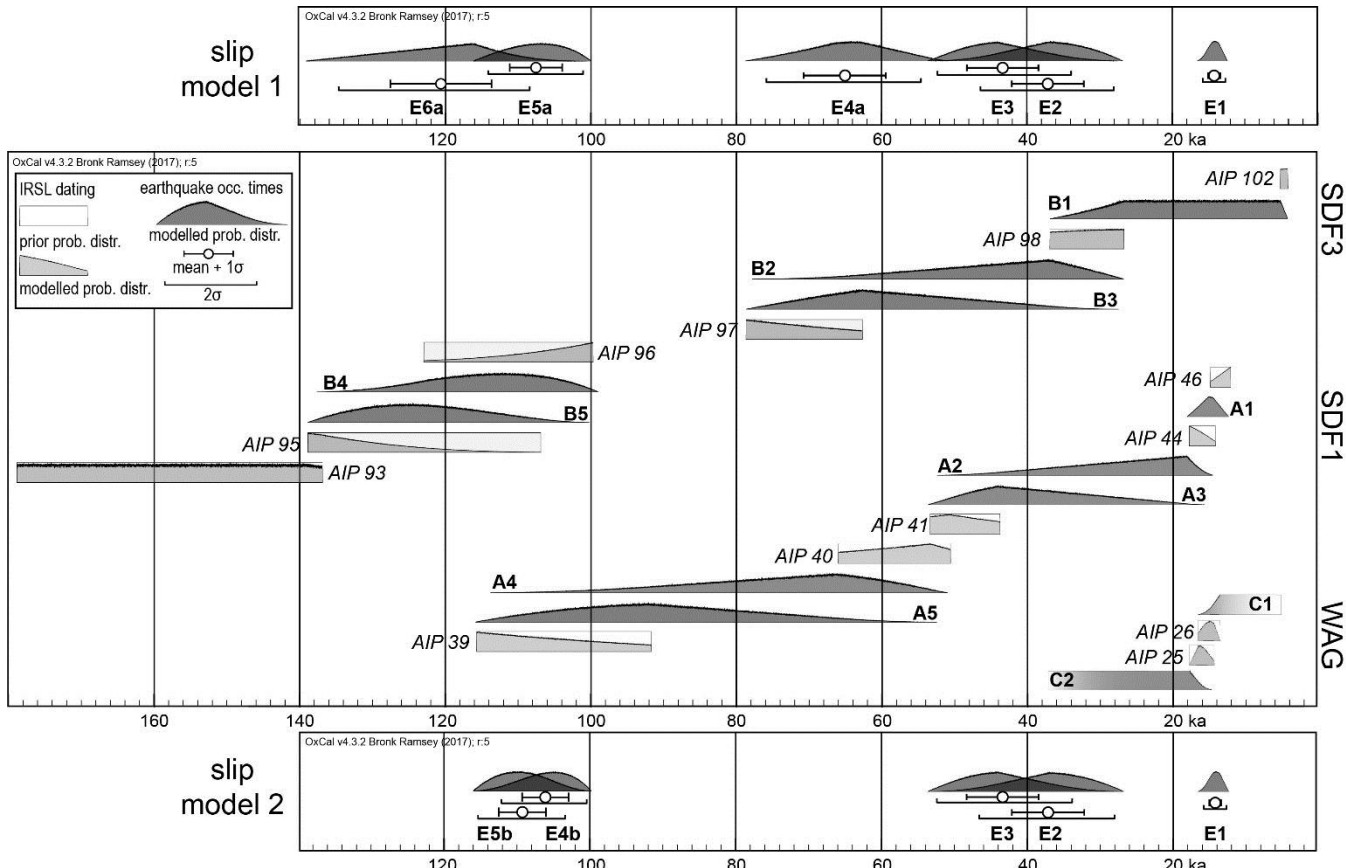

**Figure 11**

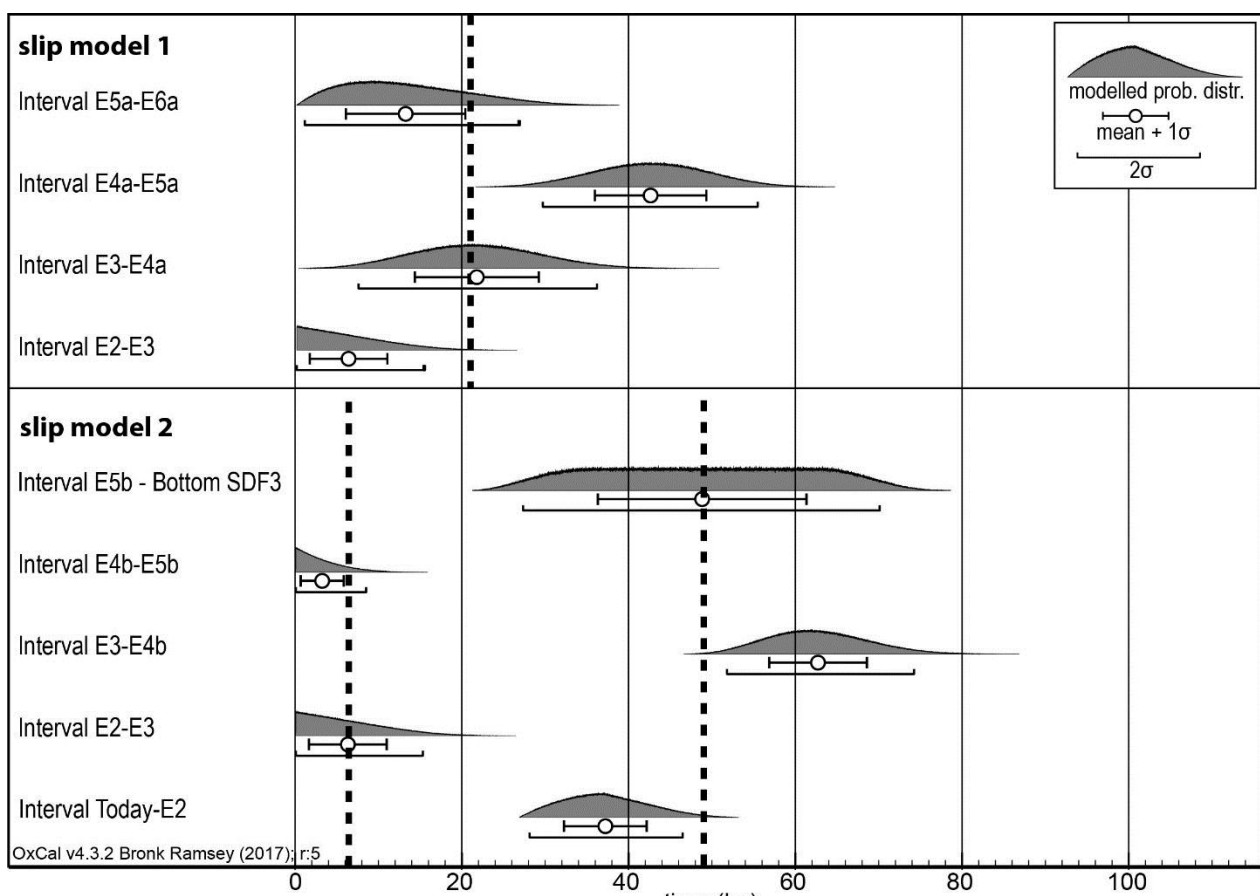

**Figure 12**

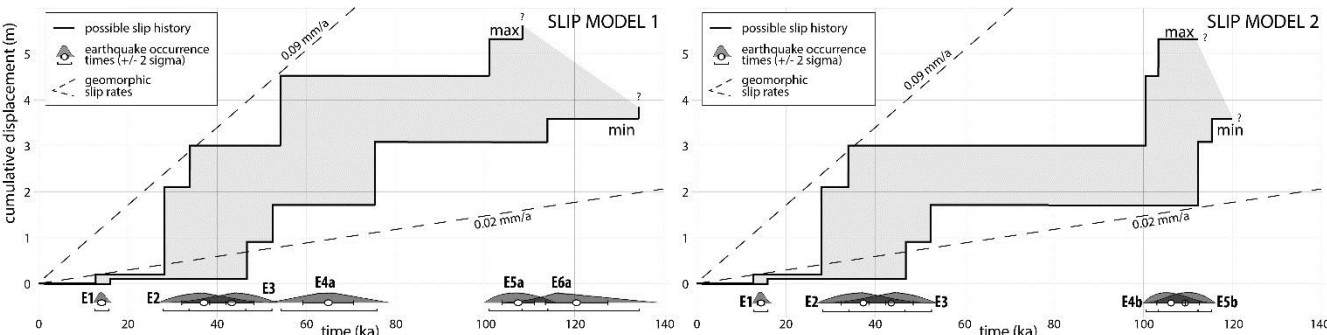

**Figure 13**

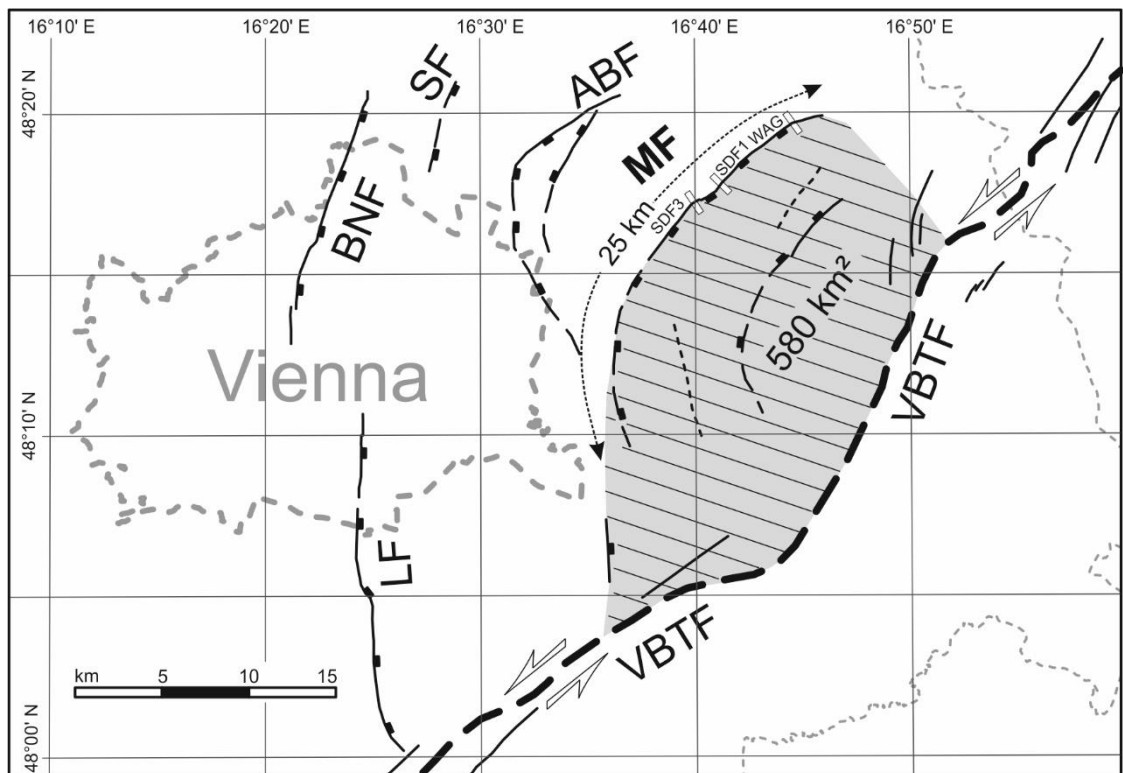

**Figure 14**