# Peer review of "Implications from palaeoseismological investigations at the Markgrafneusiedl Fault (Vienna Basin, Austria) for seismic hazard assessment"

_Natural Hazards and Earth System Sciences, 2017_

## Referee Comment (RC1)

**Implications from palaeoseismological investigations at the Markgrafneusiedl Fault (Vienna Basin, Austria) for seismic hazard assessment**

Esther Hintersberger[1], Kurt Decker[1], Johanna Lomax[2,3], Christopher Lüthgens[2]

[1]Department of Geodynamics and Sedimentology, University of Vienna, 1090 Vienna, Austria
[2]Institute of Applied Geology, University of Natural Resources and Life Sciences (BOKU), 1190 Vienna, Austria
[3]Department of Geography, Justus Liebig University Gießen, 35390 Giessen, Germany

*Correspondence to*: Esther Hintersberger (esther.hintersberger@univie.ac.at)

**Abstract.** Including faults into seismic hazard assessment depends strongly on their level of seismic activity. Intraplate regions are characterized by low seismicity, so that the evaluation of existing earthquake catalogues does not necessarily reveal all active faults that contribute to seismic hazard. In the Vienna Basin (Austria), moderate historical seismicity (Imax/Mmax = 8/5.2) concentrates along the left-lateral strike-slip Vienna Basin Transfer Fault (VBTF). In contrast, several normal faults branching out of the VBTF show neither historical nor instrumental earthquake records, although geomorphological data indicate Quaternary displacement along those faults. Here, we present a palaeoseismological dataset of three trenches crossing one of these splay faults, the Markgrafneusiedl Fault (MF), in order to evaluate the seismic potential of the fault. Comparing the observations of the different trenches, we found evidence for 5-6 major surface-breaking earthquakes during the last 120 ka, with the youngest event occurring at around ~14 ka before present. The inferred surface displacements lead to magnitude estimates ranging between M=6.2±0.3 and M=6.8±0.1. Data can be interpreted by two possible event lines, with event line 1 showing more regular recurrence intervals of about 20-25 ka between the earthquakes with M≥6.5, and event line 2 indicating that such earthquakes cluster in two time intervals in the last 120 ka. Event line 2 appears more plausible. Trench observations also show that structural and sedimentological records of strong earthquakes with small surface offset have only low conservation potential. Vertical slip rates of 0.03-0.04 mm/a derived from the trenches compare well to geomorphically derived slip rates of 0.015-0.085 mm/a. Magnitude estimates from fault dimensions suggest that the largest earthquakes observed in the trenches activated the entire fault surface of the MF including the basal detachment that links the normal fault with the VBTF. The most important implications of these paleoseismological results for seismic hazard assessment are that: (1) The MF needs to be considered as a seismic source irrespective of the fact that it did not release historical earthquakes. (2) The maximum credible earthquakes in the Vienna Basin should be considered to be about M=7.0. The MF is kinematically and geologically equivalent to a number of other splay faults of the VBTF. It must be assumed that these faults are potential sources of large earthquakes as well. The frequency of strong earthquakes near Vienna is therefore expected to be significantly higher than the earthquake frequency reconstructed for the MF.

[Figure]

[Figure]

**1 Introduction**

During the last years, earthquakes tend to "surprise" seismologists, either by unexpectedly high magnitudes (e.g., Sumatra Earthquakes 2004, Tohuko Earthquake 2011) or/and by the fact that the generating faults were either unmapped (Christchurch Earthquake 2010) or assumed to be inactive (e.g., Haiti Earthquake 2009). Thus, it seems to be clear that historical and

5 instrumental seismicity data are not sufficient to fully characterize the seismogenic potential of a certain region (e.g., Camelbeeck et al., 2007, Liu et al., 2011). Especially in regions of low to moderate seismicity, mostly in intraplate settings, observations of historical and instrumental seismicity are not sufficient to accurately estimate the rate of earthquake activity (Liu et al., 2011). Therefore, during the last decade, geomorphological and palaeoseismological approaches have been increasingly used to map active faults and to determine the related slip rates (e.g., Clark et al., 2012 in Australia, and Vanneste

10 et al., 2013, for the Lower Rhine graben system in Central Europe). The results of those studies have dramatically changed the picture and the level of seismogenic potential in the analysed regions, mainly in the following aspects: Firstly, palaeoseismological results show that the magnitude for the maximum credible earthquake may be significantly higher than the magnitude for the largest earthquake observed during historical times (e.g, Central Europe north of the Alps, Figure 1B and references mentioned there). Secondly, the amount of active faults that are considered to be capable of generating

15 earthquakes has been increased (e.g., Clark et al., 2012 in Australia). The identification of such "silent" faults as potential seismic sources has become a vital aspect of geological contribution to seismic hazard assessment. Finally, extension of the observed earthquake records raised the question whether faults (especially single faults within fault systems) show regular earthquake patterns during time (characteristic earthquakes occuring in more or less regular time intervals) or if earthquakes occur in so-called super-cycles, where periods of high activity change with intervals of seismic quiescence (Wallace, 1987,

20 Friedrich et al., 2003). Here, we present results of a paloseismological study, where a dormant active fault has been identified close to the city of Vienna (Austria). Even though there is no historical nor instrumental seismicity that has been recorded along this fault, three trenches across the fault show evidence for five surface-breaking earthquakes. Correlation between the trenches and integration of geomorphological and borehole data helps to identify whether the fault tends to more characteristic or super-cycle behaviour.

[Figure]

**2 Geological setting**

**2.1 The Vienna Basin**

The Vienna Basin has formed as a pull-apart basin between the Eastern Alps and the Western Carpathians in the Middle and Upper Miocene (e.g. Royden, 1985; Decker et al., 2005). It is located between two left-stepping segments of the NE-SW

5   striking sinistral strike-slip Vienna Basin Transfer Fault (VBTF, Figure 1). Faulting along this fault system is related to the NE-directed movement of the block east of the Vienna Basin, caused by lateral extrusion of the central Eastern Alps towards the Pannonian Basin (Ratschbacher et al, 1991, Linzer et al, 1997, 2002). GPS data (Grenerczy et al., 2005) and geological reconstruction of Quaternary sediment deposition within the basin (Decker et al., 2005) indicate that the VBTF moves at horizontal velocities between 1.6 and 2.4 mm/y. However, seismic slip rates calculated from cumulative scalar seismic

10  moments for different segments along the fault are quite heterogeneous, varying from 0.5-1.1 mm/a at the southern and northern tips to an apparently seismically totally locked segment in the central part of the basin, the so-called Lassee segment, close to the city of Vienna (Hinsch et al, 2005, Hinsch and Decker, 2011). Fault mapping using 2D/3D reflection seismic, gravity, and geomorphology shows that these seismotectonically defined segments are delimited by major fault bends including a restraining bend (Dobra Voda) and three releasing bends with negative flower structures overlain by Pleistocene pull-apart

15  basins with up to 150 m growth strata (Beidinger and Decker, 2011). The releasing bends are connected by non-transtensive segments. In addition to the overall geometry of the strike-slip fault with releasing and restraining bends, the transfer of displacement to several normal faults splaying from the strike-slip system in the central part of the basin appears to be an important factor controlling fault segmentation. The splay faults formed during the Middle to Upper Miocene formation of the Vienna pull-apart basin (Decker et al., 2005) and seem to be kinematically linked to the VBTF via a common detachment (i.e.,

20  the Alpine floor thrust, Figure 2, Hölzel et al., 2010, Hinsch and Decker, 2011, Beidinger and Decker, 2011). Those secondary splay normal faults seem to have been seismically inactive during historic times. However, geomorphologic and subsurface geophysical data reveal that those faults indeed show Quaternary displacement of several tens of meters (Chwatal et al., 2005; Decker et al., 2005, Weissl et al., 2017). Moderate historical and instrumental seismicity (Mmax ~ 5.3/ Imax = 8) is concentrated along the VBTF with the 1972 Seebenstein (M~5.3), 1906 Dobra Woda (M~5.7) and the ~ AD 350 Carnuntum

25  (M~6) earthquakes being the largest known events (Gutdeutsch et al., 1987; Decker et al., 2006; Lenhardt et al., 2007). The scarcity of strong earthquakes and the generally low to moderate seismicity result in estimations of Mmax for earthquakes in the Vienna Basin might not exceed M = 6.0 to 6.5 (Lenhardt et al., 1995; Procházková and Šimunek, 1998; Sefara et al., 1998; Tóth et al., 2006). However, those estimations are solely based on historical and instrumental seismicity.

[Figure]

**2.2 The Markgrafneusiedl Fault (MF)**

Our palaeoseismological study is focused on the SE-dipping Markgrafneusiedl Fault (MF) in the central part of the Vienna Basin. It is one of six splay normal faults that  during the Middle to Upper Miocene formation of the Vienna Basin to accommodate transtension at a releasing bend of this sinistral strike-slip fault (Beidinger and Decker, 2011). The location of the fault, fault displacement and fault dimensions are evident from 2D and 3D industrial seismic (Hinsch et al., 2005, Spahic et al., 2013). An exemplary seismic section is shown in Figure 3. Detailed observations based on 3D industrial seismic data on the fault plane suggests that movement along the MF started on different fault segments that eventually merged together as one larger fault (Spahic et al., 2013). Quaternary fault reactivation is inferred from geomorphological evidence of
 a linear scarp paralleling the outcrop trace of the fault, high-resolution geophysical profiling (georadar, reflection seismic, geoelectrics; Chwatal et al., 2005) and the ca. 40 m offset of the base of the Quaternary sediments across the MF (Decker et al., 2005). The visible fault scarp falls together with the SE edge of the Gaenserndorf terrace, building a linear geomorphological step of ca. 12 m height in the present-day topography (Figure 3).

Despite this well documented Quaternary displacement along the MF, no historical seismicity is recorded that can be associated with this fault, except for small earthquakes with magnitudes less than 1.0 that have been recorded close to the MF in the last decade. Whether this apparently slowly moving fault can produce larger earthquakes or it is aseismically creeping, is the key question of our study, during which three trenches (from north to south WAG, SDF1, and SDF3) were excavated across the MF between the villages of Markgrafneusiedl and Gaenserndorf, about 15 km from the city limits of Vienna, the Austrian capital.

**3 Trenching results**

 excavated two trenches along the geomorphic fault scarp between the villages of Markgrafneusiedl and Gaenserndorf (Figure 3A).  40 MHz ground penetration radar (GPR) profiles were carried out showing the location of the MF at the base of the present-day scarp. In addition, a construction pit of a gas pipeline exposed the northern tip of MF, providing additional, but limited, information. In general, all outcrops show similar characteristics: at all trenching locations, the MF is exposed as narrow (1 - 2 m) fault zone consisting of one or two fault branches striking parallel to the regional strike of the fault scarp of the MF (dip direction/dip: ~120/75). The footwall cut by the MF comprises deposits typical for the Gaenserndorf terrace (Weissl et al., 2017 and references therein). The hanging wall of the trenches expose sequences of almost horizontally layered, fine-graded sediments.

[Figure]

**3.1 Trenching at SDF1**

The 40-m-long, 3-m-wide and up to 4 m deep trench SDF1 was located close to the farm house "Siehdichfür", about 20 km from the city limits of Vienna. It was excavated in a small dry valley at the central part of the NE-SW trending geomorphological fault scarp with the exact location of the MF at its base. Trench mapping in the scale of 1:10 covers both,

5    the entire SW wall of the trench and the section around the fault zone at the NE wall. The trench SDF1 exposed about 30 m of Gaenserndorf terrace deposits in the footwall and ca. 10 m of the hanging wall, divided by the 1.5 m wide fault zone of the SE-dipping MF. The fault zone includes two parallel steeply dipping faults F1 and F2, with F2 reaching almost the present-day surface (see Figure 4).

At the NW part in the footwall, alluvial deposits of the Gaenserndorf terrace are exposed, consisting of coarse gravels and

10   boulders. Pebbles show consistent NW-dipping imbrication throughout the entire footwall section. The inferred dominantly SE-directed paleocurrents are comparable to the flow direction of the Recent Danube. In addition, two approximately 8 m wide sandy ancient river channel fills are observed close to the top of the succession. Another sand lense, only partly exposed at the base of the outcrop, is cut by the fault zone. The uppermost 0.5 m of the terrace deposits directly below the recent soil horizon do not show any horizontal consistency and are most probably reworked and repositioned.

15   In the hanging wall SE of the fault zone, three types of sediments are exposed:

(A) Sequences of horizontal layers of light-grey and light-brown silt and fine sand with varying thicknesses up to 20 cm. Sediments show lamination on cm scale and intercalations of cm-thick horizons of coarse sand. The layers also include singular well-rounded pebbles and granules aligned in horizontal layers. Some sand/silt layers show fining-upward trends. Carbonate cementation is observed along the top of the uppermost silt layer and along recent root paths. The sediments are intercalated

20   with and onlap on the wedge-shaped colluvial deposits described below. We relate the deposits to high-stage floods in the floodplain of the Pleistocene Danube.

(B) Colluvial wedge deposits and associated tension crack fills. These colluvial sediments are attached to both faults and decrease in thickness towards the SE (i.e., away from the fault scarp; Figure 4). The steep contact with the SE-dipping faults and the thinning of the deposits towards SE results in a wedge-shape of the sediment layers. The tails of wedges 2, 3 and 5 can

25   be followed throughout the exposed part of the hanging wall. All wedges are associated with tension cracks adjacent to the fault, which are filled with the same material as the overlying wedge. Wedge 5 consists of matrix-supported reddish brown medium gravel with a matrix composed mainly by sand and silt together with a low content of clay. Wedge 4 is delimited by a steep irregular boundary adjacent high-stage flood sediments. While wedge 4 comprises brown to reddish brown fine to medium sand with some fine granules and pebbles in a matrix-supported fabric, the latter include rounded pebble-size clasts

30   of the reddish wedge material interpreted as mud balls. We interpret this peculiar contact to result from the partial erosion of the wedge and the wedge tail during high-stage floods and the re-deposition of the colluvial material by fluvial processes or small slumps. Wedge 3 consists well-sorted reddish brown middle sand with a few pebbles (fine gravel) showing lamination dipping away from Fault 1. These three wedges contrast by their red and reddish-brown colour from the intercalated high-

stage flood deposits. The sedimentary material was identified as redeposited soil, which by its colour, resembles ferretto soils (5YR 4/4, Y5YR 5/4 and 5YR 58 of the standard soil colour chart; L. Smolíková, pers. comm.), which derived from the soil cover of the terrace gravels in the footwall of the MF. While the lower three colluvial wedges (3-5) are bounded by F1, wedge 2 is attached to F2 and overlies the trace of F1as well as the deposits of wedge 3. The wedge consists of large well-rounded

5 pebbles and cobbles oriented sub-horizontally in a grain-supported fabric, similar to the terrace deposits found in the foot wall.
 (C) Fine-grained alluvium and loess. The uppermost part of the sedimentary succession of both the hanging wall and the footwall consists of several thin layers of sand and fine gravel overlain by up to 1 m of unstructured silt and fine sand. The latter is transitional to the overlying dark brown to black soil horizon. The succession is interpreted as alluvium of the dry valley and loess-like sediments or redeposited loess. Fault 2 offsets the alluvial sand layers for about 15 to 20 cm, but terminates

10 within the overlying loess-like sediments several cm above the base of the layer.
Structural data obtained from the two faults exposed in the outcrop show that both faults strike parallel to the regional strike of the fault scarp of the MF. The faults are marked by bands of pebbles with preferred orientations parallel to the fault planes. Pebbles in the 75 cm thick fault block between the two faults show orientations, which geometrically resemble S-C-type fabrics. Deformation bands are found in the sand wedges 3 and 4 and the related tension cracks. Detailed mapping reveals that

15 these microfaults do not penetrate into the colluvial wedge 5 most probably due to the higher clay content of these sediments. The deformation bands show orientations consistent with the main faults of the outcrop. At the lower parts, the deformation bands are dipping parallel to F1 (dip direction/dip 130/80). The upper parts of the deformation bands are rotated away from the fault resembling horsetail splays. The orientations of the sub-vertical deformation bands vary between 303/78 and 330/78. In addition to some major deformation bands, which are traced for about 1 m across the profile, there are shallow dipping

20 deformation bands with comparably large normal offsets up to several mm (145/20). Finally, small-scale normal faults with displacement in the order of several centimeters are observed within the uppermost layers that have been also affected by the youngest displacement along F2 (Figure 5E).

**3.1.1 Evidences for seismic events observed within trench SDF1**

Offset of alluvial sand layers at the tip of F2 provides direct evidence for the youngest surface-breaking slip event A1. The

25 small-scale faults observed at the same stratigraphic level are further indications for an earthquake at this fault. A1 is offsets and postdates colluvial wedge 2. The observed colluvial wedges, their geometrical relation to the adjacent faults, and the sediment-filled extension fissures prove four distinct events (A2 to A5) of rapid co-seismic displacement at the MF. In addition, the existence of deformation bands within the sandy colluvial wedges 3 and 4 indicates further deformation of both wedges during younger slip events at the MF. Among the earthquakes excavated by the trench, only slip associated with A1 is directly

30 constrained by the offset of layers correlated across Fault 2. Evidence for the earthquakes A2 to A5 comes from the colluvial wedges 2 to 5 and the refilled tension cracks 2' to 5' below the wedges. Following the generally accepted rule of thumbs that colluvial wedge height is approximately half of the surface displacement of an earthquake (McCalpin, 2008), the measured maximum thickness of each colluvial wedge can be used to estimate the minimum displacement for the associated event.
[Figure]

[Figure]

**3.2 Trenching at SDF3**

Trenching in the Vienna Basin continued with the opening of a second trench SDF3 across the same fault. This 33 m long, 3 m wide and up to 5 m deep trench is located about 1.5 km SW of the first trench SDF1. Trench mapping in the scale of 1:10 covers both, the entire W wall of the trench and the section around the fault zone exposed in the terraced E wall (Figure 6).

5   The about 0.5 m wide fault zone of the SE-dipping MF divides the N-S trending trench into two parts. The footwall W of the fault mainly consists of gravels of the Gaenserndorf terrace whereas the hanging wall in the E shows a succession of fluvial sediments, colluvial deposits originating from the uplifted footwall and reworked loess-like sediments.

The footwall mainly consists of poorly sorted, well rounded sandy gravels within a grain-supported fabric. Components mostly include metamorphic rocks, gneisses, quartzite along with minor sandstone and limestone. The few magmatic components

10   found within the gravels completely weathered. The lower 1-1.5 m of the terrace excavated in the trench contains coarse cobbles with diameters up to 25 cm. The upper part shows typical characteristics of braded river deposits, including crossbedding of better-sorted gravel layers intercalated with sand layers of up to 0.5 m of thickness and several meters of lateral extent. Furthermore, the part consists of gravel and small cobbles with diameters up to 10 cm. All layers show a slight inclination towards the SE. Throughout the terraces deposits, vadose gravitational carbonate cementation, so-called dripstone

15   cementation, along the lower side of larger gravels is observed.

The hanging wall consists of horizontally layered sediments of different origin. In the following, we describe the most important units of the hanging wall, starting with the lowermost unit. Unit 1 consists of intercalated beige to grey, medium to fine sand and gravel layers consisting of well-rounded, poorly sorted clasts. The thickness of the layers varies between ~3 cm and 20 cm, whereas the sand layers are generally thicker than the intercalated gravel layers. This sequence is the result of

20   alternate high-stage Danube floods (sand layers) and erosional events transporting gravels from the footwall into the hanging wall. Those erosional impulses may be triggered by heavy-rainfall events. The contact to the overlying unit 2 is clearly identified. Unit 2 consists of matrix-supported conglomerate with clay-rich, Fe-rich red fine sandy matrix and poorly sorted, well-rounded clasts with diameters up to 15 cm. Grain sizes decrease with increasing distance from the fault, as well as the layer thickness from 70 cm directly at the fault to less than 50 cm further away. The contact to the overlying unit 3 is diffuse.

25   Unit 3 consists of red clay-rich Fe-rich fine sand with intercalated layers and up to 5 cm thick lenses of brownish slightly coarser sand without clay or Fe components. The layering shows a slight inclination of a few degree towards the NW, i.e., towards the fault. The material is typical for distal flood basin deposits of fluvial environments. A few well rounded clasts with diameters of 2 - 20 cm have been observed. Their distribution suggests that they may be dropstones. The contact between this unit and the overlying unit 4 is characterized by a generally horizontal sharp contact, which has been affected by

30   liquification, either caused by the occurrence of an earthquake or by the deposition of the overlying coarse gravels over the still water-bearing sediments of unit 4. Furthermore, a burrow of small animals, refilled with beige coarse sand is observed at the bottom of this layer. Unit 4 consists of grain-supported conglomerate with well-rounded, poorly sorted cobbles with grain sizes up to 10 Those gravels originate from the footwall and form a colluvial wedge, which decreases in thickness with

increasing distance to the fault. Unit 5 consists olive-coloured medium sand with rare mica components. This 10 cm thick unit decreases in thickness towards the fault. This fact, together with the colour of the sand, suggests that it is a flood deposits of the Danube. Unit 6 covers both, the hanging and the foot wall and consists of a matrix-supported conglomerate with silt matrix and around 25% of components that consist of poorly sorted, well-rounded pebbles with grain sizes up to 3 cm. The silt matrix

5  consists of reworked loess that has probably eroded from the footwall, including smaller clasts from the Gaenserndorf terrace. In the top of this unit, secondary carbonate cemented a horizontal layer of up to 30 cm thickness. The layer is observed throughout the entire hanging wall. Carbonate cementation occurs due to meteoric waters dissipating carbonate from the upper layers and precipitating it at lower pH values in greater depth. Conjugated planar carbonate fissures of up to 60 cm length branch off from the cemented layer. They strike approximately parallel to the orientation of the MF. Unit 7 is the AC soil

10  horizon, consisting of a matrix-supported conglomerate of fine sand and 30-40% of components containing partly angular and rounded pebbles with grain sizes up to 2 cm.  The contact to both the underlying and overlying units, is rather diffuse. Finally, unit 8 is the A soil horizon that increases in thickness with increasing distance to the MF. Its thickness coincides with a layer of silt or loess that has been reworked as soil.

Structural data obtained from the outcrop show that both faults strike parallel to the regional strike of the fault scarp of the MF

15  (dip direction/dip: 116/74). The MF is marked by the contact between the footwall gravels and the more sandy deposits of the hanging wall. In addition, at the lower 1.5 m, clasts within a zone of about 50 cm to the fault are rotated parallel to the fault (dip direction/dip: 116/69). The upper part of the MF is only marked by a small band of rotated clasts. However, in this upper part of the fault, layers that can be correlated on both sides of the fault, are displaced by about 15 cm and, therefore, indicate the youngest movement along the fault. In addition to the main faults, several conjugated sets of normal faults are observed

20  within lower units of the hanging wall. These faults are consistently oriented parallel to the MF. The NW-dipping antithetic faults (dip direction / dip: 303/79) are generally longer than the SE-dipping faults (dip direction / dip: 137/72). Displacement observed along the faults is in the range of about 10 cm in the lowest unit 1, and up to 1 cm in the reddish clay-rich unit 3. None of the small faults seem to penetrate into the gravels overlying the reddish clay-rich sand layer (unit 3). Within this layer, the small faults are recognised as a few mm thin deformation bands, most probably filled with carbonate cement, and show

25  almost no displacement. . The sand layers of the lowermost unit 1, consisting of intercalated layers of sand and matrix-supported gravels, comprises small deformation bands with lengths up to 20 cm. They are arranged parallel to the small faults and accordingly dip towards the SE or the NW.

**3.2.1 Evidences for seismic events observed within trench SDF3**

The MF within the trench SDF3 is a very narrow fault zone of 0.5 m width at its lowest point excavated within the trench, and

30  reduces to a fault represented only by a few rotated clasts in the uppermost part. However, this reduction of thickness is not a continuous, but occurs in distinct steps. Those steps can be related to different earthquakes; The oldest earthquake that can be identified within the trench is B5 that created a colluvial wedge along the fault trace F3. This fault trace is then covered by another colluvial wege, which was most probably created by movement along F2 during B4. Evidence for the event B3 is a

tension crack between F2 and F2', that is also identified by the thin sand layer that is smeared into the crack, parallel to F2'. B2 is identified by a ~ 0.8 m thick colluvial wedge. However, the fault strand bounding this colluvial wedge is not obvious. This situation may be explained by the following scenario: In the case that coseismic surface rupture offset unconsolidated water-saturated sandy gravel, it seems plausible that no long-standing free surface and colluvial wedge adjacent to the fault

5   plane could form. Instead, the offset soft sediment may have collapsed during or shortly after the earthquake forming a wedge-shaped deposit, which overlies the uppermost part of the ruptured fault. The same geometry may result from geli-solifluction under periglacial conditions when material glides down to the hangingwall destroying a previously formed free surface. The latter scenario is supported by the observation of a smooth change between the horizontal layers of the terrace and the inclined layers in the colluvial wedge. The described situation allows for two different interpretations of the surface displacement of

10  B2. Interpreting the wedge-shaped deposit as a classical colluvial wedge adjacent to a fault plane which is not readily seen due to unfavourable outcrop conditions, a minimum displacement can be estimated by multiplying the maximum wedge height by two (McCalpin, 2008), which would result in a displacement of 2 x 0.8 m = 1.6 m for B2. In case that the wedge formed by free surface collapse of water-saturated sediment or gelifluction the coseismic surface displacement be approximately the same as the colluvial wedge height, i.e. 0.80 m.

15  Insights for the youngest event B1 in the trench are more obvious. Displacement of the upper layers for ~ 10 cm affected all layers excluding only the soil horizons (units 7 and 8), suggesting that even with such a small displacement of only 10 cm, the event B1 ruptured the surface.

**3.3 Trenching at WAG**

Additional evidence for active faulting at the MF are available from the construction pit of a gas pipeline, which crosses the

20  northern part of the fault scarp close to the city of Gaenserndorf, 6 km north of trench SDF. The outcrop revealed a 1-m-wide localized fault zone (Figure 8). The fault cuts light-grey gravel and sand of the Gaenserndorf Terrace and overlying loess-like sediments (silt, fine to medium grained sand) constituting its footwall. The exposed hanging wall succession includes poorly
 sorted sandy gravel, which is then overlaid by a banded sequence of silty sediments. This cover layer can be found all along the pipeline construction pit and has been described in detail by Weissl et al. (2017). Both the hanging wall and footwall are

25  overlain by c. 30-50 cm thick brown soil, which has been removed prior to the excavation. The exposed fault zone consists of several deformation bands within the terrace gravel marked by aligned and fractured pebbles, faults offsetting sand layers, and faults offsetting the contact between gravel units and the overlying cover silts (Figure 9). Several sheared pebbles indicate dip-slip movement along the deformation bands. The displacements of these faults are between 10 and 20 cm. On the southern wall a fault cuts up through the entire silty section to the base of the overlying soil, offsetting a thin white layer within the

30  upper part of the cover silty sediments by about 20 cm.

[Figure]

[Figure]

**4 Luminescence dating**

Luminescence dating is commonly used to date the time that feldspar and/or quartz grains in sandy or silty sediments were not exposed to sunlight, and therefore to constrain deposition ages of those sandy or silty sediment bodies. Regarding the physical background and the basics of luminescence dating methods we refer to previously published review papers of Preusser et al.

5   (2008), Wintle (2008), and Rhodes (2011).

**4.1 Sampling and experimental setup**

In analogy to the procedure described by Weissl et al. (2017), samples were collected in the field by driving an opaque steel cylinder into the freshly cleaned sediment surface and transferring the material into light tight plastic bags. All subsequent sample preparation steps were conducted under subdued red light conditions in the Vienna laboratory for luminescence dating.

10  Samples were first dried and dry sieved. The grain size fraction of 100 - 200 mm was used for further preparation steps. The material was subjected to 15% HCl to remove carbonates, treated with $Na_2C_2O_4$, (0.01 N) to disperse clay particles, and with 10% $H_2O_2$ to dissolve organic components. Quartz and feldspar separates were obtained by density separation using LST Fastfloat.

In this study, we used potassium-rich feldspar as luminescence dosimeters for age determination. All fractions were measured

15  with small aliquots of 1 mm diameter mask size using a grain size fraction of 100 - 200 mm. All measurements for determination of the equivalent dose were conducted in the Vienna laboratory for luminescence dating on RISØ TL-OSL DA 20 automated luminescence reader systems (Bøtter-Jensen et al., 2000, 2003). For De determination of the feldspar fraction, a conventional SAR IRSL protocol was applied (Wallinga et al., 2000; Blair et al., 2005), using a preheat of 250°C for 20 s and a stimulation at 50°C for 300 s. Stimulation was carried out with IR-LEDs, and signals were detected after passing through a

20  blue interference filter (410 ± 20 nm). Doses were determined on small multi-grain aliquots (mask-size 1mm). Over-dispersion (Galbraith et al. 1999) was below 11% in all samples confirming a generally well-bleached nature of the sediments. It needs to be stressed that the feldspar based ages were not corrected for fading. Fading describes an anomalous signal loss very commonly observed for potassium-rich feldspar (Wintle, 1973). If not corrected for, fading leads to the underestimation of the burial age. However, samples from the same study area investigated by Weissl et al. (2017) showed little or no fading, as

25  demonstrated by a comparison between quartz and feldspar luminescence ages. Nevertheless, all ages presented here need to be treated with caution for potential age underestimation. Radionuclide concentrations for dose-rate estimation were determined on ~900 g of bulk sediment using high resolution, low-level gamma-spectrometry. Samples were first dried, homogenised and stored in sealed Marinelli beakers (500 ml, about 1 kg dry weight) for at least a month to establish secondary secular radon equilibrium. Measurements were conducted using a Canberra HPGe detector (40% n-type). Relevant

30  luminescence data is listed in Table 2.

[Figure]

**4.2 Sedimentary and tectonic context**

In general, luminescence dating results fit well to the stratigraphic hanging wall sedimentary sequences observed in both trenches, showing continuous decrease in age from the bottom towards the top. In addition, ages derived for the Gaenserndorf terrace in the footwall fit well with other ages from this terrace (Weissl et al., 2017).

Regarding to trench SDF1, Event A1 is well constraint between 13.8 ± 1.4 ka and 16.3 ± 1.8 ka by samples from an offset sand layer and overlying undeformed sediments. Events A2 and A3 are bracketed by the ages inferred for A1 and the undeformed sediments below the colluvial wedge related to A3 and therefore occurred between 16.1 ± 1.7 ka and 48.9 ± 4.8 ka. The ages of A4 and A5 are similarly constrained in the trench SDF1 by sediments below and above the colluvial wegdes. Both events occurred between 56.6 ± 5.7 ka and 104 ± 12 ka.

10 In trench SDF3, samples (AIP93-AIP102) defining the chronology of the stratified hanging wall between 158 ± 21 ka and 4.8 ± 0.5 ka were dated in addition to two more samples (AIP103 and AIP114) determine the minimum age of the footwall to 205 ± 37 - 259 ± 35 ka. Those obtained ages agree well with other IRSL ages for the Gaenserndorf terrace (Weissl et al., 2017). In addition, IRSL data of the hanging wall constrain roughly the occurrence times of the 5 observed paleo-earthquakes along the main fault. While B1 is constrained to have occurred between 4.8 ± 0.5 ka and 32.9 ± 4.1 ka, B2 and B3 can only limited to

15 occur together within the time interval between 32.9 ± 4.1 ka and 70.8 ± 8.0 ka. Also for B4 and B5, a common time interval between 111 ± 12 ka and 123 ± 16 ka can be determined. At the trench site WAG, both the uppermost and the lowermost sediments of the fine-graded silty to sandy cover were dated by IRSL revealing ages of 15.06 ± 1.52 ka and 16.1 ± 1.7 ka, respectively (samples AIP25, 26, Weissl et al., 2017).

[Figure]

[Figure]

**5 Correlation of events between sites**

Palaeoseismological investigations along the MF include three locations, the trenches SDF1 and SDF3 as well as the pipeline outcrop WAG. For all three locations, detailed mapping and dating have been carried out and described above. Evidence for 5 possible earthquakes have been observed in the trenches SDF1 (named A1-A5) and SDF3 (B1-B5), while in the pipeline

5   outcrop WAG, observations indicate two paleo-earthquakes (C1 and C2). Figure 10 shows the constraints of earthquake occurrence times for each observation point. Based on this information, together with comparison of trench observations and displacement estimates for all three outcrops, we correlate the observations and results to generate a synthesis of the earthquake occurrence along the MF. In the following, we discuss each earthquake and the possible correlations between the trenches as well as the resultant age, displacement and magnitude estimate, starting with the youngest.

10   **5.1 Event 1 (A1 = B1 = C1)**

In all three outcrops, the youngest event is evident from a measurable offset of layers across the MF. At trench site SDF1, the youngest event A1 shows displacements of 15-25 cm and occurred in the time range between $13.8 \pm 1.4$ ka and $16.3 \pm 1.8$ ka. At trench site SDF3, markers have been displaced by the youngest event B1 by 10-15 cm. The ISRL data limits the occurrence time of B1 to the time range between $4.8 \pm 0.5$ ka and $32.9 \pm 4.1$ ka. In the pipeline outcrop WAG, the loess cover is dated between $15.1 \pm 1.5$ ka and $16.1 \pm 1.7$ ka. It is displaced by 17-20 cm. Therefore, C1 must have happened after $15.1 \pm 1.5$ ka. The occurrence time for E1 is thus constraint to the time interval between $13.8 \pm 1.4$ ka and $15.1 \pm 1.5$ ka (Table 2). Using the empiric relationship between surface displacement and magnitude (Wells and Coppersmith, 1994) for the maximum displacement of 25 cm, E1 had the magnitude $M = 6.2 \pm 0.2$. The average displacement of 17 cm would lead to a similar magnitude $M = 6.3 \pm 0.3$.

20   **5.2 Event 2 (A2 = B2 = C2)**

Event E2 is also observed in all three outcrops as a triangular-shaped colluvial wedge mainly consisting of reworked gravels that derived from the terrace in the footwall. In addition, the top of each of those colluvial wedge deposits is displaced by E1, confirming the correlation of the colluvial wedges to the penultimate seismic event E2. At trench site SDF1, the displacement related to A2 (1.5-1.9 m) is estimated from the height of the associated colluvial wedge (0.75 - 0.95 m). IRSL samples constrain

25   the occurrence time for A2 to the time interval between $16.1 \pm 1.7$ ka and $48.9 \pm 4.8$ ka. At trench site SDF3, the interpretation of deposits related to B2 are more ambiguous (see sec. 3.2), and therefore, the estimated displacement is either 0.8 m (collapsed free face scenario) or 1.6 m (colluvial wedge scenario). B2 is constrained between $32.9 \pm 4.1$ ka and $70.8 \pm 8.0$ ka. In the pipeline construction pit the displacement of E2 can only be constrained to exceed 1 m by the colluvial wedge as the base of the wedge is not exposed. Time constraints are limited to the ante quem of $16.1 \pm 1.7$ ka by the age of the overlying loess

30   covering the colluvial wedge.

[Figure]

Combining the individual time constraints in each trench site allow to determine the occurrence time of E2 between 32.9 ± 4.1 ka and 48.9 ± 4.8 ka. Magnitude calculation using the maximum of the observed surface displacements results in a magnitude of M = 6.8 ± 0.1) (Wells and Coppersmith, 1994). With the maximum value for the observed surface displacement coming from trench SDF1, the magnitude estimate does not depend on the interpretation for the B2 deposits in trench SDF3.

**5.3 Event 3 (A3, probably correlated with B3)**

For this event, a correlation based on field observations between the trenches SDF1 and SDF3 is not as clear as in the cases of E1 and E2, especially since the evidence for B3 does not allow to determine a displacement for this possible event. However, the maximum height of a well-developed sandy colluvial wedge in SDF1 gives a good estimate of an earthquake with M = 6.6 ± 0.1. Because of the similar stratigraphic constraints, the possible occurrence time of E3 is constrained by the same limits as E2, so that E3 occurred also between 32.9 ± 4.1 ka and 48.9 ± 4.8 ka.

**5.4 Event 4 (A4, if correlated with B3)**

Another possible correlation scenario between the trench sites SDF1 and SFD3 is the correlation of A4 and B3 (event line 1 in Figure 10), mainly due the loose time constraint of B3. If A4 and B3 are correlated to the same seismic event E4, the overlap of possible occurrence times of A4 and B3 narrows the resultant occurrence time for E4 to the interval between 56.6 ± 5.7 ka and 70.8 ± 8.0 ka. Observations of the maximum wedge height at trench site SDF1 indicate the magnitude of A4 (and therefore for E4) to M = 6.8 ± 0.2.

**5.5 Event 5 (A4, if correlated with B4)**

In an alternative scenario, A4 could also correlate to B4 (event line 2 in Figure 10). In this case, the combined occurrence time for the resultant seismic event E5 must be older than 111 ± 12 ka and younger than 104 ± 12 ka. Thus, the time constraint would be thigh, dating E5 to the overlap of the uncertainties of the IRSL age dating between 100 ka and 116 ka with a mean at 107.9 ± 8.0 ka. Similar to E4, the magnitude for E5 can be estimated from the observations of the maximum wedge height of A4 at trench site SDF1, indicating a magnitude for E5 of M = 6.8 ± 0.2.

**5.6 Event 6 (possible correlation between A5 and B4) and Event 7 (possible correlation between A5 and B5)**

The later back in time, the more uncertain the correlation between both trench sites becomes. So, whether A5 and B5 are correlated to one event E6 or A5 and B4 are correlated to one event E7 is not clearly determined neither by observations nor dating. Both alternatives are possible and only depend on whether A4 is correlated to B3 or to B4. In event line 1, where A4 s correlated with B3, it appears reasonable to assume subsequently that A5 = B4. In contrast, if A4 = B4 (event line 2), the remaining correlation for the next older event would be A5 = B5. Due to the loose time constraints in the lower part of all trenches, the occurrence times for E6 and E7 are identical to those used for E5, leading to a time window of approximately

100-116 ka where either E6 or E7 occurred. Since both magnitude estimates for E6 and E7 are based on the maximum colluvial wedge height of A5 from SDF1, the magnitude of E6 and E7 is M = 6.5 ± 0.1)

**5.7 Event 8 (B5, if not correlated with any event in SDF1)**

In the case that B5 is not correlated with any events recorded in trench SDF1 (event line 1), the timing of E8 would be bracketed

5 by the age dating s of 111 ± 12 ka and 123 ± 16 ka. E8 would therefore slightly older than E6 and E7. This would also imply that E8 might be older than the oldest deposits in SDF1 and therefore not visible there. Unfortunately, the magnitude of this possible seismic event cannot be constrained by trench observations.

**6 Seismotectonic implications**

**6.1 Recurrence intervals for earthquakes with magnitudes larger than 6.5 along the MF**

10 The possible correlations of paleoearthquakes between the trenches allow for two different interpretations to reconstruct the recurrence intervals of earthquakes with magnitudes larger than M = 6.5. The event E1 will be excluded during the following considerations, since the related magnitude estimate is lower than those obtained for E2-E7. It seems that the colluvial wedges associated with the larger earthquakes conceal or even erase evidences for offsets formed by smaller earthquake. The displacement of markers related to E1 is only conserved because the event happened after the last earthquake that caused a

15 colluvial wedge to form (E2). Any future event with a surface displacement that is large enough to lead to the erosion of the offset markers in the footwall will destroy the evidence for E1. This restriction also applies to earthquakes with small surface offset that occurred prior to E2. Therefore, earthquake records for magnitudes less than about 6.5 are most probably incomplete, and thus excluded from the recurrence calculation.

As mentioned in sect. 5, the crucial part for the reconstruction of recurrence intervals therefore is whether A4 is correlated

20 either to B3 or to B4 and, subsequently, whether A5 is correlated to B4 or to B5, resulting in the following event lines:

(1) E2-E3(not correlated to B3)-E4-E6-E8 (5 earthquakes);

(2) E2-E3(correlated to B3)-E5-E7 (4 earthquakes).

The determination of inter-event intervals is based on the limits for the occurrence time intervals for each earthquake as given in Table 2. Figure 10 shows clearly that both event lines represent different types of distributing earthquakes.

Event line 1 ly periodic reoccurrence of earthquakes with magnitudes larger than M = 6.5. The maximum time interval between E2 and E3 is 15.8 ka. As the occurrence time of E3 limits the occurrence time interval of E2, the minimum time interval cannot be calculated. Considering  line 1 (E2-E3-E4-E6-E8) and the range of uncertainties related to dating, the inter-event time between E3 and E4 lies between 6.3 ka and 41.5 ka, while the inter-event time between E4 and E6 is between 20.2 ka and 65.1 ka. Finally, the maximal inter-event time between E6 and E8 is constrained to 40 ka.

30 Similar to the inter-event time for E2/E3, a minimum inter-event time for E6/ E8 cannot be calculated. Taking all information

together, the average of the minimum values and the average of the maximum recurrence intervals for event line 1 would be then ~ 13 ka and ~ 40 ka, respectively.

On the other side, earthquakes in event line 2 (E2-E3-E5-E7)  in time. Therefore, instead of calculating inter-event times for all earthquakes, we calculate the minimum inter-cluster time that is identical with the inter-event time for
5    E3/E5, and the maximum intra-cluster times for E2/E3 and E5/E7, meaning the largest possible time between both earthquakes within the same cluster. A maximal inter-cluster time cannot be given, due to the poor time constraint within the both clusters. However, the maximal intra-cluster times for E2/E3 and E5/E7 are 15.8 ka and 17.0 ka, respectively. In addition, the minimum inter-cluster time interval between E2/E3 and E5/E7 is at least 54.4 ka. The time since the occurrence of E2 until today may be also considered as a minimum inter-cluster time, being at least $32.9 \pm 4.1$ ka and maximal $40.9 \pm 3.6$ ka. Another estimation
10   of the minimum elapse time between clusters can be estimated from the oldest layers in trench SDF3 (unit 8) dated to $158 \pm 21$ ka (sect. 3.2). Since there is no older record than B5, it is reasonable to assume that there was no earthquake during the time between B5 and the oldest unit 8 exposed in SDF3. Therefore, the minimum time elapsed between B5 (=E7) and any older cluster must be at least $42 \pm 21$ ka.

**6.1 Comparison of long-term Quaternary slip rates with paleoseismological slip rates**

15   Long-term Quaternary slip rates along the MF can be inferred  morphological scarp height of  17 m and the age of the top of the Gaenserndorf terrace (~ 200 ka, Weissl et al., 2017). Using the present-day scarp height as minimum displacement since the abandonment of the terrace 200 ka ago, a minimum slip rate of 0.085 mm/a  In addition, the base of the Quaternary gravels, which is equivalent to the top of Neogene sediments, is offset by approximately 40 m (Figure 3). Assuming  Neogene-Quaternary boundary (2.6 Ma), the slip rate along the
20   MF  is a minimum estimate since age data from the thick Quaternary sediments in the hangingwall of the MF are not available. Figure 11 shows the range of possible slip rates for both  falling in between the bracket of the geomorphic slip rates,  reasonable agreement.

**6.3 Comparison of magnitude estimates with fault rupture area and length**

For faults with known fault geometry, empiric relations allow to evaluate the maximum magnitude that a fault can produce
25   from rupture length and area. The surface expression of the MF is only recognizable for about 10 km along the eastern margin of the Pleistocene Gaenserdorf terrace (Figure 3A). Further to the south, the Danube has erased  geomorphic expression in its Holocene flood plain. However, the geometry and the length of the MF are well known thanks to the distribution of Quaternary sediments in the hangingwall of the fault and 2D/3D reflection seismic within the central Vienna Basin (Hölzel et al., 2010, Hinsch and Decker, 2011, Salcher et al., 2012, Spahic et al., 2013). hese data, the length of the MF
30    25 km (Salcher et al., 2012). In addition, Hinsch and Decker (20— constructed) a generalized detachment for the Vienna Basin. Beneath the MF, the detachment is assumed to be at the depth of about 10 km (Wessely et

al., 2006). Taking into account the general dip of 55° for the MF observed in seismic, the rupture area of the MF would amount to 315 km², leading to a maximal credible magnitude of 6.5 ± 0.3).

However, in case that the MF is indeed linked to the VBTF via the common detachment as proposed by Beidinger and Decker (2011), the area of the detachment between the MF and VBTF might be also activated during large events (Figure 12). The

5   total fault surface activated during such events is derived as the sum of the fault surface of the MF and the portion of the basal
detachment between the MF and the VBTF, which has a size of about 130 km². The fault length of the MF in this tectonic scenario is 36 km and the total fault area amounts to about 580 km². These fault parameters correspond to a maximal credible magnitude of 6.7 ± 0.3) using the relationships by Wells and Coppersmith (1994). This is in good agreement to the magnitude estimations derived from the trenches.

10  **7 Conclusions and implications for seismic hazard assessment**

In this study, we   splay normal fault of the VBTF that previously has not been considered as a source for seismic hazards. We show evidence that the fault  at least 5-6 strong earthquakes with magnitudes larger than 6.2 in the last 120 ka. The magnitude of the earthquake with the largest surface displacement is evaluated with 6.8 ± 0.1). This value compares well with the maximum magnitude of 6.7 ± 0.3) estimated from the potential

15  rupture area of the MF. The fault area is about 580 km² when including the detachment that links the normal fault with the VBTF. The vertical slip velocity of 0.03 to 0.04 mm/a derived from trench observations  geomorphologically determined vertical slip rates for the MF, which range from 0.085 to 0.015 mm/a.

Trench observations and uncertainties of OSL/IRSL age dating do not allow for an unequivocal conclusion of earthquake recurrence rates. Both earthquake scenarios  presented here are possible considering the available time

20  constraints.  1, however, appears less likely as it seems improbable that an earthquake with a magnitude around 6.6 has not been recorded in trench SDF3, while producing a surface displacement of 80-90 cm in trench SDF1 at a distance of less than 2 km. For us, the more plausible correlation between the trenches is therefore  2, suggesting that earthquakes with magnitudes larger than 6.5 cluster in time. This may have consequences for the application of the reconstructed recurrence intervals in seismic hazard assessments, e.g., by using cluster recurrence intervals rather than average single event recurrence

25  intervals.

Trench evidence for the youngest event E1 (magnitude 6.2 ± 0.2) further shows that strong earthquakes with magnitudes less than 6.5 also occur outside of the suggested clusters. Unfortunately, the recurrence intervals of such events cannot be constrained by trenching results. The sedimentary and structural records of events with surface displacements, which are too small to produce colluvial wedges, may be masked or even erased by subsequent larger earthquakes that lead to the erosion

30  and redeposition of material into colluvial wedges. Therefore, earthquake records for magnitudes less than about 6.5 are most probably incomplete at the MF.

[Figure]

The issues discussed above lead us to conclude the following main implications for seismic hazard assessment in the Vienna Basin:

1. The paleoseismological results  is seismically active and  to be considered as a seismogenic source in seismic hazard assessment. Earthquakes with magnitudes larger than about 6.5 occur at average recurrence times of about 25 ka ( 1),  ( 2; Figures 10 and 11). The frequency of surface-breaking earthquakes with magnitudes less than about 6.5 cannot be constrained by trenching due to the low preservation potential of such earthquake records.

2. Data from the MF provides evidence that the maximum credible earthquakes in the Vienna Basin should . This value is significantly higher than previous estimates of Mmax = 6.0 to 6.5 (Lenhardt et al., 1995; Procházková and Šimunek, 1998; Sefara et al., 1998; Tóth et al., 2006). The data presented in our study was used in the SHARE project to incorporate the MF in its active fault database and hazard calculation (Basili et al., 2013).

3. The MF is kinematically and geologically  to a number of other splay normal faults of the VBTF close to the Austrian capital, Vienna (Figure 9; Beidinger and Decker, 2011). It  that these faults are potential sources of large earthquakes as well. However, except for the Aderklaa-Bockfliess faults (Weissl et al., 2017), no paleoseismic characterisation of these faults . The frequency of strong earthquakes near Vienna is therefore expected to be  higher than the earthquake frequency reconstructed for the MF.

4. The magnitude of the largest earthquake recorded at the MF (6.8 ± 0.1) is regarded to support the assumption of a listric fault and an active basal detachment that links the normal fault with the VBTF strike-slip system.  fault geometry  ground motion  to earthquakes that activate large parts of the listric fault with ground motion expected to be more severe in the hanging wall direction, than in the footwall direction (
[revised manuscript text omitted]

|---------|----------|------------------------|------------------------|--------------|
| A1 | displ | - | - | 0.15 - 0.25 m |
| A2 | cw, tc | 0.95 m | 0.75 m | 1.50 - 1.90 m |
| A3 | cw, tc | 0.45 m | 0.40 m | 0.80 - 0.90 m |
| A4 | cw, tc | 0.75 m | 0.72 m | 1.40 - 1.50 m |
| A5 | cw | 0.25 m | 0.40 m | 0.50 - 0.80 m |
| B1 | displ | - | - | 0.10 - 0.15 m |
| B2 | cw | | 0.8 | 0.8/1.6 m |
| B3 | tc | - | - | - |
| B4 | cw | - | - | - |
| B5 | cw | - | - | - |
| C1 | displ | - | - | 0.17 - 0.20 m |
| C2 | cw | - | - | - |

Table 1: Type of evidence and inferred displacement for the paleoearthquakes A1 to A5 (trench SDF1), B1 to B5 (SDF3), and C1 to C2 (WAG). Also listed are the thicknesses of colluvial wedges observed in the NW and SE trench walls used for estimating displacement. Evidence: displacement of correlated layers (displ.), occurrence of colluvial wedges (cw), and sediment-filled tension cracks below the colluvial wedges (tc). Displacement is taken as twice the thickness of the colluvial wedge.

| Sample | Location | Method | De (Gy) | $D_0$ (Gy/ka) | Depth (m) | Water (%) | Age (ka) |
|--------|----------|--------|---------|---------------|-----------|-----------|----------|
| AIP25 | WAG cover | IRSL | 44.7 ± 2.3 | 2.78 ± 0.26 | 1.0 | 12 | 16.1 ± 1.7 |
| AIP26 | WAG cover | IRSL | 39.0 ± 1.4 | 2.01 ± 0.16 | 0.5 | 12 | 15.1 ± 1.5 |
| AIP38 | SDF1 fw | IRSL | 543.1 ± 33.7 | 2.13 ± 0.20 | 3.7 | 10 | 255 ± 29 |
| AIP39 | SDF1 hw | IRSL | 273.5 ± 20.2 | 2.64 ± 0.25 | 3.6 | 10 | 104 ± 12 |
| AIP40 | SDF1 hw | IRSL | 154.0 ± 5.5 | 2.72 ± 0.25 | 3.3 | 10 | 56.6 ± 5.7 |
| AIP41 | SDF1 hw | IRSL | 168.7 ± 5.9 | 3.45 ± 0.32 | 2.6 | 10 | 48.9 ± 4.8 |
| AIP44 | SDF1 hw | IRSL | 36.7 ± 2.1 | 2.26 ± 0.22 | 1.9 | 10 | 16.3 ± 1.8 |
| AIP46 | SDF1 hw | IRSL | 43.6 ± 1.6 | 2.01 ± 0.16 | 1.2 | 10 | 13.8 ± 1.4 |
| AIP93 | SDF3 hw | IRSL | 419.2 ± 39.4 | 3.17 ± 0.30 | 4.1 | 12 | 158 ± 21 |
| AIP95 | SDF3 hw | IRSL | 317.4 ± 29.1 | 2.58 ± 0.24 | 3.1 | 12 | 123 ± 16 |
| AIP97 | SDF3 hw | IRSL | 205.1 ± 13.3 | 2.90 ± 0.26 | 2.1 | 12 | 70.8 ± 8.0 |
| AIP98 | SDF3 hw | IRSL | 91.6 ± 7.5 | 2.78 ± 0.25 | 1.8 | 12 | 32.9 ± 4.1 |
| AIP102 | SDF3 hw | IRSL | 15.7 ± 0.9 | 3.29 ± 0.30 | 0.3 | 15 | 4.8 ± 0.5 |
| AIP103 | SDF3 fw | IRSL | 384.6 ± 59.7 | 1.88 ± 0.18 | 1.4 | 10 | 205 ± 37 |
| AIP114 | SDF3 fw | IRSL | 468.3 ± 43.4 | 1.81 ± 0.17 | 0.65 | 10 | 259 ± 35 |

Table 2: Infrared stimulated Luminescence (IRSL) and optically stimulated luminescence (OSL) dating results from the trenches SDF1, SDF3, and WAG at the Markgrafneusiedl Fault (MF). Location: refers to either of the trenches (SDF1, SDF3, WAG) and the location in respect to the MF, where hw = hanging wall and fw = footwall, De (Gy): equivalent dose in Gray (Gy), $D_0$ (Gy/ka): dose rate in Gray values (per 1.000 years); Depth (m): depth of the sampling location in meters below present-day surface.

[Figure]

[Figure]

| Event # | Correlation | *Antequem*
Event older than
(age / sample location) | *Postquem*
Event younger than
(age / sample location) |
|---|---|---|---|
| E1 | A1 = B1 = C1 | **13.8 ± 1.4 ka (SDF1)**,
4.8 ± 0.5 ka (SDF3) | 32.9 ± 4.1 ka (SDF3),
16.3 ± 1.8 ka (SDF1),
**15.1 ± 1.5 ka (WAG)** |
| E2 | A2 = B2 = C2 | **32.9 ± 4.1 ka (SDF3)**,
16.1 ± 1.7 ka (SDF1) | 70.8 ± 8.0 ka (SDF3),
**48.9 ± 4.8 ka (SDF1)** |
| E3 | A3, ?= B3? | **32.9 ± 4.1 ka (SDF3)**,
16.1 ± 1.7 ka (SDF1) | 70.8 ± 8.0 ka (SDF3),
**48.9 ± 4.8 ka (SDF1)** |
| E4 | A4, ?= B3? | **56.6 ± 5.7 ka (SDF1)**
32.9 ± 4.1 ka (SDF3), | 104 ± 12 ka (SDF1),
**70.8 ± 8.0 ka (SDF3)** |
| E5 | ?A4 =? B4 | **111 ± 12 ka (SDF3)**,
56.6 ± 5.7 ka (SDF1) | 123 ± 16 ka (SDF3),
**104 ± 12 ka (SDF1)** |
| E6 | A5 ?= B4? | **111 ± 12 ka (SDF3)**,
56.6 ± 5.7 ka (SDF1) | 123 ± 16 ka (SDF3),
**104 ± 12 ka (SDF1)** |
| E7 | ?A5 = B5? | **111 ± 12 ka (SDF3)**,
56.6 ± 5.7 ka (SDF1) | 123 ± 16 ka (SDF3),
**104 ± 12 ka (SDF1)** |
| E8 | B5 | **111 ± 12 ka (SDF3)** | **123 ± 16 ka (SDF3)** |

**Table 3: Overview of common IRSL constraint for each possible earthquake derived from all different sites. Ages in bold mark the upper and lower limit for each occurrence time. For details about correlation between the trenches, see sect. 5.**

[Figure]

**Figure captions**

Figure 1: A) Active faults (black solid and dashed lines), seismicity (black circles) and Quaternary basins (light grey areas) within the Vienna Basin (Austria) plotted on a shaded DEM. The borders of the Austrian capital, Vienna, is outlined by a dashed white line. Modified after Beidinger and Decker (2011). White box shows the location of the close up in Figure 2; (B) Major earthquakes

5  from historical, instrumental and paleoseismological data in intra-plate central Europe. Historical and instrumental seismicity is based on the CENEC Catalogue by Grünthal et al., 2009. Paleosites are compiled from Camelbeeck and Meghraoui, 1998; Camelbeeck et al., 2000; 2007; Meghraoui et al., 2001; Vanneste and Verbeeck, 2001; van den Berg et al., 2002, Peters, et al., 2005; Štěpančíková et al., 2010. Labels indicate the magnitudes of the largest paleoearthquakes observed at the respective site. Black box shows area of the close up in (A). MF= Markgrafneusiedl Fault; VBTF = Vienna Basin Fault System.

10  Figure 2: Cross section through the Vienna Basin at its central part based on reflection seismic and deep boreholes indicating the common detachment of the Alpine floor thrust, which links the splay normal faults to the Vienna Basin Transfer Fault (VBTF). Redrawn from Hölzel et al. (2010).

Figure 3: Overview of the MF. (A) DEM the Pleistocene terraces north of the Danube dissected by faults creating fault scarps (fs). Dashed line: trace of the topographic profile in B, solid line: trace of the seismic line in C. (B) Topographic profile (black) and cross-
15  section indicating the base of Quaternary sediments (grey) across the MF. Note the thickness of Quaternary growth strata in the fault-delimited basin above the MF. (C) seismic section across the same area showing offset along the Markgrafneusiedl Fault (MF) and the flower structure at the Vienna Basin Transfer Fault (right). See text for details.

Figure 4: Photo mosaic and interpretation of the SW-facing wall of the trench SDF1 across the Markgrafneusiedl Fault (for location
20  see Figure 2). Colluvial wedges and underlying tension cracks related to earthquakes A2-A5 are numbered. The displacement related to A1 is marked. Numbers indicate the age and the location of IRSL and OSL samples. See text for further explanation.

Figure 5: Details from the SW-facing wall of the trench SDF1. (A) Evidence for earthquake A1 from the displacement of a marker horizon (white arrows), which is correlated across the fault (red arrows). (b) Evidence for earthquake A2 from a colluvial wedge composed of sandy gravel overlying a tension gash filled with the same material (white arrows). To the right the wedge abuts against
25  fault 1 (red arrows). (C) Colluvial wedge associated with earthquake A3 (white arrows) overlying a tension gash adjacent to fault 2. Several deformation bands that branch from fault 2 and formed during a later earthquake cut the wedge. It overlies wedge 4, which equally contains reddish redeposited soil. Wedge 4 shows an erosional contact to grey high-stage flood sediments (around box E). (D) Deformation bands offsetting laminated fluvial sand (red arrows) above wedge 2. The deformation bands are correlated to the event horizon of E1 (detail of picture B). (E) Detail of (C). Erosional contact of wedge 4 to flood sediments. Armoured mudballs
30  (arrow) derive from the eroded colluvium.

Figure 6: Photo mosaic and interpretation of the SW-facing wall of the trench SDF3 across the Markgrafneusiedl Fault (for location see Figure 2). Colluvial wedges and underlying tension cracks related to earthquakes B2-B5 are numbered. The displacement related to B1 is marked. Numbers indicate the age and the location of IRSL and OSL samples. Additional information is provided in the
35  text.

[Figure]

**Figure 7: Details from the SW-facing wall of the trench SDF3. (A) Wedge associated with earthquake B2 (white arrows; see text for discussion) overlying wide fault (red arrows). The upward widening fault is recognized from pebbles, which are oriented parallel to the fault. The top of the wedge (white arrows) is offset by a narrow deformation band that emerges from the fault below the wedge (purple arrows). Offset occurred during B1. (B, C) Laminated flood sediments (clay, silt and fine sand) underlying colluvium of wedge B2. Pebbles sunken into the soft sediment (B) and flame structures protruding into the overlying gravel (C) are indicative for liquefaction.**

**Figure 8: (A) Trench WAG, photo mosaic of the SW-facing trench wall. Red arrows denote locations of faults, white arrows point to offset contact between colluvium and overlying loess. Boxes refer to details shown in Figure 9 B and C.**

**Figure 9: (A) Trench WAG looking E toward the footwall of the MF (fault trace denoted by red arrows). Note offset of bright layer of loess (white arrows) corresponding to C1. CW denotes the colluvial wedge related to earthquake C2. (b) Detail of the SW-facing trench wall. Red arrows denote locations of faults, white arrows point to the offset contact between grey and brown silt and clay. Box shows location of details shown in D. (C) Offset of the top of the colluvial wedge associated with earthquake C2 (white arrows). (D, E) Fractured and sheared pebbles indicating normal displacement parallel to the slip of the MF. Note that fractures in pebbles are filled with sandy matrix excluding fracture formation during construction work.**

**Figure 10: Comparison of age constraints from all trench sites SDF1, SDF3, and WAG and possible occurrence times of the observed earthquakes for the two possible correlations.**

**Figure 11: Comparison of surface slip rates for the Markgrafneusiedl Fault (MF) from geomorphic constraints and from trench results. On the left the constraints for event line 1 are plotted, on the right, those for event line 2 are shown.**

**Figure 12: Geometry and fault area of the Markgrafneusiedl Fault (MF). Also shown are the Vienna Basin Transfer Fault (VBTF) and other active normal splay faults branching from the VBTF. ABF: Adreklaa-Bockfliess fault system; BNF: Bisamberg-Nussdorf fault; LF: Leopoldsdorf fault; SF: Seyring fault (redrawn from Decker et al., 2015). Broken grey line marks the city limits of Vienna.**

[Figure]

[Figure]

**Figures**

**Figure 1**

[Figure]

[Figure]

[Figure]

[Figure]

**Figure 2**

[Figure]

[Figure]

**Figure 3**

[Figure]

[Figure]

[Figure]

Figure 4

[Figure]

[Figure]

Figure 5

[Figure]

[Figure]

[Figure]

Figure 6

[Figure]

[Figure]

**Figure 7**

[Figure]

[Figure]

[Figure]

**Figure 8**

[Figure]

[Figure]

**Figure 9**

[Figure]

[Figure]

**Figure 10**

[Figure]

[Figure]

**Figure 11**

[Figure]

**Figure 12**

---

## Referee Comment (RC2) · D. Clark (Referee) · 10 Jul 2017

dan.clark@ga.gov.au Received and published: 10 July 2017
The authors present new paleoseismic data for three sites along the Markgrafneusiedl Fault in the Vienna Basin, Austria. This is one of several normal fault splays within

a releasing bend along the Vienna Basin Transfer Fault (VBTF). Evidence for seismic disturbance at each site is thoroughly and meticulously documented, demonstrating a definite seismic hazard. The data are then combined to derive two possible event chronologies for M $\sim$ 6.2-6.8 earthquakes on the MF involving 5-6 events over the past  $\sim$ 120 ka. The finding that multiple large earthquakes have occurred within the Vienna Basin has the potential to constrain future seismic hazard assessments, and so is of great interest to researchers constructing seismic hazard models for the region, and emergency managers. Researchers of neotectonics more generally will also find the results interesting.

However, I have concerns with the method by which events were correlated between trenches. For paleoseismological investigations with the richness of data that this one possesses, it is common practise to estimate event ages and their uncertainties using a probabilistic framework such as OxCAL (Lienkaemper & Ramsey, 2009). Without such a rigorous framework, the confidence that can be placed in the event chronology derived, and the slip models proposed, is significantly diminished. For example, such a treatment may invalidate or provide support for either a periodic or a clustered slip model, significantly simplifying the discussion. Given the complex linkages between splay faults and the VBTF, it is intuitive to suspect some fault interaction that might lead to clustering behaviour. This possibility could be more fully explored if a re-analysis supported it as a probable mechanism.

Following revision, I think the manuscript will be an important contribution to the seismic hazard community and might drive greater awareness of the potential seismic hazard in the Vienna Basin relating to the splay faults of the VBTF (and to the VBTF itself). As such the study is well suited for publication in NHESS.

**GENERAL COMMENTS:**

ć The labelling of 'events' (e.g. A1-A5, B1-B5 etc) on the trench logs is a bit confusing. Consider labelling the colluvial units (in your unit notation), or event horizons. aĂć Combination of age data between trenches (Section 5). Section 5 is perhaps more complex/convoluted than it needs to be. For paleoseismological investigations with the richness of data that this one possesses, it is common practise to estimate event ages between trenches, and their uncertainties, using a probabilistic framework such as Ox-CAL (Lienkaemper & Ramsey, 2009). The people working on the Wasatch Fault in Utah have taken this to the next level (DuRoss et al., 2016; Personius et al., 2012). A rigorous analysis of this kind will lead to a clearer understanding of what range of event timings are possible within the uncertainties of your data. The conclusions regarding recurrence interval and slip model (periodic or clustered) may then be more boldly stated. ć Recurrence model. It's hard to get a good understanding of whether a periodic (with a calculated coefficient of variation) or clustered recurrence model might be more appropriate to describe rupture on the MF until the above analysis is completed. However, Figures 1 and 2 present some possibilities worth consideration/discussion. The figures imply that all of the faults shown on Figure 1 are connected, either at the surface, or at depth. Excellent potential exists for fault interaction throughout the slip history of individual faults, and stress triggering between faults in individual ruptures. Figure 1 shows concentrations of epicentres where the normal splays branch from the VBTF. The first question is then how do events on the MF relate to events on the Vienna Basin Transfer Fault? Does any data exist (or could an average recurrence on the VBTF be calculated using its slip rate)? Does rupture on the VBTF trigger rupture on the splays, do the splay faults rupture individually or with only a small segment of the VBTF, bound by the intersections? A potentially much larger rupture area than you have considered bit result from such an interaction (for example, the smaller displacement of your most recent event could relate to rupture 'leaking' from a VBTF event). Does sharing of slip between the splay faults result in what appears to be a clustered slip history for the MF when considered in isolation?

SPECIFIC COMMENTS:

Page 2, line 23: there seems to be interchanging between the terms 'periodic be-

СЗ

haviour' and 'characteristic behaviour'. They are not alike. A fault characterised by a periodic slip distribution (or a clustered slip distribution for that matter) need not rupture characteristically. Page 4, line 16: "Whether this apparently slowly moving fault can produce larger earthquakes or it is aseismically creeping, is the key question of our study". The potential for creep has not been discussed. What light does the observations in the trenches shed on this question?

Page 4, lines 23-30: It would be helpful to provide some contextual detail of the trench sites for the reader not familiar with Vienna Basin stratigraphy and fluvial evolution. In particular, a few words regarding Gaenserndorf terrace. It would also help set context to provide figures showing the geomorphology of the trenching sites. Perhaps these cold be provided in supplemental information? Also, the mentioned towns are not marked on the Figure 3.

Page 12 line 10: Your displacement estimates for the most recent event relate to just the displacement across the active fault trace. In each case the far field displacement may be much more (e.g. the vertical separation of the red horizon in Figure 4). How do you explain this? Does it relate to pre-existing topography, is there a near surface slip deficit on the fault, or may there be afterslip, or interseismic creep on the fault?

Page 14, line 13: "It seems that the colluvial wedges associated with the larger earthquakes conceal or even erase evidences for offsets formed by smaller earthquake". A related question here is what is the threshold for surface rupture and the threshold for discoverability of a surface rupture in this area? An interesting article where thresholds have been assessed is found here: http://gfzpublic.gfz-potsdam.de/pubman/faces/viewItemOverviewPage.jsp?itemId=escidoc:691901:3

Page 14, line 25: The aperiodicitiy of the 'event line 1' (and 'event line 2') could be quantified by calculating a coefficient of variation.

Page 14, line 7: or they don't break the surface.

Page 17, line 8: I would suggest that you express your results in terms of minimum magnitude events based upon your displacements. e.g. the Mmax should be considered to be at least X. This accounts for the potential that the events you see on the MF might be part of a larger, mainly strike slip, rupture on the VBTF.

Page 17, point 3. The splay to the southeast of the MF appears to have a larger Quaternary throw. Does this imply that it is more active than the MF?

Page 17, line 15. This is the first mention of data for the Aderklaa-Bockfliess fault. What is this data and how can it be interpreted in terms of fault interaction/clustering etc?

**TABLES/FIGURES**

Table 2: are the stated uncertainties one sigma or otherwise? Figure 1: Parts A and B are not marked on the Figure. "PDZ" is not explained. ACORN (2004) in the legend is not in the reference list. - It is interesting that there are concentrations of historic epicentres apparent where the mapped releasing bend normal faults splay off the main VBTF trend. I wonder if this association could be used for a proxy to assess activity on each splay, or segmentation behaviour? - As for Figure 12, please mark the names of the other faults. Figure 2: Location of this cross section should be marked on Figure 1. Figure says "for location see Figure 2". MF and VBTF should be marked for clarity. What does NCA stand for? What is the white material uppermost in the section? Figure 3: It is not easy to reconcile the fault scarps that are marked in the inset box on Figure 1 with the scarps shown on Figure 3A. At least mark the features that are shown on Fig. 1 on Fig. 3A. Do parts B and C have a vertical exaggeration? Figure 4: It is good practice to present interpreted and uninterpreted trench photomosaics (or an uninterpreted photomosaic and an interpretation with patterned fill) side by side for comparison. Using line work in the interpreted version would assist with developing and explaining the interpretation. For example, the uppermost (darker) unit thickens significantly across the F2 fault trace. Could this be interpreted to mean that although the discrete fault

displacement relating to the A1 event is small, the far field displacement was significantly more (and taken up by distributed deformation). It's a bit confusing that 'events' are labelled on the trench wall, rather than units, or event horizons. Perhaps mention in the figure caption that the 1 m square trench grid is lettered on the vertical axis and numbered on the horizontal axis (this would make it easier for the reader to orient on Figure 5). The scale bar in the figure seems to be twice the size as the grid. Figure 5: Most of the parts of this figure are not cited in the text. Are they necessary? Figure 6: present uninterpreted and interpreted parts as per suggestion for Figure 4. Figure 7: the parts of this figure are not cited in the text. Figure 9A: there should be consistency between Figures 8 and 9A as to which horizons and faults are indicated with the arrow heads. Figure 10: While it is good to see all the data in one figure, there are perhaps more rigorous ways to analyse event timing. Consider developing an Oxcal model, and combining event probability density functions. Figure 12: perhaps these faults could be labelled on Figure 1 also?

**SUPPLEMENTAL MATERIAL**

It would be valuable to include detailed site maps for the trench locations to support your interpretations of site geomorphology. There is a big jump from the scale of Figure 3 to the trench log scale.

Note that an annotated version of the manuscript has been provided with grammatical etc corrections suggested.

**REFERENCES CITED**

DuRoss, C.B., Personius, S.F., Crone, A.J., Olig, S.S., Hylland, M.D., Lund, W.R. & Schwartz, D.P. 2016. Fault segmentation: New concepts from the Wasatch Fault Zone, Utah, USA. Journal of Geophysical Research: Solid Earth, 121: 1131-1157, doi:10.1002/2015JB012519. Lienkaemper, J.J. & Ramsey, C.B. 2009. Versatile Tool for Developing Paleoearthquake Chronologies - A Primer. Seismological Research Letters, 80(3): 431-434, doi: 10.1785/gssrl.80.3.431. Personius, S.F., DuRoss, C.B.

& Crone, A.J. 2012. Holocene Behavior of the Brigham City Segment: Implications for Forecasting the Next Large-Magnitude Earthquake on the Wasatch Fault Zone, Utah. Buletin of the Seismological Society of America, 102(6): 2265-2281, doi: 10.1785/0120110214.

---

## Referee Comment (RC3) · D. Clark (Referee) · 10 Jul 2017

The comment was uploaded in the form of a supplement:
https://www.nat-hazards-earth-syst-sci-discuss.net/nhess-2017-126/nhess-2017-126-RC3-supplement.pdf

---

## Referee Comment (RC4) · M. Ortuño (Referee) · 13 Jul 2017

The paper presents the analysis of three trenches studied along the Markgrafneusiedl fault, a normal fault associated to the Vienna Basin Transfer Fault. Slip history and timing of paleo ruptures is performed through a paleosiemsological analysis incorporating OSL and IRSL dating.

I think the paper could be suitable for NHESS after moderate revisions are undertaken. Evidence of paleo-earthquake are based on cumulative displacements and on

the presence of colluvial wedges. These wedges should be better described and discussed; gelifluction processes seem, in my opinion, could be the responsible of the wedge-shaped geometry of some of them (which do not necessary are tectonic colluvial wedges).

Given the implications on seismic hazards, three clue issues should be better discussed: 1) periodic vs clustered behavior; 2) primary vs secondary ruptures; 3) relation of EQ chronology with glacial retreat.

I encourage the authors to address these points in order to turn this manuscript in an improved presentation and discussion of their excellent findings.

Please fill free to contact me if my comments are not clear,

Sincerely

Maria Ortuño (Univ. of Barcelona)

13th July 2017 Paleoseismological data:

-Well and comprehensive presented paleo-seismological data. However, as a paper reporting paleoseismological data, I miss:

1. A general sketch of the sites, with the geomorphological features is missing. 2. Some pictures of the landscape (at least one) would help the reader to understand better the setting. 3. The logs (Fig. 4, 6 and 8) need to show subunits discussed in the text. Photologs as supplementary material would help a better understanding of the descriptions. Often, deformational features are referred in the text but cannot be identified in the logs or in the pictures. Make sure you indicate/locate them. In general, I miss more references to the already provided pictures of the trench walls (Fig. 5 and 7). 4. Event horizons should be included in the logs as lines, or at least points. Only letters are insufficient to exactly locate the stratigraphic position of the event. Within the text, the events should be described as defined by bracketing units (upper/lower). That is a unit-constrain of events, usually present in paleoseismic studies. This is different

than the time constrain provided in section 5. If in the future, units are re-dated (which is quite common), the definition of events can be kept, still is valid. 5. The internal structure of the sections describing the trenches should be parallel. For instance, colluvial wedges in SDF1 are described together with the stratigraphic unit, but in SDF3 they are not mentioned until the section discussing the events. 6. "deformation bands". I might be wrong, but in most papers dealing with fault exposure, those bands are simply called fault zone banding of foliation. 7. The section analyzing the events in trench WAG is missing. Expected section 3.3.1.

There are two parts of the text where I suggest alternative interpretations to the deformational features observed in trench SDF1 (where colluvial wedges should be discussed better. The role of geli-fluxion in the formation of "tailed wedges" should be considered, in my opinion.) and in trench WAG (where the structure of "deformational bands" resembles a fold-scarp, not a foliated fault zone). See specific comments to these issues below.

The trenches always show very distinctive materials in the hanging compared to the footwall. Do you think the faults controlled the sedimentation in the hanginwall in most cases? o.e., they acted as physical barriers hampering the sedimentation in the footwall (with some exceptions). I think it is interesting to briefly discuss this subject.

Dating results.

-This part is quite methodological, but I found it interesting. Perhaps, this could be included as an appendex, because it "breaks the flow" of the manuscript. I would only leave in the main text the Dating results.

Paleoseismological discussion.

I found four main problems/issues (commented below in the comments to sections): 1) The definition of the events (particular and common events) should be first done in terms of bracketing units. Then, age constrains can be exposed (based on limited

number of samples, as usually). Figure 10 should be improved. 2) Maximum expected magnitudes. I suggest to compare values with those derived for surface ruptures, and move all those reference to section 6. In the last years, the Mmax derived from observed displacement is not well accepted (as far as I understand, I give some references). Event displacement might be highly variable along the trace. Wells and Coopersmith might not be representative for slow faults in continental settings. 3) I have done some comments in the Discussion of the periodic/aperiodic behavior (see below). I don't think any of the two proposed scenarios lead to infer periodic behavior. 4) The possibility of the MF being a secondary fault of the VBTF should be discussed. It is quite relevant for hazard estimates to consider these two as primary Eqs sources. Is this paper providing robust data in a primary seismogenic nature of the MF?

Please also note the supplement to this comment:
https://www.nat-hazards-earth-syst-sci-discuss.net/nhess-2017-126/nhess-2017-126-RC4-supplement.pdf

**Supplement:**

Specific Comments to NHESS MS
"Implications from
palaeoseismological investigations at the
Markgrafneusiedl Fault (Vienna Basin, Austria) for
seismic hazard assessment" by
Esther Hintersberger et al.

By: María Ortuño (Univ. of Barcelona)
13th July 2017

COMMENTS:

In general, well written. The writing is concise and clear. Well organized as well. It could be improved in Topic sentences are included at the begining of some sections, providing a summary of what it is coming. For instance, section 3.1.1.

English grammar and usage. I found is pretty correct but I am not really good in english. I detect some minor errors which I marked in the commented MS pdf.

Structural data are reported following the convention (dip direction/dip: 116/69). I suggest to specify this the first time one fault plane orientation is given (henceforth dip direction/dip, in this case: 116/69) and do not include the explanation in the following text, just include the values.

ABSTRACT,

1) you state that:

*Trench observations*
*also show that structural and sedimentological records of strong earthquakes with small surface offset have only low conservation potential.*

It is really showed and discussed in the text? Because as an idea, it's quite reasonable, expected. But perhaps not derived from the data discussed here.

2) you state that:

Magnitude estimates from fault dimensions suggest that the largest earthquakes observed in the trenches activated the entire fault surface of the MF including the basal detachment that links the normal fault with the VBTF.

I don't see how the Magnitude estimates can tell about the rupture lenght by themselves. The M is estimated from length, so cannot be indicative of it.

GEOLOGICAL SETTING

- Historical seismicity. Even if some moderate EQ (not rupturing the surface) nucleates in the fault splay, its location would be rough. So, with the uncertainities associated to the epicentral locations, some of the historical EQ assigned to the VBTF could have been produced by the splay faults, don't you think? They are too close, compared to epicentral

errors of pre-instrumental Eqs.

– At the end of section 2.2, you suggest that someone can think of the MF as a creeping fault. But some sentences above, you talk about small earthquakes. I would rather highlight that there are small Eqs associated to the fault (so, it is seismically visible). Include references, even if no-one did that correlation between small Eqs and the MF, just the data base of those M< 1.0 Eqs.

– Is this fault moving along? For a reader from abroad, a short summary of the number of active faults characterized by paleoseismology in the region is necessary, also to appreciate how these data are unique.

**TRENCHING RESULTS**

**SDF1**

line 28 (page 4). Please follow an easier structure. The trenches expose: 1) a fluvial terrace (description is lacking) in the MF footwall; 2) a different fluvial terrace or flood deposits (??) (include here your description) in the MF hanging wall.

SDF1 trench.
-You mention a "dry valley". It is an intermittent creek?? Please provide a geomorpfological sketch (it doesn't have to be complicated, just simple, showing the scarp and the terraces). It could be a general sketch for SDF1 and 2, and WAG.

-Deformation bands. I guess we can refer to them as "foliation". They are sub-vertical, so you can also say that, then they are identified more easily. Otherwise I could tend to think of the sub-horizontal bans within the tails of the wedges, for instance.

-Tension cracks. You refer to them as "filled fissures" in the next section (I guess). Please homogenize terms. Also, In the log, there are not so evident. Please mark and describe better. Is the infill distinctive?

-Geometry of the colluvial wedges. The "flux like" geometry of these wedges is really distinctive. We saw similar geometries at Vila Boda site (in the Suddetic Frontal Thrust, leaded by Petra Štěpančíková). In those trenches, we were considering the action of solifluxion -criofluxion? Geli-fluxion? motivated by ice creep (check some of the abstracts from Petra S., or I provide some sketches below). I found later in you text that hypothesis (line6 page 9) for trench SDF3 wedge-like bodies, which made me more confident with this explanation. Why don't you consider this also for this trench?

[Figure]

[Figure]

Figures taken from studies by Štěpančíková et al.
For instance:

Štěpančíková, P., Rockwell, T., Hartvich, F., Táborík, P., Stemberk, J., Ortuňo, M., Wechsler, N. (2013). Late Quaternary Activity of the Sudetic Marginal Fault in the Czech Republic: A signal of Ice Loading?4th International INQUA Meeting on Paleoseismology, Active Tectonics and Archeoseismology, 259-262 October 2013, Aachen, Germany. 259-263. ISBN: 978-3-00-042796-1

Štěpančíková P., Rockwell T., Nývlt D., Hartvich F., Stemberk J., Rood D. H., Hók J., Ortuňo M., Myers M., Luttrell K., Wechsler N. (2014): A signal of Ice Loading in Late Pleistocene Activity of the Sudetic Marginal Fault (Central Europe). Eos Trans. AGU 2014, Fall Meet., Abstract T41C-4631. San Francisco 15-19 December 2014.

I am not sure if these 5 wedges in SDF1 can all be interpreted as tectonic colluvial wedge s*ensu*

*stricto* by a number of observations:

1) They show lamination. The pebbles in the wedges (or at least at some of them) are well organized and follow the same general orientation than pebbles in the terraces. The pebbles, do not display chaotic orientations (as it would be expected from a sudden collapse).
2) Thee wedge shaped geometry could result from the modification of a rectangular block fault-bounded, i.e., they could be a "pieces" of the terrace that are fault bounded and then the part to the SE has been reworked. It seems that these features display sub-vertical foliation, so I would think they are involved in the fault zone (i.e., a fault would be missing in the SE parth of the "wedges", but it would have been modified by the creep.
3) It is strange to me that the sequence of colores of these wedges is so similar to the sequence of layers within the terrace in the hanging wall.

As you mention, the long tail invokes some kind of creep (you said slump) that is opossite to the idea of a sudden collapse or the pebbles where redeposited down the scarp. I don't think fluvial processes are involved, probably some local process (just colluvial processes smoothing the scarp).

This does not mean that the wedges are not telling us about episodic movement of the fault. Episodic generation of fault scarps from which pebbles (tail parts) are transported to the downtrhown bolck could be inferred as well. Tectonic wedges in trench SDF3 are more clear, and do not seem fault bounded.

**I would be able to have a more clear "judgment" if photologs were provided as supplementary materials.**

-Please give letters or numbers to the units and subunits. One cannot follow the discussion just by the description in the text. For instance, sand layers 2 and 3 (line 14 page 6). I cannot locate them. Are these the colluvial wedges?

- You only refer to photopgraph in Figure 5E. Reference to the other photographs through the text would help to follow descriptions.

SECTION 3.1.1

Need some reorganization. I would also give some more detail here. For instance, the single event displacement and the bracketting units for each of the events. I think this is the section to provide with those data.

IMPORTANT: Number of events. If each "colluvial wedge" is an event, and the younger one is affected, then 6 (and not 4) surface ruptures should be inferred.

**SDF3**

**SECTION 3.2.1**

line 31 page 8. I guess you want to say that not all fault branches were active during successive events. And that this has led to a thinning of the fault zone upwards. That's quite expectable, yes. But I would rather say "abrupt changes in width of the fault zone" and not "reduction of thickness (=thinning)". For instance, the fact that fault branch 2 is not so vertical but tilted leads to a "thickenning" of the fault zone with respect to the lower section, so what you are discussing has more to do with the number of fault branches than with the width of fault zone, I guess.

Please include the fault numbers (F2, F2', etc) in the log of Fig. 6 and refer to them in the text (in the structural description, previous section).

**WAG trench**
line 19 page 9 (evidence is? Are? I think it should be singular)

Please consider to re-interpret the exposure as a fold-scarp, affected by later faulting. The layers in the footwall are folded and form part now of the "deformational bands" that you describe in the text, as part of the fault zone. In a way, it is not so different from a fault zone *sensu stricto*, but the materials affected do not rupture, are not cut and displaced. This might tell us that this site in the termination of a fault segment. In latter stages, the fold scarp is cut by faults.
At the SE most part of the fold-scarp, layers are indeed cut, as you mention: a fault, affecting the unit underlying unit C, is clear. Other faults affetcing the thinned layers (that you call "deformational bands") are also evident. But they seem to have been originated only at the most recent events.

Please provided the units and faults with unit/fault numbers in the log and in the text. Otherwise the discussion is almost impossible to follow.

The section analyzing the events is missing here! Expected section 3.3.1. I wonder why...

**SECTION 4. LUMINESCENCE DATA**.

The tittle of section 4 in my opinion should be "**Dating of events based on luminescence age results**" or some alternative tittle. Then, I would refer to the methodological steps (in an appendix) and some other details in an introductory subsection 4.1 about dating procedure and results. (Now, section 4.1 It is not a "sedimentary and tectonic context").

As stated above, I would move most part of this section to an appendix ("protocol followed for luminescence dating"). I would leave here the discussion of the dating results, which could include the section 4.2 and addressing clue issues such as uncertainties:
Could you at least constrain how much underestimation could be reflected in the results?

**SECTION 5. EVENT CORRELATION**

-Introduction paragraph: To properly locate the reader in what is coming..I would also include in this introduction that 2 possible rupturing scenarios are discussed, implying up to 6 surface ruptures in the area during the last 140 ka.
You said 5 possible common Eqs but you have 6 common Eqs in line (secenario) 1.

EXPECTED MAGNITUDE: I suggest to include this in sub-section 6.2 (about Seismic parameters). Just move there the discussion about the magnitude expected from surface displacement, referring to the average, minimum and maximum values observed. The data observed in 3 trenches are indicative and should give an idea of the maximum event displacement, but nowadays most of the paleoseismological research use the scaling of the surface rupture lenght. See the recommendation done by Stirling et al. (2013), perhaps some other equations are more suitable than Wells and Coopersmith.

**Stirling, M., Goded, T., Berryman, K., and Litchfi eld, N., 2013. Selection of**
earthquake scaling relationships for seismic-hazard analysis: Seismological Society of
America Bulletin, v. 103, p. 2993–3011, doi: 10.1785/0120130052

**Caution!** You should explain first (for instance in section 3 or here but in a brief sub-section) the sequence of events not based in ages, but in constraining units. For that, you need to give names to the subunits. There is no place in the Manuscript where you explain that events A2 and A3 are defined by different bracketting units (the same with B2 and B3). Then, you comment that although different, your age constrains are limmited, and are the same for A2 and 3 (B2, B3). But in your figure 10, the younger limit of event A3 (and B3) is a little older than A2 (A3). This is not based in rigourous age results. I would rather reflect the real time constrain, but would state that defining units are diferent.

For instance. In SDF3, event B2 is constrained by unit 3 and the younger CW (let's say CW , you need to give names to the Cws). But B3 in constrained by CW1 and unit 4. They are different events. However, since you don't have more dating results, you cannot constrain it in a finner way, just can say that both events (B2 and B3) should have happended between the minimum age of unit 5 (I guess, although not in the log); 32.9 $\pm$ 4.1 ka and the oldest possible age of unit 3; 70.8 $\pm$ 8 ka.

**Caution!** Your representation of Common events (E), light grey, is not consistent with your definition in years in Figure 10. For instance, which is the criteria to propose (in the graphical representation) that E3 younger bracket is near 37 ka. I guess you used the 32.9 $\pm$ 4.1 ka (as inferred form the location of the sample) but you cannot do that, it does not fit with the event definition.

SEISMOTECTONIC IMPLICATIONS

Please discuss a little more the problem of variable slip. You just mentioned it very quickly (line 11 page 14) and I think is relevant for recurrence and magnitude estimations. You refer to the "incompleteness" as related to the fact that larger events erase evidence of the the smaller events. If you had a finer stratigraphy and units that you could correlate both sides of the fault, you would probably be able to detect minor events. So it is not only the "fault" of the large colluvial wedges.

Relating the "clear" periodicity of line 1, I honestly don't think that the events are defined in a sufficiently fine way as to infer periodic behavior. Also, their definitions (light gray rectangles) should be revised, as I comment above. Even if they are well located (perhaps the problem is only with E3), I can envisage an EQ distribution matching with line 1 and completely irregular (or clustered). This is the problem we have (I have it in all my studies) when working with rough stratigraphy, which would be overpassed if all trenches were in lake sediments with annual layers!!.

When comparing line 1 and line 2, the implications have not only to do with recurrence. Also with Eqs being recorded (for the case of E3) in a different way (or just not being recorded). That this mean that E3 only affected trench SDF1? Or if present also in SFD3, might be this event E3 implicit in the next event? (this is, E3 is "hidden" in the deformation assigned to B2?).
It would be good to explain which is your preferred scencario, and if you consider that rupture in E3 could have stopped between trenches (they are at both sides of a bend in the trace!).

SECTION 6.1

I think that figure 11 should be better explained in the text. It contains a nice representation of the 2 scenarios, with sub-scenarios implying maximum-minimum ruptures. So it needs further explanation if you decide to keep in in the paper. If kept, it might be used to justify the preferred scenario.

-Why is event E6 not having any associated displacement in figure 11 line 1, minimum slip? It is confusing because it seems you have 5 events (but in line 1, you have 6, if I understand well figure

10 and information in section 5).

I guess that the estimates from paleoseismology...0.03-0.04 mm/yr should be explained here. I cannot find them along the text, only in the concluding section (7).

 And discuss which slip rate you think
-It would be helpful that you give some judgment about which slip rate value you consider more robust, i.e., to take into account in calculation of the seismic hazard.

SECTION 6.3 (note 6.2 is lacking).
I suggest to include (move) here an additional the data about expected magnitudes, taking it from the former section 5 (from each subsection).

Perhaps is this section where you should explain why you think MF is a primary source of earthquakes. Its geometry and relation to the VBTF would also lead to consider that it might move as a secondary fault. I would expect that form the tail/spay geometry of faults. That possibility doesn't mean it is not a valuable source of paleoseismic information. Perhaps is it a better fault for paleoseismic studies than VBTF due to the more complete sedimentary record.
See for instance Beanland et al. 1990 to see an example of how large the secondary slip can be.

Beanland, S., Berryman, K.R., and Blick, G.H., 1989, Geological investigations of the 1987 Edgecumbe earthquake, New Zealand: New Zealand Journal of Geology and Geophysics, v. 32, p. 73–91, doi: 10.1080/00288306 .1989 .10421390.

Finally…. The EQ chronology, does it fit with data in surrounding faults? Perhaps it is the only EQ chronology available and that question does not make sense. Although GPS data indicate the fault system (VBTF) is active, I wonder if the seismogenic events seen here are related with the unloading after the Glaciers retreated in repeated glaciations, leading to "pulses" of enhanced activity. That question is of first order for the seismic hazard. Perhaps no large events are expected to occur at Present?

SECTION 7.
It is a good summary. If changes are done in the former sections, it just should be updated.

FIGURES:
Fig 1. A and B are lacking.
Fig 1A. which time span covered by the seismicity represented? Source of data? same than in figure 1B?
 Fig. 2 mentioned in the caption is not a map... Perhaps the box indicated the location of Figure 3? Please clarify.
It would be nice to locate Eastern Alps and Western Carpathians in the Figure, since they are mentioned in the text. Also the name of the hills right to the NE and SW of the sViena Basin. Someone form South Spain (me) is not so familiar with that local relief.

Fig. 2. It is a section (not a map). So the box in fig. 1A should be replaced by the location of the section.

Remove the text that says "for location see Fig. 2" and provide the meaning (or remove if not referred in the text) the accronyms (e.g., NCA, TWT).

Fig. 3. Please locate this Figure in Fig. 1. If the VBTF is evident in the seismic profile (as mentioned in the figure caption), include it in the figure (Fig.3C at least).
The Gaenserndorf terrace is not indicated in the figure but mentioned in the text.
I highly recommend to modify this figure, including an orthophoto (or aerial photo or a simple sketch) and a few more lines interpreting the geomorphology. The DEM image is insufficient to understand the setting.

Figures 4, and Fig. 8. At least for logs of SDF1 and SDF3 (only lacking in footwall), please include the sub-units. Including the name of the trench in the figure (for instance, at the top of the log) would also help, but if it is clear at the beginning of the figure caption, it is also ok.

Fig 5. Please include a picture of the trenching site. A general picture, so that someone that has not been there can have an idea of how it looks like. It is the fault scarp easy to detect in a field survey?

Fig. 10. Please include the units in this graph. I think it would help to understand the EQ chronology. In general, this graph is useful but I think it should be largely modified to reflect precise definition of events. This figure could be improved if you remark the EQ used in the different correlations. For instance, for E4, I would mark with a distinctive filling (for instance with tilted lines or dots) B3 (just the lower part of it, overlapping with A4) and A4 (in this case, the upper part, overlapping with B4)

---

## Referee Comment (RC5) · M. Ortuño (Referee) · 13 Jul 2017

**Implications from palaeoseismological investigations at the Markgrafneusiedl Fault (Vienna Basin, Austria) for seismic hazard assessment**

5 Esther Hintersberger1, Kurt Decker1, Johanna Lomax2,3, Christopher Lüthgens2

1Department of Geodynamics and Sedimentology, University of Vienna, 1090 Vienna, Austria 2Institute of Applied Geology, University of Natural Resources and Life Sciences (BOKU), 1190 Vienna, Austria 3Department of Geography, Justus Liebig University Gießen, 35390 Giessen, Germany

10 Correspondence to: Esther Hintersberger (esther.hintersberger@univie.ac.at)

Abstract. Including faults into seismic hazard assessment depends strongly on their level of seismic activity. Intraplate regions are characterized by low seismicity, so that the evaluation of existing earthquake catalogues does not necessarily reveal all active faults that contribute to seismic hazard. In the Vienna Basin (Austria), moderate historical seismicity (Imax/Mmax = 8/5.2) concentrates along the left-lateral strike-slip Vienna Basin Transfer Fault (VBTF). In contrast, several normal faults

- 15 branching out of the VBTF show neither historical nor instrumental earthquake records, although geomorphological data indicate Quaternary displacement along those faults. Here, we present a palaeoseismological dataset of three trenches crossing one of these splay faults, the Markgrafneusiedl Fault (MF), in order to evaluate the seismic potential of the fault. Comparing the observations of the different trenches, we found evidence for 5-6 majo face-breaking earthquakes during the last 120 ka, with the youngest event occurring at around ~14 ka before present. The inferred surface displacements lead to magnitude
- 20 estimates ranging between M=6.2±0.3 and M=6.8±0.1. Data can be interpreted by two possible event lines, with event line 1 showing more regular recurrence intervals of about 20-25 ka between the earthquakes with M≥6.5, and event line 2 indicating that such earthquakes cluster in two time intervals in the last 120 ka. Event line 2 appears more plausible. Trench observations also show that structural and sedimentological records of strong earthquakes with small surface offset have only low conservation potential. Vertical slip rates of 0.03-0.04 mm/a derived from the trenches compare well to geomorphically derived
- slip rates of 0.015-0.085 mm/a. Magnitude estimates from fault dimensions suggest that the largest earthquakes observed in the trenches activated the entire fault surface of the MF including the basal detachment that links the normal fault with the VBTF. The most important implications of these paleoseismological results for seismic hazard assessment are that: (1) The MF needs to be considered as a seismic source irrespective of the fact that it did not releas to be considered as a seismic source irrespective of the fact that it did not releas to be considered as a seismic source irrespective of the fact that it did not maximum credible earthquakes in the Vienna Basin should be considered to be about M=7.0. (3) The MF is kinematically and
- 30 geologically equivalent to a number of other splay faults of the VBTF. It must be assumed that these faults are potential sources of large earthquakes as well. The frequency of strong earthquakes near Vienna is therefore expected to be significantly higher than the earthquake frequency reconstructed for the MF.

Natural Hazards of Sciences

**1** Introduction**

During the last years, earthquakes tend to "surprise" seismologist. There by unexpectedly high magnitudes (e.g., Sumatra Earthquakes 2004, Tohuko Earthquake 2011) or/and by the fact that the generating faults were either unmapped (Christchurch Earthquake 2010) or assumed to be inactive (e.g., Haiti Earthquake 2009). Thus, it seems to be clear that historical and instrumental seismicity data are not sufficient to fully characterize the seismogenic potential of a certain region (e.g., Camelbeeck et al., 2007, Liu et al., 2011). Especially in regions of low to moderate seismicity, mostly in intraplate settings, observations of historical and instrumental seismicity are not sufficient to accurately estimate the rate of earthquake activity (Liu et al., 2011). Therefore, during the last decade, geomorphological and palaeoseismological approaches have been increasingly used to map active faults and to determine the related slip rates (e.g., Clark et al., 2012 in Australia, and Vanneste

- et al., 2013, for the Lower Rhine graben system in Central Europe). The results of those studies have dramatically changed the picture and the level depismogenic potential in the analysed regions, mainly in the following aspects: Firstly, palaeoseismological results show that the magnitude for the maximum credible earthquake may be significantly higher than the magnitude to the following largest earthquake observed during historical times (e.g., Central Europe north of the Alps, Figure 1B and references mentioned there). Secondly, the amount of active faults that are considered to be capable of generating
- 15 earthquakes has been increased (e.g., Clark et al., 2012 in Australia). The identification of such "silent" faults as potential seismic sources has become a vital aspect of geological contribution to seismic hazard assessment. Finally, extension of the observed earthquake records raised the question whether faults (especially single faults within fault systems) show regular earthquake patterns during time (characteristic earthquakes occuring in more or less regular time intervals) or if earthquakes occur in so-called super-cycles, where periods of high activity chang with intervals of seismic quiescence (Wallace, 1987, 1987).
- 20 Friedrich et al., 2003). Here, we present results of a paloseismological study, where a dormant active fault has been identified close to the city of Vienna (Austria). Even though there is no historical nor instrumental seismicity that has been recorded along this fault, three trenches across the fault show evidence for five surface-breaking earthquakes. Correlation between the trenches and integration of geomorphological and borehole data helps to identif the fault tends to more characteristic or super-cycle behaviour.

**2 Geological setting**

**2.1 The Vienna Basin**

The Vienna Basin has formed as a pull-apart basin between the Eastern Alps and the Western Carpathians in the Middle and Upper Miocene (e.g. Royden, 1985; Decker et al., 2005). It is located between two left-stepping segments of the NE-SW
striking sinistral strike-slip Vienna Basin Transfer Fault (VBTF, Figure 1). Faulting along this fault system is related to the NE-directed movement of the block east of the Vienna Basin, caused by lateral extrusion of the central Eastern Alps towards the Pannonian Basin (Ratschbacher et al, 1991, Linzer et al, 1997, 2002). GPS data (Grenerczy et al., 2005) and geological reconstruction of Quaternary sediment deposition within the basin (Decker et al., 2005) indicate that the VBTF moves at horizontal velocities between 1.6 and 2.4 mm/y. However, seismic slip rates calculated from cumulative scalar seismic

- 10 moments for different segments along the fault are quite heterogeneous, varying from 0.5-1.1 mm/a at the southern and northern tips to an apparently seismica bally locked segment in the central part of the basin, the so-called Lassee segment, close to the city of Vienna (Hinsch et al, 2005, Hinsch and Decker, 2011). Fault mapping using 2D/3D reflection seism gravity, and geomorphology shows that these seismotectonically defined segments are delimited by major fault bends including a restraining bend (Dobra Voda) and three releasing bends with negative flower structures overlain by Pleistocene pull-apart
- basins with up to 150 m of growth strat bidinger and Decker, 2011). The releasing bends are connected by non-transtensive segments. In addition to the overall geometry of the strike-slip fault with releasing and restraining bends, the transfer of displacement to several normal faults splaying from the strike-slip system in the central part of the basin appears to be an important factor controlling fault segmentation. The splay faults forme big from the Middle to Upper Miocene formation of the Vienna pull-apart basin (Decker et al., 2005) and seem to be kinematically linked to the VBTF via a common detachment (i.e.,
- 20 the Alpine floor thrust, Figure 2, Hölzel et al., 2010, Hinsch and Decker, 2011, Beidinger and Decker, 2011). Those secondary splay normal faults seem to have been seismically inactive during historic times. However, geomorphologic and subsurface geophysical data reveal that those faults indeed show Quaternary displacement of several tens of meters (Chwatal et al., 2005; Decker et al., 2005, Weissl et al., 2017). Moderate historical and instrumental seismicity (Mmax ~ 5.3/ Imax = 8) is concentrated along the VBTF with the 1972 Seebenstein (M~5.3), 1906 Dobra Woda (M~5.7) and the ~ AD 350 Carnutum
- 25 (M~6) earthquakes being the largest known events (Gutdeutsch et al., 1987; Decker et al., 2006; Lenhardt et al., 2007). The scarcity of strong earthquakes and the generally low to moderate seismicity result in estimations of Mmax for earthquakes in the Vienna Basin might not exceed M = 6.0 to 6.5 (Lenhardt et al., 1995; Procházková and Šimunek, 1998; Sefara et al., 1998; Tóth et al., 2006). However, those estimations are solely based on historical and instrumental seismicity.

5

**2.2 The Markgrafneusiedl Fault (MF)**

Our palaeoseismological study is focused on the SE-dipping Markgrafneusiedl Fault (MF) in the central part of the Vienna Basin. It is one of spip splay normal faults that were generated during the Middle to Upper Miocene formation of the Vienna Basin to accommodate transtension at a releasing bend of this sinistral strike-slip fault (Beidinger and Decker, 2011). The

- location of the fault, fault displacement and fault dimensions are evident from 2D and 3D industrial seism 2005, Spahic et al., 2013). An exemplary seismic section is shown in Figure 3. Detailed observations based on 3D industrial seismic data on the fault plane suggests that movement along the MF started on different fault segments that eventually merged together as one larger fault (Spahic et al., 2013). Quaternary fault reactivation is inferred from geomorphological evidence of
- a linear scarp paralleling the outcrop trace of the fault, high-resolution geophysical profiling (georadar, reflection seismic, geoelectrics; Chwatal et al., 2005) and the ca. 40 m offset of the base of the Quaternary sediments across the MF (Decker et al., 2005). The visible fault scarp falls togethe bit the SE edge of the Gaenserndorf terrace, building a linear geomorphological step of ca. 12 m height in the present-day topography (Figure 3).

Despite this well documented Quaternary displacement along the MF, no historical seise by is recorded that can be associated with this fault, except for small earthquakes with magnitudes less than 1.4 the been recorded close to the MF in the last

decade. Whether this apparently slowly moving fault can produce larger earthquakes or it is aseismically creeping, is the key question of our study, during which three trenches (from north to south WAG, SDF1, and SDF3) were excavated across the MF between the villages of Markgrafneusiedl and Gaenserndorf, about 15 km from the city limits of Vienna, the Austrian capital. The results show that these normal faults are indeed capable of generating earthquakes and therefore must be considered as potential seismogenic sources. In addition, the observations indicate that earthquakes within the Vienna Basin

could exceed the maximum magnitudes estimated from historical and instrumental seismicity.

**3** Trenching results**

In total, we excavated two trenches along the geomorphic fault scarp between the villages of Markgrafneusiedl and Gaenserndorf (Figure 3A). For the exact position of the trenches, 40 MHz ground penetration radar (GPR) profiles were carried

- 25 out showing the location of the MF at the base of the present-day scarp. In addition, a construction pit of a gas pipeline exposed the northern tip of MF, providing additional, but limited, information. In general, all outcrops show similar characteristics: at all trenching locations, the MF is exposed as narrow (1 - 2 m) fault zone consisting of one or two fault branches striking parallel to the regional strike of the fault scarp of the MF (dip direction/dip: ~120/75). The footwall cut by MF comprises deposits typical for the Gaenserndorf terrace (Weissl et al., 2017 and references therein). The hanging wall of the trees expose
- 30 sequences of almost horizontally layered, fine-graded sedimen

**3.1 Trenching at SDF1**

The 40-m-long, 3-m-wide and up to 4 m deep trench SDF1 was located close to the farm house "Siehdichfür", about 20 km from the city limits of Vienna. It was excavated in a small dry valley at the central part of the NE-SW trending geomorphological fault scarp with the exact location of the MF at its base. Trench mapping in the scale of 1:10 covers both,

5 the entire SW wall of the trench and the section around the fault zone at the NE wall. The trench SDF1 exposed about the fault zone of the Gaenserndorf terrace deposits in the footwall and ca. 10 m of the hanging wall, divided by the 1.5 m wide fault zone of the SE-dipping MF. The fault zone includes two parallel steeply dipping faults F1 and F2, with F2 reaching almost the present-day surface (see Figure 4).

At the NW part in the footwall, alluvial deposits of the Gaenserndorf terrace are exposed, consisting of coarse gravels and

- 10 boulders. Pebbles show consistent NW-dipping imbrication throughout the entire footwall section. The inferred dominantly SE-directed paleocurrents are comparable to the flow direction of the Recent Danube. In addition, two approximately 8 m wide sandy ancient river channel fills are observed close to the top of the succession. Another sand lense, only partly exposed at the base of the outcrop, is cut by the fault zone. The uppermost 0.5 m of the terrace deposits directly below the recent soil horizon do not show any horizontal consistency and are most probably reworked and repositioned.
- 15 In the hanging wall SE of the fault zone, three types of sediments are expose (A) Sequences of horizontal layers of light-grey and light-brown silt and fine sand with varying thicknesses up to 20 cm. Sediments show lamination on cm-scale and intercalations of cm-thick horizons of coarse sand. The layers also include singular well-rounded pebbles and granules aligned in horizontal layers. Some sand/silt layers show fining-upward trends. Carbonate cementation is observed along the top of the uppermost silt layer and along recent root paths. The sediments are intercalated
- 20 with and onlap on the wedge-shaped colluvial deposits described below. We relate the deposits to high-stage floods in the floodplain of the Pleistocene Danub

(B) Colluvial wedge deposits and associated tension crack fills. These colluvial sediments are attached to both faults and decrease in thickness towards the SE (i.e., away from the fault scarp; Figure 4). The steep contact with the SE-dipping faults and the thinning of the deposits towards SE results in a wedge-shape of the sediment layers. The tails of wedges 2, 3 and 5 can

- 25 be followed throughout the exposed part of the hanging wall. All wedges are associated with tension cracks adjacent to the fault, which are filled with the same material as the overlying wedge. Wedge onsists of matrix-supported reddish brown medium gravel with a matrix composed mainly by sand and silt together with a low content of clay. Wedge 4 is delimited by a steep irregular boundary adjacent high-stage flood sediments. While wedge 4 comprises brown to reddish brown fine to medium sand with some fine granules and pebbles in a matrix-supported fabric, the latter include rounded pebble-size clasts
- 30 of the reddish wedge material interpreted as mud balls. We interpret this peculiar contact to result from the partial erosion of the wedge and the wedge tail during high-stage floods and the re-deposition of the colluvial material by fluvial processes or small slum vedge 3 consists well-sorted reddish brown middle sand with a few pebbles (fine gravel) showing lamination dipping away from Fault 1. These three wedges contrast by their red and reddish-brown colour from the intercalated highstage flood deposits. The sedimentary material was identified as redeposited soil, which by its colour, resembles ferretto soils (5YR 4/4, Y5YR 5/4 and 5YR 58 of the standard soil colour chart; L. Smolíková, pers. comm.), which derived from the soil cover of the terrace gravels in the footwall of the MF. While the lower three colluvial wedges (3-5) are bounded by F1, wedge 2 is attached to F2 and overlies the trace of F1as well as the deposits of wedge 3. The wedge consists of large well-rounded

- 5 pebbles and cobbles oriented sub-horizontally in a grain-supported fabric, similar to the terrace deposits found in the foot wall. (C) Fine-grained alluvium and loess. The uppermost part of the sedimentary succession of both the hanging wall and the footwall consists of several thin layers of sand and fine gravel overlain by up to 1 m of unstructured silt and fine sand. The latter is transitional to the overlying dark brown to black soil horizon. The succession is interpreted as alluvium of the dry valley and loess-like sediments or redeposited loess. Fault 2 offsets the all values for about 15 to 20 cm, but terminates
- 10 within the overlying loess-like sediments several cm above the base of the layer. Structural data obtained from the two faults exposed in the outcrop show that both faults strike parallel to the regional strike of the fault scarp of the MF. The faults are marked by bands of pebbles with preferred orientations parallel to the fault planes. Pebbles in the 75 cm thick fault block between the two faults show orientations which geometrically resemble S-C-type fabrics. Deformation bands are found in the sand wedges 3 and (a)t the related tension crack period or the mapping reveals that
- 15 these microfaults do not penetrate into the colluvial wedge 5 most probably due to the higher clay content of these sediments. The deformation bands show orientations consistent with the main faults of the outcrop. At the lower parts, the deformation bands are dipping parallel to F1 (dip direction/dip 130/80). The upper parts of the deformation bands are rotated away from the fault resembling horsetail splays. The orientations of the sub-vertical deformation bands vary between 303/78 and 330/78. In addition to some major deformation bands, which are traced for about 1 m across the profile, there are shallow dipping
- 20 deformation bands with comparably large normal offsets up to several mm (145/20). Finally, small-scale normal faults with displacement in the order of several centimeters are observed within the uppermost layers that have been also affected by the youngest displacement along F2 (Figure 5E).

**3.1.1 Evidences for seismic events observed within trench SDF1**

Offset of alluvial sand layer the tip of F2 provides direct evidence for the youngest surface-breaking slip event A1. The small-scale faults observed at the same stratigraphic level are further indications for an earthquake at this fault. A1 is offsets and postdates colluvial wedge 2. The observed colluvial wedges, their geometrical relation to the adjacent faults, and the sediment-filled extension fissures prove four distinct events (A2 to A5) of rapid co-seismic displacement at the MF relation, the existence of deformation band within the sandy colluvial wedges 3 and 4 indicates further deformation of both wedges during younger slip events at the MF. Among the earthquakes excaved by the trench, only slip associated with A1 is directly constrained by the offset of layers correlated across Fault 2. Evidence for the earthquakes A2 to A5 comes from the colluvial wedges 2 to 5 and the refilled tension cracks 2' to 5' below the wedges. Following the generally accepted rule of thumbs that colluvial wedge height is approximately half of the surface displacement of an earthquake (McCalpin, 2008), the measured

maximum thickness of each colluvial wedge can be used to estimate the minimum displacement for the associated even

**3.2 Trenching at SDF3**

Trenching in the Vienna Basin continued with the opening of a second trench SDF3 across the same fault. This 33 m long, 3 m wide and up to 5 m deep trench is located about 1.5 km SW of the first trench SDF1. Trench mapping in the scale of 1:10 covers both, the entire W wall of the trench and the section around the fault zone exposed in the terraced E wall (Figure 6).

[revised manuscript text omitted]

- 15 (dip direction/dip: 116/74). The MF is marked by the contact between the footwall gravels and the more sandy deposits of the hanging wall. In addition, at the lower 1.5 m, clasts within a zone of about 50 cm to the fault are rotated parallel to the fault (dip direction/dip: 116/69). The upper part of the MF is only marked by a small band of rotated clasts. However, in this upper part of the fault, layers that can be correlated on both sides of the fault, are displaced by about 15 cm and, therefore, indicate the youngest movement along the fault. In addition to the main faults, several conjugated sets of normal faults are observed
- 20 within lower units of the hanging wall. These faults are consistently oriented parallel to the MF. The NW-dipping antithetic faults (dip direction / dip: 303/79) are generally longer than the SE-dipping faults (dip direction / dip: 137/72). Displacement observed along the faults is in the range of about 10 cm in the lowest unit 1, and up to 1 cm in the reddish clay-rich unit 3. None of the small faults see penetrate into the gravels overlying the reddish clay-rich sand layer (unit 3). Within this layer, the small faults are recognised as a few mm thin deformation bands, most probably filled with carbonate cement, and show
- 25 almost no displacement. The sand layers of the lowermost unit 1, consisting of intercalated layers of sand and matrixsupported gravels, comprises small deformation ban pith lengths up to 20 cm. They are arranged parallel to the small faults and accordingly dip towards the SE or the NW.

**3.2.1 Evidences for seismic events observed within trench SDF3**

MF within the trench SDF3 is a very narrow fault zone of 0.5 m width at its lowest point excavated within the trench, and reduces to a fault represented only by a few rotated clasts in the uppermost part. However, this reduction of thickness is not a continuous, but occurs in distinct steps. Those steps can be related to different earthquat. The oldest earthquake that can be identified within the trench is B5 that created a colluvial wedge along the fault trace F3. This fault trace is then covered by another colluvial wege, which was most probably created by movement along F2 during B4. Evidence for the event B3 is a

tension crack between F2 and 1000 hat is also identified by the thin sand layer that is smeared into the crack, parallel to F2'. B2 is identified by a ~ 0.8 m thick colluvial wedge. However, the fault strand bounding this colluvial wedge is not obvious. This situation may be explained by the following scenario: In the case that coseismic surface rupture offset unconsolidated water-saturated sandy gravel, it seems plausible that no long-standing free surface and colluvial wedge adjacent to the fault

- 5 plane could form. Instead, the offset soft sediment may have collapsed during or shortly after the earthquake forming a wedgeshaped deposit, which overlies the uppermost part of the ruptured fault. The same geometry may result from geli-solifluction under periglacial conditions when material glides down to the hangingwall destroying a previously formed free surface. The latter scenario is supported by the observation of a smooth change between the horizontal layers of the terrace and the inclined layers in the colluvial wedge. The described situation allows for two different interpretations of the surface displacement of
- B2. Interpreting the wedge-shaped deposit as a classical colluvial wedge adjacent to a fault plane which is not readily seen due to unfavourable outcrop conditions, a minimum displacement can be estimated by multiplying the maximum wedge height by two (McCalpin, 2008), which would result in a displacement of  $2 \times 0.8 \text{ m} = 1.6 \text{ m}$  for B2. In case that the wedge formed by free surface collapse of water-saturated sediment or gelifluction the coseismic surface displacement be approximately the same as the colluvial wedge height, i.e. 0.80 m.
- 15 Insights for the youngest event B1 in the trench are more obvious. Displacement of the upper layers for ~ 10 cm affected all layers excluding only the soil horizons (units 7 and 8), suggesting that even with such a small displacement of only 10 cm, the event B1 ruptured the surface.

**3.3 Trenching at WAG**

Additional evidence for active faulting at the MF are available from the construction pit of a gas pipeline, which crosses the

- 20 northern part of the fault scarp close to the city of Gaenserndorf, 6 km north of trench SDF1. The outcrop revealed a 1-m-wide localized fault zone (Figure 8). The fault cuts light-grey gravel and sand of the Gaenserndorf (Date and overlying loess-like sediments (silt, fine to medium grained sand) constituting its footwall. The exposed hanging wall succession includes poorly sorted sandy gravel, which is then overlaid by a banded sequence of silty sediments. This cover layer can be found all along the pipeline construction pit and has been described in detail by Weissl et al. (2017). Both the hanging wall and footwall are
- 25 overlain by c. 30-50 cm thick brown soil, which has been removed prior to the excavation. The exposed fault zone consists of several deformation bar vithin the terrace gravel marked by aligned and fractured pebbles, faults offsetting sand layers, and faults offsetting the contact between gravel units and the overlying cover silts (Figure veral sheared pebbles indicate dip-slip movement along the deformation bands. The displacements of these faults are between 10 and 20 cm. On the southern wall a fault cuts up through the entire silty section to the base of the overlying soil, offsetting a thin white layer within the
- 30 upper part of the cover silty sediments by about 20 cm.

**4 Luminescence dating $\int$**

Luminescence dating is commonly used to date the time that feldspar and/or quartz grains in sandy or silty sediments were not exposed to sunlight, and therefore to constrain deposition ages of those sandy or silty sediment bodies. Regarding the physical background and the basics of luminescence dating methods we refer to previously published review papers of Preusser et al. (2008). Wintle (2008) and Phodes (2011)

5 (2008), Wintle (2008), and Rhodes (2011).

**4.1 Sampling and experimental setup**

In analogy to the procedure described by Weissl et al. (2017), samples were collected in the field by driving an opaque steel cylinder into the freshly cleaned sediment surface and transferring the material into light tight plastic bags. All subsequent sample preparation steps were conducted under subdued red light conditions in the Vienna laboratory for luminescence dating.

Samples were first dried and dry sieved. The grain size fraction of 100 - 200 mm was used for further preparation steps. The material was subjected to 15% HCl to remove carbonates, treated with Na2C2O4, (0.01 N) to disperse clay particles, and with 10% H2O2 to dissolve organic components. Quartz and feldspar separates were obtained by density separation using LST Fastfloat.

In this study, we used potassium-rich feldspar as luminescence dosimeters for age determination. All fractions were measured

- 15 with small aliquots of 1 mm diameter mask size using a grain size fraction of 100 200 mm. All measurements for determination of the equivalent dose were conducted in the Vienna laboratory for luminescence dating on RISØ TL-OSL DA 20 automated luminescence reader systems (Bøtter-Jensen et al., 2000, 2003). For **E** termination of the feldspar fraction, a conventional SAR IRSL protocol was applied (Wallinga et al., 2000; Blair et al., 2005), using a preheat of 250°C for 20 s and a stimulation at 50°C for 300 s. Stimulation was carried out with IR-LEDs, and signals were detected after passing through a
- blue interference filter ( $410 \pm 20$  nm). Doses were determined on small multi-grain aliquots (mask-size 1 mm). Over-dispersion (Galbraith et al. 1999) was below 11% in all samples confirming a generally well-bleached nature of the sediments. It needs to be stressed that the feldspar based ages were not corrected for fading. Fading describes an anomalous signal loss very commonly observed for potassium-rich feldspar (Wintle, 1973). If not corrected for, fading leads to the underestimation of the burial age. However, samples from the same study area investigated by Weissl et al. (2017) showed little or no fading, as
- 25 demonstrated by a comparison between quartz and feldspar luminescence ages. Nevertheless, all ages presented here need to be treated with caution for potential age underestimatic concentrations for dose-rate estimation were determined on ~900 g of bulk sediment using high resolution, low-level gamma-spectrometry. Samples were first dried, homogenised and stored in sealed Marinelli beakers (500 ml, about 1 kg dry weight) for at least a month to establish secondary secular radon equilibrium. Measurements were conducted using a Canberra HPGe detector (40% n-type). Relevant
- 30 luminescence data is listed in Table 2.

---

## Author Comment (AC1) · 25 Aug 2017

Response to RC1 (Ryan Gold):

First of all, thank you for the detailed review that help us not only improve the manuscript, but also pushed us to use OxCal to better present the correlation between the trenches.

Major comments 1: Probabilistic framework. This topic has been also brought up by RC2 (Dan Clark). We were not aware that OxCal can also be used for IRSL dating

results; we thought that it was mostly used for calibration of radiocarbon ages. Since all the time constraints for our trenches come from IRSL dating, we thought that OxCal was not applicable in this study. But following the suggestion of both reviewers, we managed to transfer our IRSL dating results into OxCal and obtained good results. However, the results are comparable to the results previously shown in the manuscript. We added the resultant occurrence intervals to table 3.

2: Earthquake correlation between sites. Again, this topic was also raised by RC2. We were not aware that the numbering that we used suggests 8 earthquakes in total (when there is evidence for 5-6). What we had in mind was the overall number of possible correlations between trenches and an easy way of reference to each single correlation. Unfortunately, this caused more confusion than the intended clarification. Therefore, we changed the labeling in text, figures and tables in the way suggested by RC1 (E1, E2, E3, E4a, E4b, E5a, E5b, E6a). We also changed the seemingly confusing term "event line" to "slip model". Regarding the comment on earthquake occurrence times given as ranges vs. as PDFs: The dating constraints along the MF in the Vienna Basin are loose for each single event. The resultant PDFs for earthquakes along the MF are broad distributions with large standard deviations. Therefore, we though that it would be more straight-forward to rather show the time brackets than to construct mean dates with large error bars. However, we added the OxCal results to table 3 to provide both types of information.

3: Periodic vs. clustered. Yes, you are right, the uncertainties for the recurrence rates are large. Nevertheless, we wanted to stress out the importance of such possibilities. We did calculate the COV for each slip model and obtained higher COV for the clustered slip model than for the periodic slip model. However, we are hesitant to use it because of the small sample size. Most studies, where COV were applied to distinguish between periodic and aperiodic behavior, had at least 10, or even 25 earthquake occurrence times. In such cases, the COV are more meaningful than in the study here.

4: Characteristic vs. super-cycle. Yes, you are right, we got confused here. The

reason why we mentioned it here was because it seems that these faults also are quiet for a long time and then are switched on (maybe triggered by the VBTF). Hence the comparison to the characteristic vs. super-cycle. But in hindsight, we agree that it is better to stick with the discussion about periodic vs. clustered. We changed the introduction in this sense.

5: Unit ages. Yes, we agree. It was hard to find the right place within the paper to describe the dating results. We did not want to present the dating results before the method. By including a methodology section into the manuscript, this problem is solved, and we added the age of the units, where available.

6: Linkage to the Vienna Basin Transfer Fault. We do think that the MF is connected to the VBTF via the common detachment, and we also mention it shortly that in the discussion about the possible activation of the detachment during an earthquake along the MF. However, there is no final/published paleoseismological data yet to link both faults.

7: Geomorphic site/topo profiles. In order to keep the number of figures in check, we thought that Figure 3 would be enough to present the general situation along and below the trench sites. But as suggested by all three reviewers, we have added a close-up geomorphic map for both trenches as well as topographic profiles and hope that this gives a better understanding for the reader.

Moderate/general comments: 1: Event lines. We changed the seemingly confusing term "event line" to "slip model".

2: Subjective word choice. Thanks to your detailed supplementary commentary, we changed/deleted the respective terms.

3: Mmax for the Vienna Basin. You are right, a combined rupture would lead to a larger earthquake magnitude. And we also think that this a very important part to keep in mind. But since we focused in this paper on data for the MF and the impact of this fault

to the seismic hazard, we thought that the scenario of a combined rupture of the VBTF and the MF might be beyond the scope of this manuscript. There is a study by Hinsch & Decker (2011) presenting different rupture scenarios along the VBTF. The resultant Mmax are in the order of 5.9-6.8 for different segments of the VBTF. So, therefore, a possible combined rupture of the VBTF and the MF would include several parameters and additional options that is better addressed in a manuscript by its own.

4: Luminescence dating. Following the suggestion, we have reduced the chapter about dating and placed it into the methodology part (see also response to major comment 5 on unit ages above and methodology section below).

5: Uncertainties. We added the information in the tables and explicitly stated in the methodology section that the luminescene dating results are given with $1\sigma$ uncertainties (see below).

6: Methods section. Thank you for the suggestion. We restructured the manuscript and added a methodology section including the photomosaic generation, logging, sampling and dating, and details of the OxCal calculations.

7: Haiti earthquake. Sorry for the typo. As far as I remember the discussion, you were right that the fault that generated the earthquake was already mapped, but was assumed to be not the active fault strand within the fault system. We will check this again and, if this was wrong, change the relevant sentences.

Table 1 and Table 2: We added the additional information to avoid confusion.

Table 3: We changed the numbering of events to make clear that they are only different correlation scenarios. We added also the obtained occurrence interval for each earthquake. Figures: We made the changes to the figures as suggested by all reviewers, added the uninterpreted photo mosaics to the supplementary, added also a figure of the geomorphic/geological situation around the trenches, and rearranged Figure 10 using the OxCal results.

Please also note the supplement to this comment:
https://www.nat-hazards-earth-syst-sci-discuss.net/nhess-2017-126/nhess-2017-126-AC1-supplement.pdf

―――――――――――――――――――

---

## Author Comment (AC3) · 25 Aug 2017

**Response to RC2 (Dan Clark):**

First of all, thanks for the review and the interesting questions that have been mentioned within the review. They definitively will inspire further studies and publications. However, at the moment, trying to answer them would be mostly speculation, because the MF is only the second fault in the Vienna Basin that has been investigated with paleoseismological methods. The other studied fault is a much smaller fault on the western margin of the Gaenserndorf terrace, where there was no clear exposure of the fault within the trench (Weissl et al., 2017). Even though there is information about the long-term Quaternary displacement along most of the faults from boreholes (Decker et al., 2005), detailed information about paleoseismic events along the faults is not (yet) available. Paleoseismological investigations at the VBTF are, at the moment, not finished. Therefore, more work must be carried out before we can address these questions. Hence, we think addressing those questions is beyond the scope of this paper and the data presented here.

**Probabilistic framework.** This topic has been also brought up by RC1 (Ryan Gold). We were not aware that OxCal can also be used for IRSL dating results; we thought that it was mostly used for calibration of radiocarbon ages. Since all the time constraints for our trenches come from IRSL dating, we thought that OxCal was not applicable in this study. But following the suggestion of both reviewers, we managed to transfer our IRSL dating results into OxCal and obtained good results. However, the results are comparable to the results previously shown in the manuscript. We added the resultant occurrence intervals to table 3.

**Linkage to the Vienna Basin Transfer Fault**. We do think that the MF is connected to the VBTF via the common detachment, and we also mention it shortly that in the discussion about the possible activation of the detachment during an earthquake along the MF. However, there is no ready to be published paleoseismological data yet to link both faults. See also comment to RC1 and RC4.

**General comments:**

**Labeling of the units.** We followed your suggestion and labeled the colluvial units. In addition, due to the comments also provided by the other reviewers, we changed the figures for better understanding of the units and the position of the detailed figures in the trench logs.

**Combination of age data between trenches.** Due to the recalculation of the occurrence times in OxCal, we have rewritten the section and changed Figure 10 accordingly. See also respective comment to RC1. We did calculate the COV for each slip model and obtained higher COV for the clustered slip model than for the periodic slip model. However, we are hesitant to use it because of the small sample size. Most studies, where COV were applied to distinguish between periodic and aperiodic behavior, had at least 10, or even 25 earthquake occurrence times. In such cases, the COV are more meaningful than in the study here.

A larger area than only the MF by rupture of the VBTF and MF. You are right, a combined rupture would lead to a larger earthquake magnitude. And we also think that this a very important part to keep in mind. But since we focused in this paper on data for the MF and the impact of this fault to the seismic hazard, we thought that the scenario of a combined rupture of the VBTF and the MF might be beyond the scope of this manuscript. See also comment to RC1.

**Periodic/aperiodic vs. characteristic/supercycle.** Yes, you are right, we got confused here. The reason why we mentioned it here was because it seems that these faults also are quiet for a long time and then are switched on (maybe triggered by the VBTF). Hence the comparison to the characteristic/super-cycle. We changed the introduction in this sense. See also comment to RC1.

**Creep.** Prior to our study, the MF was suggested to creep aseismically. However, we did not find any evidence for creep in the trenches. See also comment below and comment to RC4.

The **lack of a geomorphic site sketch/map** has also been mentioned by the other reviewers. We included the relevant figure as well as the topographic profile and the landscape picture as suggested and added the names of the towns. See respective responses to RC1 and RC4.

**Far field displacement.** Separation of red horizon is due to the sedimentation on pre-existing topography. Besides this, the total topographic step is about 17 m (as visible in Figure 3), but in the trenches, there is no sign for afterslip or interseismic creep.

**Threshold for surface rupture in this area and their discoverability.** One of the largest earthquakes in the Vienna Basin, the Dobra Voda Earthquake of 1906 had a macroseismic magnitude of M=5.7 and epicentral intensities of Io=8-9. From field surveys after this earthquake, discontinuous surface cracks in the order of 1-2 m depth are documented. Slip along those cracks was recorded to be between 50 and 100 cm. However, it is hard to tell from the available description whether those features were primary or secondary ruptures. Regarding the discoverability and preservation of such small single-event fault scarps: the Vienna Basin is an intensively agriculturated area. This together with strong erosion may hide single event scarps outside of forested areas.

**Mmax as minimum Mmax.** Yes, you are right. Of course, this is the Mmax that at least should be considered, based on the data presented here. If interaction between faults and rupture of several segments is considered, the Mmax for the Vienna Basin would be definitively increase. See also respective comment to RC1.

**Figures:** We went through the figures with the annotations in mind, removed all inconsistencies, added and explained labels, where missing, changed Figure 10 to add the OxCal results. We finally added the uninterpreted photomosaiques to the supplementary.

---

## Author Response (AR1)

**Authors' Response:**

First of all, thank you to all reviewers for the detailed reviews that help us immensely to improve the manuscript.

**1. List of major changes:**

- New section about Methodology (section 3) including subsections about 1) field work, trenching, and logging; 2) luminiscene dating; 3) modelling of earthquake occurrence times using OxCal.
- Extensive revision of Section 4: trench descriptions, especially adding subsections about age control to descriptions of trenches SDF1 and SDF3.
- Revision of Section 5, including the OxCal results.
- New section about Completeness of preservation potential of earthquakes in the trenches (section 6.1)
- Extensive revision of section 6.2 (formerly section 6.1) about recurrence rates of earthquakes with M≥6.5
- Addition of explicit description how paleoseismological slip rates were obtained (section 6.3)
- New section about Mmax in the Vienna Basin and the linkage between MF and VBTF (section 6.5)
- Table 1 (formerly table 2) was extended by additional IRSL dating information.
- Table 2 (formerly table 1) was extended by additional information about the single earthquake occurrence times (OxCal results).
- Table 3 was extended by additional information about the combined earthquake occurrence times and respective magnitude estimates.
- Minor changes were applied to Figures 1, 2, 3, 6, 8, 9, 10, 14. New Figures 4 and 12 were added. Figures 5 and 7 were changed to provide more detailed trench logs. Figures 11 and 12 were changed to account the OxCal results.
- Supplementary files containing high-resolution photomosaiques of trenches SDF1 and SDF3 are attached, as well as the OxCal code.

**2. Response to RC1 (Ryan Gold):**

**Major comments**

**1: Probabilistic framework.** This topic has been also brought up by RC2 (Dan Clark). We were not aware that OxCal can also be used for IRSL dating results; we thought that it was mostly used for calibration of radiocarbon ages. Since all the time constraints for our trenches come from IRSL dating, we thought that OxCal was not applicable in this study. But following the suggestion of both reviewers, we managed to transfer our IRSL dating results into OxCal and obtained good results. However, the results are comparable to the results previously shown in the manuscript. We added the resultant occurrence intervals to tables 2 and 3. We also changed Figure 10 (now figure 11) and added Figure 12 showing the recurrence rates for both slip models.

**2: Earthquake correlation between sites.** Again, this topic was also raised by RC2. We were not aware that the numbering that we used suggests 8 earthquakes in total (when there is evidence for 5-6). What

we had in mind was the overall number of possible correlations between trenches and an easy way of reference to each single correlation. Unfortunately, this caused more confusion than the intended clarification. Therefore, we changed the labeling in text, figures and tables in the way suggested by RC1 (E1, E2, E3, E4a, E4b, E5a, E5b, E6a). We also changed the seemingly confusing term "event line" to "slip model". Regarding the comment on earthquake occurrence times given as ranges vs. as PDFs: The dating constraints along the MF in the Vienna Basin are loose for each single event. The resulting PDFs for earthquakes along the MF are broad distributions with large standard deviations. Therefore, we though that it would be more straight-forward to rather show the time brackets than to construct mean dates with large error bars. However, we added the OxCal results to tables 2 and 3 to provide both types of information.

**3: Periodic vs. clustered**. Yes, you are right, the uncertainties for the recurrence rates are large. Nevertheless, we wanted to stress out the importance of such possibilities. We did calculate the COV for each slip model and obtained a COV > 1 for the clustered slip model, higher than for the periodic slip model. However, we are hesitant to use it because of the small sample size. Most studies, where COV were applied to distinguish between periodic and aperiodic behavior, had at least 10, or even 25 earthquake occurrence times. In such cases, the COV are more meaningful than in the study here.

**4: Characteristic vs. super-cycle.** Yes, you are right, we got confused here. The reason why we mentioned it here was because it seems that these faults also are quiet for a long time and then are switched on (maybe triggered by the VBTF). Hence the comparison to the characteristic vs. super-cycle. But in hindsight, we agree that it is better to stick with the discussion about periodic vs. clustered. We changed the introduction in this sense.

**5: Unit ages.** Yes, we agree. It was hard to find the right place within the paper to describe the dating results. We did not want to present the dating results before the method. By including a methodology section into the manuscript, this problem is solved, and we added the age of the units to the new sections about age control for each trench, where available.

**6: Linkage to the Vienna Basin Transfer Fault**. We do think that the MF is connected to the VBTF via the common detachment, and we also mention it shortly that in the discussion about the possible activation of the detachment during an earthquake along the MF. However, there is no final/published paleoseismological data yet to link both faults.

**7: Geomorphic site/topo profiles.** In order to keep the number of figures in check, we thought that Figure 3 would be enough to present the general situation along and below the trench sites. But as suggested by all three reviewers, we have added a close-up geomorphic map for both trenches as well as pictures from the trench sites and hope that this gives a better understanding for the reader (Figure 4).

**Moderate/general comments:**

**1: Event lines.** We changed the seemingly confusing term "event line" to "slip model".

**2: Subjective word choice.** Thanks to your detailed supplementary commentary, we changed/deleted the respective terms.

**3: Mmax for the Vienna Basin.** You are right, a combined rupture would lead to a larger earthquake magnitude. And we also think that this a very important part to keep in mind. But since we focused in this paper on data for the MF and the impact of this fault to the seismic hazard, we thought that the

scenario of a combined rupture of the VBTF and the MF might be beyond the scope of this manuscript. There is a study by Hinsch & Decker (2011) presenting different rupture scenarios along the VBTF. The resultant Mmax are in the order of 5.9-6.8 for different segments of the VBTF. So, therefore, a possible combined rupture of the VBTF and the MF would include several parameters and additional options that is better addressed in a manuscript by its own. To address the issue about Mmax that has been brought up also by the other reviewers, we added a section (section 6.5).

**4: Luminescence dating.** Following the suggestion, we have reduced the chapter about dating and placed it into the methodology part (see also response to major comment 5 on unit ages above and methodology section below).

**5: Uncertainties.** We added the information in the tables and in the text and explicitly stated in the methodology section that the luminescene dating results are given with 1σ uncertainties (see below).

**6: Methods section.** Thank you for the suggestion. We restructured the manuscript and added a methodology section including the photomosaic generation, logging, sampling and dating, and details of the OxCal calculations (section 3).

**7: Haiti earthquake.** Sorry for the typo. As far as I remember the discussion, you were right that the fault that generated the earthquake was already mapped, but was assumed to be not the active/relevant fault strand within the fault system. The then used seismic hazard map of GSHAP (Giardini et al., 1999) shows a much lower exceedance probability for the region around Port-au-Prince than seismic hazard maps after the earthquake (e.g. Frankel et al., USGS Open-File Report 2011). We changed the text to "not considered in seismic hazard analysis" in order to avoid confusion.

[Figure]

**Table 1 and Table 2:** We added the additional information to avoid confusion.

**Table 3:** We changed the numbering of events to make clear that they are only different correlation scenarios. We added also the obtained occurrence interval for each earthquake.

**Figures:** We made the changes to the figures as suggested by all reviewers, added the uninterpreted photo mosaics to the supplementary, added also a figure of the geomorphic situation/landscape around the trenches, and rearranged Figure 10 (now Figure 11) and added Figure 12 using the OxCal results.

**Additional annotated PDF:** Thank you for taking the time to annotate the PDF, we changed the manuscript accordingly.

**3. Response to RC2 (Dan Clark):**

First of all, thanks for the review and the interesting questions that have been mentioned within the review. They definitively will inspire further studies and publications. However, at the moment, trying to answer them would be mostly speculation, because the MF is only the second fault in the Vienna Basin that has been investigated with paleoseismological methods. The other studied fault is a much smaller fault on the western margin of the Gaenserndorf terrace, where there was no clear exposure of the fault within the trench (Weissl et al., 2017). Even though there is information about the long-term Quaternary displacement along most of the faults from boreholes (Decker et al., 2005), detailed information about paleoseismic events along the faults is not (yet) available. Paleoseismological investigations at the VBTF are, at the moment, not finished. Therefore, more work must be carried out before we can address these questions. Hence, we think addressing those questions is beyond the scope of this paper and the data presented here.

**Probabilistic framework.** This topic has been also brought up by RC1 (Ryan Gold). We were not aware that OxCal can also be used for IRSL dating results; we thought that it was mostly used for calibration of radiocarbon ages. Since all the time constraints for our trenches come from IRSL dating, we thought that OxCal was not applicable in this study. But following the suggestion of both reviewers, we managed to transfer our IRSL dating results into OxCal and obtained good results. However, the results are comparable to the results previously shown in the manuscript. We added the resultant occurrence intervals to tables 2 and 3.

**Linkage to the Vienna Basin Transfer Fault**. We do think that the MF is connected to the VBTF via the common detachment, and we also mention it shortly that in the discussion about the possible activation of the detachment during an earthquake along the MF. However, there is no ready to be published paleoseismological data yet to link both faults. See also comment to RC1 and RC4. We added a section about the Mmax in the Vienna Basin, where we also discussed this (section 6.5).

**General comments:**

**Labeling of the units.** We followed your suggestion and labeled the colluvial units. In addition, due to the comments also provided by the other reviewers, we changed the figures for better understanding of the units and the position of the detailed figures in the trench logs.

**Combination of age data between trenches.** Due to the recalculation of the occurrence times in OxCal, we have rewritten the section and changed Figure 10 (now Figure 11) accordingly. See also respective

comment to RC1. We did calculate the COV for each slip model and obtained higher COV for the clustered slip model than for the periodic slip model. However, we are hesitant to use it because of the small sample size. Most studies, where COV were applied to distinguish between periodic and aperiodic behavior, had at least 10, or even 25 earthquake occurrence times. In such cases, the COV are more meaningful than in the study here.

**A larger area than only the MF by rupture of the VBTF and MF.** You are right, a combined rupture would lead to a larger earthquake magnitude. And we also think that this a very important part to keep in mind. But since we focused in this paper on data for the MF and the impact of this fault to the seismic hazard, we thought that the scenario of a combined rupture of the VBTF and the MF might be beyond the scope of this manuscript. See also comment to RC1. We added section 6.5 to address this question.

**Periodic/aperiodic vs. characteristic/supercycle.** Yes, you are right, we got confused here. The reason why we mentioned it here was because it seems that these faults also are quiet for a long time and then are switched on (maybe triggered by the VBTF). Hence the comparison to the characteristic/super-cycle. We changed the introduction in this sense. See also comment to RC1.

**Creep.** Prior to our study, the MF was suggested to creep aseismically. However, we did not find any evidence for creep in the trenches. See also comment below and comment to RC4.

The **lack of a geomorphic site sketch/map** has also been mentioned by the other reviewers. We included the relevant figure as well as the topographic profile and the landscape picture as suggested and added the names of the towns (see Figure 4). See respective responses to RC1 and RC4.

**Far field displacement.** Separation of red horizon is due to the sedimentation on pre-existing topography. Besides this, the total topographic step is about 17 m (as visible in Figure 3), but in the trenches, there is no sign for afterslip or interseismic creep.

**Threshold for surface rupture in this area and their discoverability.** We added a section about the completeness and preservation of earthquake records to discuss this in greater detail (section 6.1).

**Mmax as minimum Mmax.** Yes, you are right. Of course, this is the Mmax that at least should be considered, based on the data presented here. If interaction between faults and rupture of several segments is considered, the Mmax for the Vienna Basin would be definitively increase. We added a section about Mmax to discuss this in greater detail (section 6.5). See also respective comment to RC1.

**Figures:** We went through the figures with the annotations in mind, removed all inconsistencies, added and explained labels, where missing, changed Figure 10 (now Figure 11) to add the OxCal results. We finally added the uninterpreted photomosaiques to the supplementary.

**4. Responses to RC4 (Maria Ortuño):**

First of all, thank you for the detailed review and the suggestions. It was an interesting process to digest the points raised by you, but it helped to address some topics (e.g., gelifluction process, colluvial wedge formation) that we have overlooked to explain previously. In addition, your comments helped immensely to erase any inconsistencies from the figures and to improve them for easier understanding.

**Description of wedges and their generation rather by gelifluction processes then by tectonic processes:**

During logging and interpreting the trench exposures, we have been very aware of the difficulties to distinguish between gelifluction and tectonic processes, especially because both processes may interfere during periglacial climatic conditions in the Vienna Basin. We have considered this for all the colluvial wedges that they were of non-tectonic origin, but all wedges are either bound by faults or cover faults. In trench SDF1, most of the colluvial wedges are covering tension cracks that are filled with similar material, but showing less overall orientation. The combination of both, the chaotically filled tension cracks together with the colluvial on top of them indicated their tectonic origin for us. Nevertheless, you are right, the process of transporting material from the footwall to the hanging wall might have partly been gelifluction.

**The implications to seismic hazard:**

**1: Periodic vs. clustered behavior:**  We thought that we have discussed the implication that in the case of clustered earthquakes, the intervals between the clusters should be taken instead of the average recurrence intervals between single earthquakes.  We also talked about and calculated the time interval between both clusters to between 32 and 41 ka in section 6.1. (now section 6.2).  But you are right, we agree that this an important message and we have stressed this out in greater detail in the conclusions. Due to the comments of the other reviewers, we have rewritten section 6.1 (now section 6.2) and highlighted the differences between periodic and clustered earthquakes. See also comments to RC1 and RC2.

**2: Primary vs secondary ruptures:** This is definitely a noteworthy topic for discussion that we did not yet discussed in detail. So, thank you for mentioning the topic. This topic is twofold: First, if there are more fault branches reaching the surface during an earthquake apart from the main fault zone exposed in the trenches. We can exclude that on the observations made in the pipeline trench (WAG) that crosses almost the entire area from E to W and proofs that faulting is only observed within the 1-2 m wide zones, just as in the paleoseismological trenches. The trenches were also about 40 long, but there was no additional faulting observed.  Second, the observed surface rupture is secondary faulting to earthquakes along the VBTF, which in turn would be a much larger earthquake than just the rupture along the MF. We added a section about Mmax in the Vienna Basin, where we also mentioned a possible secondary rupture along the MF (section 6.5).  This was also mentioned by the other reviewers and we answered it further below.

**3: Ice loading.** We did not discuss this because of the reasons listed below. For the Scandinavian ice shield, the effect would be quite low and would only accounting for the youngest (and smallest) earthquake (around 14 ka). The Alpine ice shield was too small to contribute to a significant loading and the Vienna Basin by itself was not glaciated during the Quaternary. However, normally, this effect is mostly seen for reverse faults and not for normal faults (like the MF).

**Paleoseismological data:**

**1: More detailed geomorphic map.** This has been also mentioned by the other reviewers. As mentioned before, we thought Figure 3 would be enough to show the surrounding of the trenches. However, we do see your point and have added figure of the geomorphic/geological situation around the trenches (Figure 4).

**2: Picture of the landscape.** We added such a pictures to the figure mentioned above (Figure 4).

**3. Subunits in logs, location of deformed units, and references in the text.** We changed the trench logs following your suggestions and checked the text to include more references to the figures (check Figures 5 and 7). We added the uninterpreted photo mosaics to the supplementary.

**4. Event horizons.** We did not include event horizons in the trench logs because for most of the faulting events, the event horizon can be only seen in the hanging wall. Therefore, we followed rather the suggestions of RC2 to label the colluvial layers that indicate deposition close after the earthquakes. We hope that marking the colluvial layers in the trench logs help to identify the single earthquakes.

**5. Structure of trench log description.** Rereading this section, we know that the section description for trench SDF1 and SDF3 look differently. We changed the text and included the colluvial layers in the stratigraphic description of SDF3.

**6. Deformation bands.** Deformation bands by themselves are defined as small-scale faults with no visible displacement or with displacement in the range of mm. So, the term is used here correctly, because we want to describe exactly those small lines especially visible in sand layers because of their reduced compaction. Maybe there is a misunderstanding, but the deformation bands are not dipping necessarily parallel to the fault zone (they do so in trench SDF1), but are outside the narrow fault zone. We checked the text to avoid any misuse.

**7. Description of earthquakes in WAG.** We do recognize that this section is too short, especially regarding the event description. We added a more detailed description for the evidence exposed there. However, since this was a construction pit with limited access, exposure and description is rather thin compared to the trenches SDF1 and SDF3.

**Different material in hanging and footwall.** Yes, we do think that the fault acted as physical barrier for deposition of the fine-grained sediments in the hanging wall, that we interpret mostly as sediments that have been deposited by the River Danube during flooding events. We briefly addressed that in the trench description, but also added that to the interpretation section of the trenches to make it clearer.

**Dating results.** As suggested by RC1, we moved the dating description to the newly added methodology section (section 3). We added also a subsection about age control to the trench descriptions. This should also solve this problem addressed here.

**Paleoseismological discussion:**

**1) Event definition.** We do see your point of firstly addressing the bracketing units and changed the relevant sections accordingly. We though that this is clearly seen the trench logs, but of course you are right, it is better to explicitly mention it in the text. So, we changed it and added an "Age control" section to the trench descriptions (section 4). Figure 10 (now Figure 11) has been changed to accommodate the OxCal results as suggested by RC1 and RC2.

**2) Mmax.** We are a little confused by this comment, so I hope that I address it correctly. In the first section 6.3 (sorry for the typo, now 6.4), we do compare the maximum magnitude from the trenches (derived from inferred surface displacement, 6.8 ± 0.1) to the magnitudes derived from the fault length and from the fault area (6.7 ± 0.3). In order to make it clearer for the reader, we added the resulting magnitude to table 3 and referenced it to this section. However, we do prefer to keep the discussion of each earthquake together. Regarding the use of Wells & Coppersmith (1994): We are aware that the use of this correlations is slightly outdated, but on the other hand, most paleoseismologists in Central Europe have used those equations to estimate the magnitudes. Therefore, we decided to use the same equations for better comparison of the events within Central Europe. Nevertheless, for further recalculations, we added the observed displacements that are used for the calculations (table 2).

**3: Periodic vs. clustered behavior.** See comments below and above.

**4: Linkage to the Vienna Basin Transfer Fault**. The other reviewers also raised this question. We do think that the MF is connected to the VBTF via the common detachment, and we also mention it shortly that in the discussion about the possible activation of the detachment during an earthquake along the MF. The topic about primary vs. secondary faulting is very interesting one, and a topic to explore in further studies. However, at the moment, the data presented here strongly suggest the inclusion of the MF as primary earthquake source. We cannot, and don't want to, exclude the possibility that the MF is also activated as secondary source for the VBTF. But since we focused in this paper on data for the MF and the impact of this fault to the seismic hazard, we thought that the scenario of a combined rupture of the VBTF and the MF might be beyond the scope of this manuscript. We added a section about the Mmax and discussed here also the linkage between both faults (section 6.5). See also comment to RC1 and RC2.

**Comments in the supplementary:**

**Abstract:**

**1: conservation potential of earthquake surface ruptures smaller than 6.5.** Yes, we think that we have shown and discussed that at the beginning of section 6.1, in respect to the exclusion of E1 from the recurrence interval calculation. This might be different in areas with finer sedimentary record, but here in the setting of our trenches, we think this is valid conclusion to draw. We rewrote the section in order to state this more clearly (see the new section 6.1).

**2: Magnitude estimates.** This is discussed in the second section 6.3 (now 6.4, sorry again for the typo). The largest inferred surface ruptures in both trenches are up to 2 m, suggesting an earthquake around magnitude M~7.0. The magnitude can also derived from the rupture area of an earthquake, not only the length. The fault area of the MF without the detachment area would be a little too small to generate an earthquake of such size. Including the detachment area, the resultant magnitude would fit better to the magnitude observed from the surface displacement. We know that this is not a fact, but we think that it is a possible valid interpretation.

**Geological setting:**

**Historical earthquakes.** In principal, you are right. The uncertainties are too large to exclude the activation of the splay faults via small historical earthquakes, especially since earthquakes seem to cluster close to the areas where the splay faults connect with the VBTF. So, it would be possible that there have been small earthquakes at the southern tips of the splay faults. However, north of the River Danube, close to and in Vienna, where the splay faults have their largest throw (shown in industrial

seismic data), there is a significant lack of earthquakes, and no historical earthquakes. The few earthquakes there are all instrumental recordings and not larger than ML=3.0. We changed the sentence to avoid further confusion.

**MF as creeping fault vs. small earthquakes.** Thank you for the comment. We did not realize this paradox and changed the manuscript accordingly.

**Paleoseismologically characterized faults:** There are none so far, except the Aderklaa-Bockfliess fault, addressed in Weissl et al. (2017). The trenching there did not exposed the fault. The offset of the Quaternary was inferred from geoelectrical data. We stated this more clearly in section 2.

**Trenching results:**

**SDF1.** We took your suggestions (also see below) and have rewritten the trench description by using a simpler structure and referencing to the figures, where applicable. See also general comments on paleoseismological data above.

**Gelifluction vs. colluvial wedge.** Thank you for raising this question here, because the differentiation between gelifluction and tectonic processes is a task that we have been challenged with several times. Interestingly, we did not find evidence for gelifluction in trench SDF1. The colors are post-deposition. However, in trench SDF3 and in another trench in a similar setting (which is not ready to be published yet), we have seen colluvial wegdes that have been affected by gelifluction. These look very different from the wedges in SDF1. We did also not find overturned faults as are typical for fault zones affected by gelifluction. As far as we are aware of, colluvial wegdes can be also formed by (episodic) erosion from the foot wall towards the hanging wall. And this what we think happened here. This would also lead to a layered wedge, but bound at least partly by the fault, which is exactly what we have seen in these trenches here. The initial, more chaotic layering is observed in the underlying tension cracks. We have rewritten the trench description. We hope that with the improved version of the manuscript, the evidence for tectonic origin of the colluvial wedges is better presented.

**Tension cracks / filled fissures**. We homogenized the terms. The infill consists of the same material as the overlying wedge, but with less oriented. This is one of the reasons why we favor the interpretation of colluvial wedges instead of gelifluction.

**Section 3.1.1.** According to your earlier comments, we changed this section to provide more information about the bracketing units. However, there are 4 colluvial wedges associated with chaotically filled tension cracks (A2-A5) plus the displacement caused by the youngest earthquake (A1). That are 5 earthquakes. I think there is a misunderstanding, because we state the 4 colluvial wedges are evidence for 4 earthquakes, and then the displacement of the youngest colluvial wedge caused by another earthquake. We changed the wording to avoid further misunderstandings.

**Section 3.2.1.** Yes, that is the observation that we wanted to describe. We changed the wording to avoid any confusion. We included the fault numbers into the log figure.

**WAG trench.** As mentioned above, we do recognize that this section is short, especially regarding the event description. This outcrop being a construction pit with limited access, exposure and description is rather thin compared to the trenches SDF1 and SDF3. However, we added a more detailed description for the evidence exposed there. We better stress out the most important observation which is that the fault displaces the loess, reaches the surface, and cuts off the terrace (now Fig. 10A). At the beginning,

we were confused what you mean with folding, but then we understand how you came to the conclusion, but folding was not observed there. We hope that with the new description and the improved figures, the situation is better to understand for outsiders.

**Luminescence data.** As already mentioned above, we followed the suggestions of RC1 and added a section about methodology and moved the description of the dating technique and protocol. We added the uncertainties to table 2 and discussed the meaning of the uncertainties.

**Figures:**

We took all your suggestions and improved the figures accordingly. Thank you for pointing out the parts that needed improvement. As mentioned above, we included uninterpreted photo mosaiques in the supplementary.

**Implications from palaeoseismological investigations at the Markgrafneusiedl Fault (Vienna Basin, Austria) for seismic hazard assessment**

Esther Hintersberger[1], Kurt Decker[1], Johanna Lomax[2,3], Christopher Lüthgens[2]

[1]Department of Geodynamics and Sedimentology, University of Vienna, 1090 Vienna, Austria
[2]Institute of Applied Geology, University of Natural Resources and Life Sciences (BOKU), 1190 Vienna, Austria
[3]Department of Geography, Justus Liebig University Gießen, 35390 Giessen, Germany

*Correspondence to*: Esther Hintersberger ()(esther.hintersberger@univie.ac.at)

**Abstract.**  Intraplate regions  characterized by low rates of seismicity,  are challenging for seismic hazard assessment, mainly for two reasons: Firstly, evaluation of historic earthquake catalogues may not  reveal all active faults that contribute to regional seismic hazard. Secondly, slip rates determination is limited by sparse geomorphic preservation of slowly moving faults. In the Vienna Basin (Austria), moderate historical seismicity (Imax/Mmax = 8/5.2) concentrates along the left-lateral strike-slip Vienna Basin Transfer Fault (VBTF). In contrast, several normal faults branching out from the VBTF show neither historical nor instrumental earthquake records, although geomorphological data indicate Quaternary displacement along those faults. Here, we present a palaeoseismological dataset of three trenches crossing one of these splay faults, the Markgrafneusiedl Fault (MF), in order to evaluate its seismic potential . Comparing the observations of the different trenches, we found evidence for 5-6  surface-breaking earthquakes during the last 120 ka, with the youngest event occurring at around ~14 ka ago. The inferred surface displacements lead to magnitude estimates ranging between 6.2±0.3 and 6.8±0.1. Data can be interpreted by two possible slip models, with slip model 1 showing more regular recurrence intervals of about 20-25 ka between the earthquakes with M≥6.5, and slip model 2 indicating that such earthquakes cluster in two time intervals in the last 120 ka. Direct correlation between trenches favours slip model 2 as the more plausible option. Trench observations also show that structural and sedimentological records of strong earthquakes with small surface offset have only low preservation potential. Therefore, the earthquake frequency for magnitudes between 6 and 6.5 cannot be constrained by the trenching records. Vertical slip rates of 0.02-0.05 mm/a derived from the trenches compare well to geomorphically derived slip rates of 0.02-0.009 mm/a. Magnitude estimates from fault dimensions suggest that the largest earthquakes observed in the trenches activated the entire fault surface of the MF including the basal detachment that links the normal fault with the VBTF. The most important implications of these paleoseismological results for seismic hazard assessment are that: (1) The MF is a capable seismic source despite the lack of historical earthquakes. (2) The maximum credible

earthquakes in the Vienna Basin should be considered to be at least about M=7.0. (3) The MF is kinematically and geologically equivalent to a number of other splay faults of the VBTF. It must be assumedis reasonable to assume that these faults are potential sources of large earthquakes as well. The frequency of strong earthquakes near Vienna is therefore expected to be significantly higher than the earthquake frequency reconstructed for the MF.

**1 Introduction**

During the last few years, earthquakes tendhave tended to "surprise" seismologiststhe scientific community, either by unexpectedly high magnitudes (e.g., Sumatra Earthquakes 2004, Tohuko Earthquake 2011) or/and by the fact that the generating faults were either unmapped (Christchurch Earthquake 2010) or assumed to be inactivenot considered in hazard

10   assessments (e.g., Haiti Earthquake 2009). 2010). Thus, it seems to be clear that historical and instrumental seismicity data are not sufficient to fully characterize the seismogenic potential of a certain region (e.g., Camelbeeck et al., 2007, Liu et al., 2011). Especially, in regions of low to moderate seismicity, mostly in intraplate settings, observations of historical and instrumental seismicity are not sufficient to accurately estimate the rate of earthquake activity (Liu 
[revised manuscript text omitted]
 another colluvial wedge W4, which was most probably created by movement along F2 during earthquake B4. W5 and W4 are bracketed between the hanging wall units U4 and U5. Evidence for  event B3 is a tension crack T3 between F2 and F2', that is also identified by

20 the thin sand layer that is smeared into the crack, parallel to F2'. ~~B2 is identified by a ~ 0.8 m thick colluvial wedge.~~W3 is most probably deposited before the occurrence of B3, and the material was smeared into the fault zone along F2' during B2. As the deposits of W3 partly cover the deposits of U5, the occurrence of B3 must be later than U5. The primary evidence for B2 is the ~ 0.8 m thick colluvial wedge W2 (Figure 7, 8A). However, the fault strand bounding this colluvial wedge is not obvious. This situation may be explained by the following scenario: In the case that coseismic surface rupture offset unconsolidated

25 water-saturated sandy gravel, it seems plausible that no long-standing free surface and colluvial wedge adjacent to the fault plane could form. Instead, the offset soft sediment may have collapsed during or shortly after the earthquake forming a wedge-shaped deposit, which overlies the uppermost part of the ruptured fault. The same geometry may result from geli-solifluction under periglacial conditions when material glides down to the hangingwall destroying a free surface previously formed by B2. The latter scenario is supported by the observation of a smooth change between the horizontal layers of the

[revised manuscript text omitted]

**4.2 Sedimentary and tectonic context**

In general, luminescence dating results fit well to the stratigraphic hanging wall sedimentary sequences observed in both trenches, showing continuous decrease in age from the bottom towards the top. In addition, ages derived for the Gaenserndorf terrace in the footwall fit well with other ages from this terrace (Weissl et al., 2017).

Regarding to trench SDF1, Event A1 is well constraint between 13.8 ± 1.4 ka and 16.3 ± 1.8 ka by samples from an offset sand layer and overlying undeformed sediments. Events A2 and A3 are bracketed by the ages inferred for A1 and the undeformed sediments below the colluvial wedge related to A3 and therefore occurred between 16.1 ± 1.7 ka and 48.9 ± 4.8 ka. The ages of A4 and A5 are similarly constrained in the trench SDF1 by sediments below and above the colluvial wedges. Both events occurred between 56.6 ± 5.7 ka and 104 ± 12 ka.

In trench SDF3, samples (AIP93-AIP102) defining the chronology of the stratified hanging wall between 158 ± 21 ka and 4.8 ± 0.5 ka were dated in addition to two more samples (AIP103 and AIP114) determine the minimum age of the footwall to 205 ± 37 - 259 ± 35 ka. Those obtained ages agree well with other IRSL ages for the Gaenserndorf terrace (Weissl et al., 2017). In addition, IRSL data of the hanging wall constrain roughly the occurrence times of the 5 observed paleo-earthquakes along the main fault. While B1 is constrained to have occurred between 4.8 ± 0.5 ka and 32.9 ± 4.1 ka, B2 and B3 can only limited to occur together within the time interval between 32.9 ± 4.1 ka and 70.8 ± 8.0 ka. Also for B4 and B5, a common time interval between 111 ± 12 ka and 123 ± 16 ka can be determined. At the trench site WAG, both the uppermost and the lowermost sediments of the fine-graded silty to sandy cover were dated by IRSL revealing ages of 15.06 ± 1.52 ka and 16.1 ± 1.7 ka, respectively (samples AIP25, 26, Weissl et al., 2017).

**5 Correlation of events between sites**

Palaeoseismological investigations along the MF include three locations, the trenches SDF1 and SDF3 as well as the pipeline outcrop WAG. For all three locations, detailed mapping and dating have been carried out and described above. Evidence for 5five possible earthquakes have been observed in the trenches SDF1 (named A1-A5) and SDF3 (B1-B5), while in the pipeline

5   outcroptrench WAG, observations indicate two paleo-earthquakes (C1 and C2). Table 2 summarizes the observations, time constraints and occurrence times for each earthquake and each trench separately. The central panel of Figure 1011 shows the constraints of earthquake occurrence times for each observation point.trench separately. Based on this information, together with comparison of trench observations and displacement estimates for all three outcrops, we correlate thetrench observations and occurrence timing results to generate a synthesis of the earthquake occurrence along the MF. In the following, we discuss

10  each earthquake and the possible correlations between the trenches as well as the resultant ageoccurrence time, displacement and magnitude estimate, starting with the youngest. The summary of this discussion is shown in Table 3.

**5.1 Event 1E1 (A1 = B1 = C1)**

In all three outcropsexcavations, the youngest event is evident from a measurable offset of layers across the MF. At trench site SDF1, the youngest event A1 shows displacements of 15-25 cm and occurred in the(Figure 6A). Its occurrence time rangeis

15  bracketed between $13.8 \pm 1.4$ ka and $16.3 \pm 1.8$ ka. At trench site SDF3, markers have been displaced by the youngest event B1 by 10-15 cm. The (Figure 8A). ISRL datadating results limits the occurrence time of B1 to the time range between $4.8 \pm 0.5$ ka and $32.9 \pm 4.1$ ka. In the pipeline outcroptrench WAG, the loess cover is dated between $15.1 \pm 1.5$ ka and $16.1 \pm 1.7$ ka. It is displaced by 17-20 cm., visible by a white marker within the loess (Figure F10A). Therefore, C1 must have happened after $15.1 \pm 1.5$ ka.

20  TheCombination of constraints for E1 from all three sites yields to an occurrence time for E1 is thus constraint to the time interval between 13of $14.2 \pm 0.8 \pm 1.4$ ka and $15.1 \pm 1.5$ ka (Table 2).. Using the empiricempirical relationship between surface displacement and magnitude (Wells and Coppersmith, 1994) for the maximum displacement of 25 cm, E1 had the magnitude estimate for E1 is $M(d_{max}) = 6.2 \pm 0.25$. The average displacement of 17 cm would lead to a similar magnitude Mestimate of $M(d_{ave}) = 6.3 \pm 0.36$.

**5.2 Event 2E2 (A2 = B2 = C2)**

Event E2 is also observed inat all three outcropssites as a triangular-shaped colluvial wedge mainly consisting of reworked gravels that derived from the terrace in the footwall. (Figures 6B, 8A, 9). In addition, the top of each of those colluvial wedgewedges deposits isrelated to A2 and B2 are displaced by the younger event E1, confirming the correlation of the colluvial wedges to the penultimate seismic event E2. At trench site SDF1, the displacement related to A2 (1.5-1.9 m) is estimated from

30  the height of the associated colluvial wedge (0.75 - 0.95 m). IRSL samples from sediments above and below the colluvial wedge constrain the occurrence time for A2 to the time interval between $16.1 \pm 1.7$ ka and $48.9 \pm 4.8$ ka. At trench site SDF3,

the interpretation of deposits related to B2 are more ambiguous (see sec. 3.2), and therefore, the estimated displacement is either 0.8 m (collapsed free face scenario) or 1.6 m (colluvial wedge scenario). B2 is constrained between 32.9 ± 4.1 ka and 70.8 ± 8.0 ka. by the IRSL ages of sediments above and below the colluvial wedge material. In the pipeline construction pittrench WAG, the displacement of E2 can only beis not constrained to exceed 1 m by the colluvial wedge as, because the

5    base of the wedge iswas not exposed. Time constraints are limited to the ante quemexcavated. The maximum age of 16.1 ± 1.7 ka byfor the age of the overlying loess covering the colluvial wedge. gives a minimum age for the occurrence of C1. Combining the individual time constraints in eachfrom all trench sitesites allow to determine the occurrence time of E2 between 32.9to 37.1 ± 4.1 ka and 9 48.9 ± 4.8 ka. Magnitude calculation using the maximum of the observed surface displacements results in a magnitude of M(d$_{max}$) = 6.8 ± 0.1)4 (Wells and Coppersmith, 1994). With the maximum value for the observed

10   surface displacement coming from trench SDF1, the magnitude estimate does not depend on the interpretation for the B2 deposits in trench SDF3.

**5.3 Event 3E3 (A3, probably correlated with B3)**

For this event, a correlation based on field observations between the trenches SDF1 and SDF3 is not as clear as in the cases of E1 and E2, especially since the evidence for B3 does not allow to determine a displacement for this possible event. However,

15   the maximum height of a well-developed sandy colluvial wedge in SDF1 gives a good estimate of an earthquake with M(d$_{max}$) = 6.6 ± 0.1.4 (Figure 6C). Because of the similar stratigraphic constraints, the combined possible occurrence time of E3 is constrained by the same limits as E2, soat 43.4 ± 4.9 ka overlaps with that E3 occurred also between 32.9 ± 4.1 ka one of E2 (Figure 11, lower and 48.9 ± 4.8 ka.upper panels).

**5.4 Event 4E4a (A4, if correlated with B3)**

20   AnotherDue to the loose time constraint of B3, another possible correlation scenario between the trench sites SDF1 and SFD3 is the correlation of events A4 and B3 (event lineslip model 1 in Figure 10), mainly due the loose time constraint of B3.11, upper panel). If A4 and B3 are correlated to the same seismic event E4E4a, the overlap of possible occurrence times of A4 and B3 narrows the resultant occurrence time for E4E4a to the interval between 56.664.9 ± 5.7 ka and 70.8 ± 8.06 ka. Observations of the maximum wedge height at trench site SDF1 indicate the magnitude of A4 (and therefore for E4E4a) to

25   M = 6.8 ± 0.25.

**5.5 Event 5E4b (A4, if correlated with B4)**

In an alternative scenario, A4 could also correlate to B4 (event lineslip model 2 in Figure 1011, lower panel). In this case, the combined occurrence time for the resultant seismic event E5E4b must be older than the 111 ± 12 ka old sediments below the wedge associated with B4 in trench SDF3 (Figure 7) and younger than the lowest sediments in trench SDF1 dated to 104 ±

30   12 ka. (Figure 5). Thus, the time constraintconstraints for E4b would be thigh, dating E5 to the overlaplead to a narrow time

window for its occurrence of 106 ± 3 ka . Similar to E4a, the magnitude for E4b can be estimated from the observations of the maximum wedge height of A4 at trench site SDF1, indicating a magnitude for E4b of M = 6.8 ± 0.5.

**5.6 Event E5a (possible correlation between A5 and B4) and Event 5b (possible correlation between A5 and B5)**

5      The further back in time, the more uncertain the correlation between both trench sites becomes. So, whether A5 and B5 are correlated to one event E5a, or A5 and B4 are correlated to one event E5b is not clearly determined neither by observations nor dating. Both alternatives are possible and only depend on whether A4 is correlated to B3 or to B4. In slip model 1 (Figure 11, upper panel), where A4 is correlated with B3, it appears reasonable to assume subsequently that A5 equals to B4. In contrast, if A4 is correlated with B4 (slip model 2, Figure 11, lower panel), the remaining
10    correlation for the next older event would be that A5 corresponds to B5. Due to the loose time constraints in the lower part of all trenches, the  available chrono-stratigraphic constraints for E5a and E5b are identical to those used for E4b (Figure 11, central panel). However, additional constraints of chronological order demand that E5a and E5b must have happened earlier than E4a and E4b, respectively. This yields to slightly different occurrence times of 107 ± 4 ka and 109 ± 3 ka for E5a and E5b, respectively.
15    Since both magnitude estimates for E5a and E5b are based on the maximum colluvial wedge height of A5 from trench SDF1, the magnitude of E5a and E5b is M = 6.5 ± 0.4.

**5.7 Event E6a (B5, if not correlated with any event in SDF1)**

For the case that B5 is not correlated with any events recorded in trench SDF1 (slip model 1), the timing of E6a would be bracketed by the IRSL dating results of 111 ± 12 ka and 123 ± 16 ka.
20    In addition, with A5 (=E5a) being the oldest earthquake evidence in SDF1, E6a might be older than the oldest deposits in SDF1 and therefore not visible there. Both time constraints yield to an occurrence time of 120 ± 7 ka for E6a. Unfortunately, the magnitude of this possible seismic event cannot be constrained by trench observations.

**6 Seismotectonic implications**

**6.1 Completeness and preservation potential of earthquake records along the MF**

25    The comparison between the evidences for youngest surface breaking earthquake E1, which did not lead to the formation of colluvial wedges, and the older events, which are characterized by larger offsets and colluvial wedges throughout, raises the question on the completeness of the paleoseismological earthquake record. It seems that the colluvial wedges associated with larger earthquakes that rupture the same surface-breaking fault conceal or even erase evidence for samller offsets by earthquakes such as E1. The displacement of the markers by E1 is only preserved because the event happened after the last

earthquake (E2) that caused a colluvial wedge to form. Any future event with a surface displacement that is large enough to lead to the erosion of the offset markers in the footwall will destroy the evidence for E1.We conclude that this restriction also applies to earthquakes with small surface offset that occurred prior to E2. Therefore, earthquake records for magnitudes less than about 6.5 are most probably incomplete, and thus excluded from the recurrence calculation.6.1

**6.2 Recurrence intervals for earthquakes with magnitudes larger than 6.5 along the MF**

The possible correlations of paleoearthquakes between the trenches allow for two different interpretations to reconstructassess the recurrence intervals of earthquakes with magnitudes larger than M = 6.5. The event E1 will be excluded during the following considerations, since the related magnitude estimate is lower than those obtained for E2-E7. It seems that the colluvial wedges associated with the larger earthquakes conceal or even erase evidences for offsets formed by smaller earthquake. The displacement of markers related to E1 is only conserved because the event happened after the last earthquake that caused a colluvial wedge to form (E2). Any future event with a surface displacement that is large enough to lead to the erosion of the offset markers in the footwall will destroy the evidence for E1. This restriction also applies to earthquakes with small surface offset that occurred prior to E2.E5a/b. Therefore, earthquake records for magnitudes less than about 6.5 are most probably incomplete, and thus excluded from the recurrence calculation.

As mentioned in sect. 5, the crucial part for the reconstruction of recurrence intervals therefore is whether A4 is correlated either to B3 or to B4 and, subsequentlyconsequently, whether A5 is correlated to B4 or to B5, resulting in the following event linesslip models:

(Slip model 1): E2-E3(not correlated to B3)-E4-E6-E8E4a-E5a-E6a (5 earthquakes);

(Slip model 2): E2-E3(correlated to B3)-E5-E7E4b-E5b (4 earthquakes).

The determination of inter-event intervals is based on the limits for the occurrence time intervals for each earthquake as given in Table 2. Figure 10 shows clearly that both event lines represent different types of distributing earthquakes. Event line 1 represents an approximately periodic reoccurrence of earthquakes with magnitudes larger than M = 6.5. The maximum time interval between E2 and E3 is 15.8 ka. As the occurrence time of E3 limits the occurrence time interval of E2, the minimum time interval cannot be calculated. Considering event line 1 (E2-E3-E4-E6-E8) and the range of uncertainties related to dating, the inter-event time between E3 and E4 lies between 6.3 ka and 41.5 ka, while the inter-event time between E4 and E6 is between 20.2 ka and 65.1 ka. Finally, the maximal inter-event time between E6 and E8 is constrained to 40 ka. Similar to the inter-event time for E2/E3, a minimum inter-event time for E6/ E8 cannot be calculated. Taking all information together, the average of the minimum values and the average of the maximum recurrence intervals for event line 1 would be then ~ 13 ka and ~ 40 ka, respectively.

On the other side, earthquakes in event line 2 (E2-E3-E5-E7) seem to cluster in time. Therefore, instead of calculating inter-event times for all earthquakes, we calculate the minimum inter-cluster time that is identical with the inter-event time for E3/E5, and the maximum intra-cluster times for E2/E3 and E5/E7, meaning the largest possible time between both

earthquakes within the same cluster. A maximal inter-cluster time cannot be given, due to the poor time constraint within the both clusters. However, the maximal intra-cluster times for E2/E3 and E5/E7 are 15.8 ka and 17.0 ka, respectively. In addition, the minimum inter-cluster time interval between E2/E3 and E5/E7 is at least 54.4 ka. The time since the occurrence of E2 until today may be also considered as a minimum inter-cluster time, being at least 32.9 ± 4.1 ka and maximal 40.9 ±

[revised manuscript text omitted]

**Tables**

| Event # / Sample | Evidence / Location (displ) | Depth (m) | Moisture (%) | 238U (ppm) / 0.15-0.25 m | 232Th (ppm) | K (%) | De (Gy) | D₀ (Gy/ka) | Age (ka) |
|---|---|---|---|---|---|---|---|---|---|
| A2/AIP25* | WAG cover ew,te | 1.0 | 12 ± 7 | 2.58 ± 0.95 m04 | 8.42 ± 0.75 m23 | 1.12 ± 0.02 | 44.7 ± 2.3 | 2.78 ± 0.26 | 16,1.50 ± 1.90 m7 |
| A3/AIP26* | WAG cover ew,te | 0.45 m5 | 12 ± 7 | 2.35 ± 0.40 m04 | 7.62 ± 0.21 | 1.01 ± 0.01 | 39.0 ± 1.4 | 2.01 ± 0.80 -0.90 m16 | 15.1 ± 1.5 |
| A4/AIP38* | SDF1 fw ew,te | 3.7 | 10 ± 7 | 0.75 m82 ± 0.02 | 3.54 ± 0.72 m11 | 1.40-1.50 m17 ± 0.02 | 543.1 ± 33.7 | 2.13 ± 0.20 | 255 ± 29 |
| A5/AIP39* | ew SDF1 hw | 3.6 | 10 ± 7 | 1.78 ± 0.25 m03 | 7.56 ± 0.40 m21 | 1.18 ± 0.02 | 273.5 ± 20.2 | 2.64 ± 0.50 -0.80 m25 | 104 ± 12 |
| B1/AIP40* | SDF1 hw displ | -3.3 | -10 ± 7 | 1.84 ± 0.10-0.15 m03 | 7.64 ± 0.21 | 1.24 ± 0.02 | 154.0 ± 5.5 | 2.72 ± 0.25 | 56.6 ± 5.7 |
| AIP41* | SDF1 hw | 2.6 | 10 ± 7 | 2.69 ± 0.04 | 10.46 ± 0.28 | 1.61 ± 0.02 | 168.7 ± 5.9 | 3.45 ± 0.32 | 48.9 ± 4.8 |
| AIP44* | SDF1 hw | 1.9 | 10 ± 7 | 1.49 ± 0.03 | 5.92 ± 0.17 | 0.91 ± 0.01 | 36.7 ± 2.1 | 2.26 ± 0.22 | 16.3 ± 1.8 |
| B2/AIP46* | ew SDF1 hw | 1.2 | 0.8 10 ± 7 | 2.39 ± 0.04 | 9.12 ± 0.25 | 1.44 ± 0.02 | 0.8/43.6 ± 1.6 m | 2.01 ± 0.16 | 13.8 ± 1.4 |
| B3/AIP93 | te SDF3 hw | -4.1 | -12 ± 7 | -1.64 ± 0.03 | 8.06 ± 0.22 | 1.25 ± 0.02 | 419.2 ± 39.4 | 3.17 ± 0.30 | 158 ± 21 |
| B4/AIP95 | ew SDF3 hw | -3.1 | -12 ± 7 | -1.87 ± 0.03 | 9.41 ± 0.26 | 0.97 ± 0.02 | 317.4 ± 29.1 | 2.58 ± 0.24 | 123 ± 16 |
| B5/AIP96 | ew SDF3 hw | -2.6 | -12 ± 7 | -2.22 ± 0.04 | 8.96 ± 0.24 | 1.26 ± 0.02 | 292.0 ± 13.5 | 2.62 ± 0.25 | 111 ± 12 |
| AIP97 | SDF3 hw | 2.1 | 12 ± 7 | 2.22 ± 0.04 | 8.96 ± 0.24 | 1.26 ± 0.02 | 205.1 ± 13.3 | 2.90 ± 0.26 | 70.8 ± 8.0 |
| AIP98 | SDF3 hw | 1.8 | 12 ± 7 | 2.19 ± 0.04 | 8.28 ± 0.25 | 1.19 ± 0.02 | 91.6 ± 7.5 | 2.78 ± 0.25 | 32.9 ± 4.1 |
| AIP102 | SDF3 hw | 0.3 | 15 ± 7 | 3.01 ± 0.04 | 12.14 ± 0.33 | 1.30 ± 0.02 | 15.7 ± 0.9 | 3.29 ± 0.30 | 4.8 ± 0.5 |
| AIP103 | SDF3 fw | 1.4 | 10 ± 7 | 0.57 ± 0.01 | 1.79 ± 0.06 | 1.04 ± 0.02 | 384.6 ± 59.7 | 1.88 ± 0.18 | 205 ± 37 |
| C1/AIP114 | SDF3 fw displ | -0.65 | -10 ± 7 | 0.57 ± 0.01 | 1.79 ± 0.06 | 1.04 ± 0.02 | 468.3 ± 43.4 | 1.81 ± 0.17 -0.20 m | 259 ± 35 |
| C2 | ew | – | | – | | – | | | |

Table 1: Type of evidence and inferred displacement for the paleoearthquakes A1 to A5 (trench SDF1), B1 to B5 (SDF3), and C1 toC2 (WAG). Also listed are the thicknesses of colluvial wedges observed in the NW and SE trench walls used for estimating displacement.Table 1. Evidence: displacement of correlated layers (displ.), occurrence of colluvial wedges (cw), and sediment-filled tension cracks below the colluvial wedges (tc). Displacement is taken as twice the thickness of the colluvial wedge.

| Sample | Location | Method | De (Gy) | Dₐ(Gy/ka) | Depth (m) | Water (%) | Age (ka) |
|---|---|---|---|---|---|---|---|
| AIP25 | WAG cover | IRSL | 44.7 ± 2.3 | 2.78 ± 0.26 | 1.0 | 12 | 16.1 ± 1.7 |
| AIP26 | WAG cover | IRSL | 39.0 ± 1.4 | 2.01 ± 0.16 | 0.5 | 12 | 15.1 ± 1.5 |
| AIP38 | SDF1 fw | IRSL | 543.1 ± 33.7 | 2.13 ± 0.20 | 3.7 | 10 | 255 ± 29 |
| AIP39 | SDF1 hw | IRSL | 273.5 ± 20.2 | 2.64 ± 0.25 | 3.6 | 10 | 104 ± 12 |
| AIP40 | SDF1 hw | IRSL | 154.0 ± 5.5 | 2.72 ± 0.25 | 3.3 | 10 | 56.6 ± 5.7 |
| AIP41 | SDF1 hw | IRSL | 168.7 ± 5.9 | 3.45 ± 0.32 | 2.6 | 10 | 48.9 ± 4.8 |
| AIP44 | SDF1 hw | IRSL | 36.7 ± 2.1 | 2.26 ± 0.22 | 1.9 | 10 | 16.3 ± 1.8 |
| AIP46 | SDF1 hw | IRSL | 43.6 ± 1.6 | 2.01 ± 0.16 | 1.2 | 10 | 13.8 ± 1.4 |
| AIP93 | SDF3 hw | IRSL | 419.2 ± 39.4 | 3.17 ± 0.30 | 4.1 | 12 | 158 ± 21 |
| AIP95 | SDF3 hw | IRSL | 317.4 ± 29.1 | 2.58 ± 0.24 | 3.1 | 12 | 123 ± 16 |
| AIP97 | SDF3 hw | IRSL | 205.1 ± 13.3 | 2.90 ± 0.26 | 2.1 | 12 | 70.8 ± 8.0 |
| AIP98 | SDF3 hw | IRSL | 91.6 ± 7.5 | 2.78 ± 0.25 | 1.8 | 12 | 32.9 ± 4.1 |
| AIP102 | SDF3 hw | IRSL | 15.7 ± 0.9 | 3.29 ± 0.30 | 0.3 | 15 | 4.8 ± 0.5 |
| AIP103 | SDF3 fw | IRSL | 384.6 ± 59.7 | 1.88 ± 0.18 | 1.4 | 10 | 205 ± 37 |
| AIP114 | SDF3 fw | IRSL | 468.3 ± 43.4 | 1.81 ± 0.17 | 0.65 | 10 | 259 ± 35 |

Table 2: Infrared stimulated Luminescence (IRSL) and optically stimulated luminescence (OSL) dating results from the trenches SDF1, SDF3, and WAG at the Markgrafneusiedl Fault (MF), * refer to samples already published by Weissl et al. (2017). Location refers to either of the trenches (SDF1, SDF3, WAG) and the location in respect to the MF, where hw = hanging wall and fw = footwall. N: number of used aliquots. De (Gy): equivalent dose in Gray (Gy), Dₐ (Gy/ka): dose rate in Gray values (per 1,000 years); Depth (m): depth of the sampling location in meters below present-day surface.

| Event # | Evidence / Correlation | Ante quem — Event older than (age / sample location) Thickness of colluvial wedge @ NE wall | Postquem — Event younger than (age / sample location) Thickness of colluvial wedge @ SW wall | Displacement | Occurrence time |
|---|---|---|---|---|---|
| A1 | displ | - | - | 0.15 - 0.25 m | 15.1 ± 2.2 ka |
| E1 A2 | cw, tc / A1 = B1 = C1 | 0.95 m | 0.75 m |  1.50 - 1.90 m |  27.5 ± 13. ka (SDF3),  **15.1 ± 1.5 ka (WAG)** |
| A3 | cw, tc | 0.45 m | 0.40 m | 0.80 - 0.90 m | 38.4 ± 14.4 ka |
| E2 A4 | cw, tc / A2 = B2 = C2 | 0.75 m | 0.72 m | 1.40 - 1.50 m  |  74.0 ± 2 ± 8.0 ka (SDF3), **48.9 ± 4.8 ka (SDF1)** |
| E3 A5 | A3, ? = B3? cw | 0.25 m | 0.40 m | 0.50 - 0.80 m  90.1 ± 2 ka (SDF3),  | 70.8 ± 8.0 ka (SDF3), **48.9 ± 4.8 ka (SDF1)** |
| B1 | displ | - | - | 0.10 - 0.15 m | 18.3 ± 13.5 ka |
| E4 B2 | A4, ? = B3? cw | | 0.8 | 0.8/1.6 m | **56.6 ± 5.7 ka (SDF1)** 32 44.9 ± 17.4 ka (SDF3) | 104 ± 12 ka (SDF1), **70.8 ± 8.0 ka (SDF3)** |
| E5 B3 | ?A4 =? B4 tc | - | - | - | 111 ± 12 57 ± 18.5 ka (SDF3),  | 123 ± 16 ka (SDF3), **104 ± 12 ka (SDF1)** |
| B4 | cw | - | - | - | 115 ± 14 ka |
| E6 B5 | A5 ? = B4? cw |  | - | - | 123 ± 16 14 ka (SDF3), **104 ± 12 ka (SDF1)** |
| C1 | displ | - | - | 0.17 - 0.20 m | <15.1 ± 1.5 ka |
| E7 C2 | ?A5 = B5? cw | - | - | - | 111 ± 12 ka (SDF3), | 123 ± 16 ka (SDF3), **104 ± 12 ka (SDF1)** |

| | | | | 56.6 ± 5 >16.1 ± 1.7 ka (SDF1) | |
|---|---|---|---|---|---|
| | E8 | B5 | 111 ± 12 ka (SDF3) | 123 ± 16 ka (SDF3) | |

**Table 2: Type of evidence, inferred displacement for the paleoearthquakes A1 to A5 (trench SDF1), B1 to B5 (SDF3), and C1 to C2 (WAG) and possible occurrence time. Colluvial wedge thickness observed on NE and SW trench walls used for estimating displacement. Displacement is taken as twice the colluvial wedge thickness. Evidence: displacement of correlated layers (displ.), occurrence of colluvial wedges (cw), and sediment-filled tension cracks below the colluvial wedges (tc). 3:Occurrence times (mean ± 2σ) are calculated with OxCal using chronological constraints from respective trenches.**

| Event # | Correlation | *Antequem* Event older than (age / sample location) | *Postquem* Event younger than (age / sample location) | *Magnitude* M($d_{max}$) M($d_{ave}$) | *Occurrence interval* OxCal result (mean ± 1σ) |
|---|---|---|---|---|---|
| E1 | A1 = B1 = C1 | **13.8 ± 1.4 ka (SDF1),** 4.8 ± 0.5 ka (SDF3) | 32.9 ± 4.1 ka (SDF3), 16.3 ± 1.8 ka (SDF1), **15.1 ± 1.5 ka (WAG)** | 6.2 ± 0.5 *6.3 ± 0.6* | 14.2 ± 0.8 ka (SM1/2) |
| E2 | A2 = B2 = C2 | **32.9 ± 4.1 ka (SDF3),** 16.1 ± 1.7 ka (SDF1) | 70.8 ± 8.0 ka (SDF3), **48.9 ± 4.8 ka (SDF1)** | 6.8 ± 0.4 | 37.1 ± 4.9 ka (SM1/2) |
| E3 | A3, ?= B3? | **32.9 ± 4.1 ka (SDF3),** 16.1 ± 1.7 ka (SDF1) | 70.8 ± 8.0 ka (SDF3), **48.9 ± 4.8 ka (SDF1)** | 6.6 ± 0.4 | 43.4 ± 4.9 ka (SM1/2) |
| E4a | A4, ?= B3? | **56.6 ± 5.7 ka (SDF1)** 32.9 ± 4.1 ka (SDF3), | 104 ± 12 ka (SDF1), **70.8 ± 8.0 ka (SDF3)** | 6.8 ± 0.5 | 64.9 ± 5.6 ka (SM1) |
| E4b | ?A4 =? B4 | **111 ± 12 ka (SDF3),** 56.6 ± 5.7 ka (SDF1) | 123 ± 16 ka (SDF3), **104 ± 12 ka (SDF1)** | 6.8 ± 0.5 | 106 ± 3 ka (SM2) |
| E5a | A5 ?= B4? | **111 ± 12 ka (SDF3),** 56.6 ± 5.7 ka (SDF1) | 123 ± 16 ka (SDF3), **104 ± 12 ka (SDF1)** | 6.5 ± 0.4 | 107 ± 4 ka (SM1) |
| E5b | ?A5 = B5? | **111 ± 12 ka (SDF3),** 56.6 ± 5.7 ka (SDF1) | 123 ± 16 ka (SDF3), **104 ± 12 ka (SDF1)** | 6.5 ± 0.4 | 109 ± 3 ka (SM2) |
| E6a | B5 | **111 ± 12 ka (SDF3)** | **123 ± 16 ka (SDF3)** | - | 120 ± 7 ka (SM1) |

**Table 1: Overview of common IRSL constraint for characteristics of each possible earthquake, derived from all different sites. Agescommon. IRSL constraints in bold mark the upper and lower limit for each occurrence time. Occurrence times are calculated with OxCal (see Figure 11) using stratigraphic constraints from all sites. SM1 and SM2 refer to slip model 1 and 2, respectively. For details about correlation between the trenches, see sect. 5.**

**Figure captions**

Figure 1: (A) Active faults (black solid and dashed lines), seismicity (black circles) and Quaternary basins (light grey areas) within the Vienna Basin (Austria) plotted on a shaded DEM. Seismicity is based on the ACORN catalogue (2004). The borders of the Austrian capital, Vienna, is outlined by a dashed white line. Modified after Beidinger and Decker (2011).

5 VBTF = Vienna Basin Transfer Fault. Important splay normal faults of the VBTF: ABF = Aderklaa-Bockfliess Fault, BNF = Bisamberg-Nussdorf Fault, LF = Leopoldsdorf Fault, MF = Markgrafneusiedl Fault. White box shows the location of Figure 3A, white line the position of the cross section of Figure 2; (B) Major earthquakes from historical, instrumental and paleoseismological data in intra-plate Central Europe. Historical and instrumental seismicity is based on the CENEC Catalogue by Grünthal et al., 2009. Paleoseismological study sites are compiled from Camelbeeck and Meghraoui, 1998; Camelbeeck

10 et al., 2000; 2007; Meghraoui et al., 2001; Vanneste and Verbeeck, 2001; van den Berg et al., 2002, Peters, et al., 2005; Štěpančíková et al., 2010. Labels indicate the magnitudes of the largest paleoearthquakes observed at the respective site. Black box shows area of the close up in (A). VBFS = Vienna Basin Fault System.

Figure 2: Cross section through the Vienna Basin at its central part based on reflection seismic and deep boreholes indicating the

15 common detachment of the Alpine floor thrust, which links the splay normal faults, such as the Markgrafneusiedl Fault (MF), to the Vienna Basin Transfer Fault (VBTF). The generalized detachment corresponds to the Alpine-Carpathian floor thrust, which is reactivated by normal faulting.

Figure 3: Overview of the Markgrafneusiedl Fault (MF). (A) DEM showing the Pleistocene terraces north of the River Danube

20 dissected by faults creating fault scarps (fs). GDT = Gaenserndorf terrace, SHT = Schlosshof terrace. Dashed line: trace of the topographic profile in B, solid line: trace of the seismic line in C, white circle: locations of the villages Markgrafneusiedl (MGNS) and Gaenserndorf (GD). (B) Topographic profile (black) and cross-section indicating the base of Quaternary sediments (grey) across the MF. Note the thickness of Quaternary growth strata in the fault-delimited basin above the MF. Vertical exaggeration: 8.6 (C) seismic section across the same area showing offset along the MF and the flower structure at the Vienna

25 Basin Transfer Fault (VBTF). Vertical exaggeration at 2s TWT: 4.5. See text for details.

Figure 4:  Trench locations of SDF1 and SDF3. (A) Locations of trenches SDF1 and SDF3 relative to the margin of the Gaenserndorf terrace (light grey in DEM) and the Markgrafneusiedl Fault (MF). Depressions normal to the MF are currently dry valleys resulting from Pleistocene drainage of the terrace. Digital elevation model with a resolution of 10x10 m. (B)

30 View of the fault scarp south of the forested area where the trenches are located, looking towards the N. (C) Trench location of SDF3, looking NW towards the footwall. Trench runs parallel to the trees. (D) Trench SDF 3 during excavation, looking SE towards the hanging wall. Note the difference in elevation between standing position and trees in the back. (E) Trench location of SDF1,

looking NW towards the footwall. (F) Trench SDF1 during excavation, looking SE towards the hanging wall. Note the difference in elevation between standing position and crop in the back.

[revised manuscript text omitted]

---

## Referee Report (RR1)

Review of revised version of Hintersberger et al. 2017 "**Implications from** palaeoseismological investigations at the Markgrafneusiedl Fault (Vienna Basin, Austria) for seismic hazard assessment"

I read the three reviews and subsequent major revision of the manuscript, including the response to the reviewers and the revised paper and supplement. I think that the authors have addressed the issues raised by the three referees in a comprehensive manner and documented these changes accordingly.

Below, I will briefly comment only on the main arguments and follow up with a few minor suggestions to be addressed before publication. To do so, I summarize the reviewer's statements in black and my evaluation of their treatment by the authors in blue font color.

Three referees who are all experts in paleoseismology and familiar with trenching, earthquake histories and associated geochronology have seen the original paper. They all agree that this is an important and appropriate study to be published in *Natural Hazards and Earth System Sciences*, that it provides new paleoseismic results and will be an important contribution to the neotectonic and seismic hazard communities, including emergency managers. To push the article to it's full potential, all reviewers basically agree that the following issues need to be addressed:

**Data & Discussion**

(1) The event history and the event correlation between the sites, which is based on limited age control of bracketing units, needs some reworking to clearly outline the events by their confining units and to provide a common framework for their timing including uncertainties. To do so, two of the reviewers suggest using a probabilistic analysis with OxCal that is good practice in many paleoseismic studies worldwide.

The authors have adapted their model using the suggested OxCal model. The modeled results are comparable to the previous "hands-on" calculation, which seems to me a good indication of a robust interpretation. The new approach and results are incorporated in the associated tables, method section, and figure and the code is added to the supplement.  $\rightarrow$  Point seems fully addressed in this new version

(2) The anticipated fault behavior needs some redefinition or clarification. While the authors aim to differentiate between potential *characteristic* and *super-cycle* earthquake occurrence, the discussion and review revealed that *periodic vs. clustered* scenarios would be better end member scenarios to evaluate. Two of the referees additionally suggest to use to the coefficient of variation to statistically discriminate between the two.

The new manuscript has clarified terms and used only quasi-periodic vs. clustered behavior.  $\rightarrow$  Point seems fully addressed in this new version

(3) All reviewers wish to see a discussion (or data) on a potential linkage between the MF and the more active VBTF with the effect of such a scenario on earthquake timing and maximum magnitude for the Vienna basin.

Although only briefly, this possibility is now discussed and the consequences of potentially larger magnitudes for combined ruptures mentioned.  $\rightarrow$  Point is briefly addressed in this new version; authors give good arguments in the rebuttal letter that the data so far don't allow for a more detailed discussion of this issue.

(4) Reviewers would like to see an improved discussion on the preservation potential of the earthquake evidence, which also includes alternative interpretation (or discussion) for origin and/or overprint of colluvial wedges. This is an important issue in low-strain areas, especially under influence of geomorphic consequences of glacial-interglacial cycles.

This issue got a new paragraph in the discussion chapter  $\rightarrow$  fully addressed there and referred to also when discussing smaller ruptures in between clusters.

**Presentation & Style**

(1) The reviewers suggest that a methods section before the results will help to build a framework around the study without interrupting the text flow within the results by such information.

The new manuscript has a method chapter in a comprehensive and well-structured format  $\rightarrow$  point fully addressed.

(2) A detailed site description (in figure) needs to supplement the trenching information to provide the reader with the geomorphic understanding of the area.

The new manuscript has a new figure (Figure 4) to account for these issues. Here, however, I feel that the points are not fully taken, see also my additional comments below.  $\rightarrow$  needs revision

(3) The original (uninterpreted) photomosaics should be provided (at least as supplement).

 $\rightarrow$  Point fully addressed by incorporating the photomosaics into the supplemment

(4) The reviewers suggest a relabeling/restructuring of events to avoid misunderstandings about the number/sequence of earthquakes at or between the sites.

**Additional comments**

- (1) Response to RC4 (Ice loading). While this may not necessarily need to find its way into the paper, I would like to emphasize that there is some discussion in the literature suggesting significant rebound of the Alps after LGM deglaciation (e.g., Norton and Hampel, 2010; Mey et al., 2016). Without knowing the exact location of the Alpine extent in Austria, the wavelength of isostatic compensation might be larger and includes the previous forebulge area, where vertical adjustment might have a different effect on normal faults than known for reactivated thrust faults that have been located below the Ice. In Switzerland, indications for extended glaciations also exist for MIS 6, so a time period just pre-dating the time interval of the earthquakes dated in this study.
- (1) Please add references for the statement in the first paragraph of the introduction (i.e., the Sumatra, Tohoku, Haiti and Christchurch EQ-cases).
- (2) Same paragraph, pg. 2, lines 5-8: There is much doubling, I suggest merging these sentences.
- (3) It might be worth citing the introductory paper of the book Landgraf, A., Kuebler, S., Hintersberger, E. & Stein, S. (eds) Seismicity, Fault Rupture and Earthquake Hazards in Slowly Deforming Regions. Geological Society, London, Special Publications, 432, http://doi.org/10.1144/SP432.13 that addresses many of the mentioned issues (E. Hintersberger is one of the editors).
- (4) Section 3.3: Did you follow the approaches of DuRoss et al., or Personius et al., as suggested by the reviewers? They might be cited here.
- (5) Figure 4: I'm not sure, if this selection of field photos provided here adds much to the understanding (i.e., 4 C, D, F). 4 E is good, but could benefit from some annotation (e.g., fault scarp) – same for B. Also, perhaps a hillshade, instead of the DEM (in 4 A, but perhaps could be even considered for Fig. 3)) would bring out the geomorphology better. As a geomorphic map, it would also benefit from some annotation (perhaps having (A) as hillshade and (B) next to it as interpreted geomorphic map within the same extent (see Meriaux et al., 2014/2015 for nice examples of geomorphic mapping). Focus might be the fault scarp(s), but also incision. As the reviewer was asking for topographic profiles, such complementary profiles, derived from the DEM next to the trenches and annotated to highlight the trench dimensions and vertical separation at each site, would help the reader to better acknowledge the geomorphology of the sites. Additional comments on Fig. 4: Please provide orientation at field photos; consider moving the scale bar in A to the bottom so you can leave the coordinates in place. Indicate position and viewing direction of field photos on the overview map.
- (6) "collapsed free-face" vs. "colluvial wedge": I understand the difference from the description, but find this terminology confusing. An earthquakedriven colluvial wedge also at least partly derives from collapse of the free face (until the angle of repose is established and the diffusive

processes take over). Maybe, you can call it "nontectonic wedge" or similar?

- (7) Page 17, Line 1: Typo! Slip should be 1.8 m instead of 10.8 m also, I did not quite follow the calculation. E1 –slip (based on B1 event) was measured as dip-slip between offset markers, so I guess, you assumed a vertical fault dip to translate the 10 cm of vertical slip for E1 that you added to the 0.8 m of B2 wedge height? Then, this amount (0.1 m for E1) should not be doubled for the second scenario, adding to 1.7 m only instead of 1.8? – Please clarify.
- (8) In general, did you account for the listric shape and fault-dips of about 70° in places when calculating magnitudes from slip? I see that you consider vertical slip only, so the actual dip-slip might be slightly higher.
- (9) Figures 1,3, and 4: give data source for the DEM